# Error Broadcast and Decorrelation as a Potential Artificial and Natural Learning Mechanism

## Abstract

We introduce the *Error Broadcast and Decorrelation* (EBD) algorithm, a novel learning framework that addresses the credit assignment problem in neural networks by directly broadcasting output error to individual layers. The EBD algorithm leverages the orthogonality property of the optimal minimum mean square error (MMSE) estimator, which states that estimation errors are orthogonal to any nonlinear function of the input, specifically the activations of each layer. By defining layerwise loss functions that penalize correlations between these activations and output errors, the EBD method offers a principled and efficient approach to error broadcasting. This direct error transmission eliminates the need for weight transport inherent in backpropagation. Additionally, the optimization framework of the EBD algorithm naturally leads to the emergence of the experimentally observed three-factor learning rule. We further demonstrate how EBD can be integrated with other biologically plausible learning frameworks, transforming time-contrastive approaches into single-phase, non-contrastive forms, thereby enhancing biological plausibility and performance. Numerical experiments demonstrate that EBD achieves performance comparable to or better than known error-broadcast methods on benchmark datasets. The scalability of algorithmic extensions of EBD to very large or complex datasets remains to be explored. However, our findings suggest that EBD offers a promising, principled direction for both artificial and natural learning paradigms, providing a biologically plausible and flexible alternative for neural network training with inherent simplicity and adaptability that could benefit future developments in neural network technologies.

## 1 Introduction

Neural networks have been central to both biological and artificial intelligence research, providing key models for understanding cognitive functions. One major challenge in these networks is determining how to adjust individual synaptic weights to optimize a global objective, a problem referred to as the *credit assignment problem*. In Artificial Neural Networks (ANNs), the most common solution to this problem is the *backpropagation* (BP) algorithm (Rumelhart et al., 1986). This method involves propagating errors—calculated at the network's output—back through the network using a distinct layered pathway, employing the same synaptic values used during forward processing.

In contrast to ANNs, the global mechanisms for credit assignment within biological neural networks remain less understood. Although there are dynamical models for local synaptic changes (Magee & Grienberger, 2020), a comprehensive and biologically feasible theory of credit assignment that integrates these dynamics remains unresolved. The backpropagation algorithm, despite its effectiveness in training ANNs, is not directly implementable in biological systems. This is due to biologically implausible requirements, such as weight symmetry between forward and backward pathways (Crick, 1989), meaning that the same weights must be used in both signal transmission and error feedback—a condition not observed in biological neurons, as illustrated by Figure 1a.

To address the credit assignment problem in biological networks, researchers have proposed a set of methods collectively known as *error broadcasting* (Williams, 1992; Werfel et al., 2003; Nokland, 2016; Baldi et al., 2018; Whittington & Bogacz, 2019; Clark et al., 2021). These approaches

distribute error information throughout the network without relying on precise backward pathways or symmetric weights, thereby eliminating the weight symmetry issue inherent in backpropagation. This elimination of weight symmetry not only makes error broadcasting potentially useful for modeling biological neural networks but also offers practical advantages for hardware implementations. As recently demonstrated by Wang et al. (2024), the straightforward mechanism of error broadcasting enables efficient hardware implementations of neural networks, raising hopes for future neuromorphic systems. Despite promising developments in both theory and applications (Bordelon & Pehlevan, 2022; Launay et al., 2019), error broadcasting schemes still require further theoretical support to confirm and enhance their effectiveness in training networks.

In this context, we introduce a novel learning paradigm termed the *Error Broadcast and Decorrelation* (EBD) algorithm. The fundamental principle of EBD is to adjust the network weights to minimize the correlation between the broadcast errors and the activations of each layer. This method is grounded on two key observations: first, that the output error of an optimal minimum mean square error (MMSE) estimator is orthogonal to any nonlinear function of the input; and second, that each network layer represents a specific nonlinear function of the input. By viewing the network as a nonlinear MMSE estimator and leveraging the orthogonality property of optimal estimators, we define layer-specific training losses that adjust individual layer parameters to make their activations orthogonal to the broadcast errors. The EBD algorithm directly connects the output errors to the

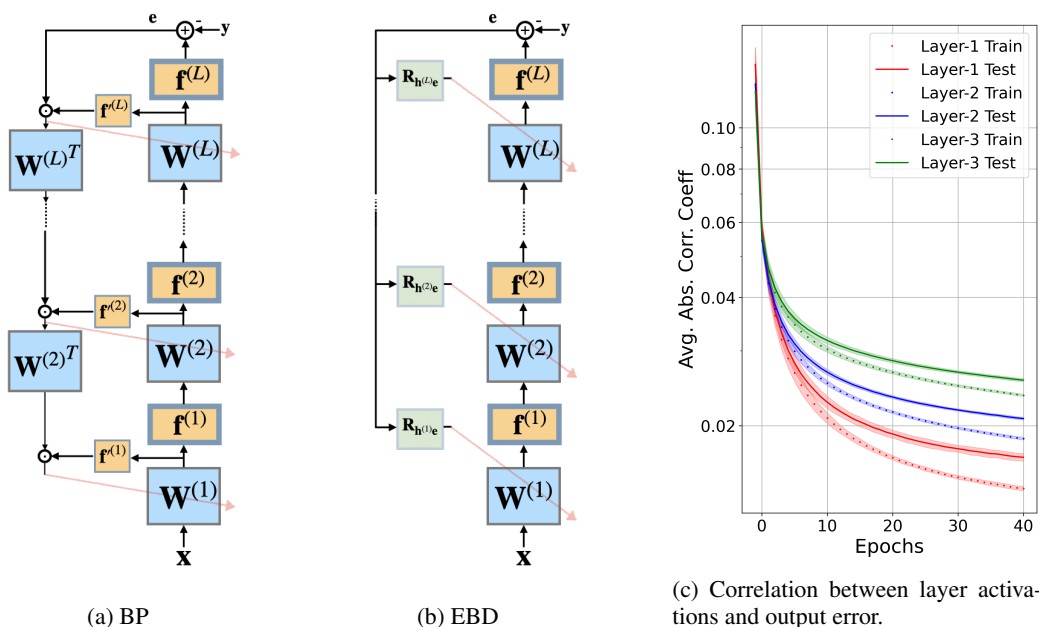

(a) BP        (b) EBD        (c) Correlation between layer activations and output error.

Figure 1: Comparison of backpropagation and error broadcast and decorrelation mechanisms in multilayer perceptron networks, along with the correlation dynamics during BP training. (a) Depicts the traditional backpropagation approach, where errors are transmitted sequentially through symmetric backward pathways. (b) Represents the Error Broadcast and Decorrelation (EBD) approach, where output errors are broadcast to each layer via cross-correlation matrices between the errors and layer activations. (c) Shows the evolution of the average absolute correlation between layer activations and the error signal during backpropagation training of an MLP with three hidden layers (with MSE criterion) on CIFAR-10 dataset, illustrating how this correlation decreases over epochs (see Appendix F for details).

network layers, simplifying the mechanism for credit assignment and enabling parallel synaptic updates that may accelerate training. In providing a framework for biologically realistic networks, the EBD algorithm has two key advantages. First, optimizing the loss function of EBD naturally leads to the experimentally observed three-factor learning rule (Gerstner et al., 2018; Kuśmierz et al., 2017). Second, by broadcasting errors directly to the layers as shown in Figure 1b, it overcomes the weight transport problem inherent in backpropagation and some more biologically plausible credit assignment approaches (Whittington & Bogacz, 2017; Qin et al., 2021).

We demonstrate the utility of the EBD algorithm by applying it to both artificial and biologically realistic neural networks. While our experiments show that EBD performs comparably to state-of-the-art error-broadcast approaches on benchmark datasets, offering a promising direction for theoretical and practical advancements in neural network training, its scalability to more complex tasks and larger networks remains to be investigated.

## 1.1 RELATED WORK AND CONTRIBUTIONS

Several frameworks have been proposed as alternatives to the backpropagation algorithm for modeling credit assignment in biological networks (Whittington & Bogacz, 2019). These include predictive coding (Rao & Ballard, 1999; Whittington & Bogacz, 2017; Golkar et al., 2022), similarity matching (Qin et al., 2021; Bahroun et al., 2023), time-contrastive approaches (Ackley et al., 1985; O'Reilly, 1996; Scellier & Bengio, 2017), forward-only methods (Hinton, 2022; Farinha et al., 2023; Dellaferrera & Kreiman, 2022), target propagation (Le Cun, 1986; Bengio, 2014; Lee et al., 2015), random feedback alignment (Lillicrap et al., 2016), and learned feedback weights (Kolen & Pollack, 1994; Ji-An & Benna, 2024).

Another significant alternative is error-broadcast methods, where output errors are directly transmitted to network layers without relying on precise backward pathways or symmetric weights. Two important examples of this approach are weight and node perturbation algorithms Williams (1992); Dembo & Kailath (1990); Cauwenberghs (1992); Fiete & Seung (2006), in which global error signals are broadcast to all network units. These signals reflect the change in overall error caused by individual perturbations in the network's weights or units. A more recent and prominent example of error-broadcast is Direct Feedback Alignment (DFA) (Nokland, 2016). In DFA, the output errors are projected onto the hidden layers through fixed random weights, effectively replacing the symmetric backward weights required in traditional backpropagation. This approach first emerged as a modification to the feedback alignment approach (which replaced the symmetric weights of the backpropagation algorithm with random ones). DFA has been extended and analyzed in several studies (Bartunov et al., 2018; Han & Yoo, 2019; Launay et al., 2019; 2020; Bordelon & Pehlevan, 2022), demonstrating its potential in training neural networks with less biologically implausible mechanisms. Clark et al. (2021) introduced another broadcast approach for a network with vector units and nonnegative weights for which three factor learning based update rule is applied.

Our proposed framework for error broadcasting differentiates itself through

- a **principled method** based on the orthogonality property of nonlinear MMSE estimators,
- error projection weights determined by the **cross-correlation** between the output errors and the layer activations as opposed to random weights of DFA,
- dynamic Hebbian updating of projection weights as opposed to fixed weights of DFA,
- updates involving **arbitrary nonlinear functions** of layer activities, encompassing a family of three-factor learning rules,
- the option to project layer activities forward to the output layer.

In summary, our approach provides a theoretical grounding for the error broadcasting mechanism and suggests ways to its effectiveness in training networks.

## 2 ERROR BROADCAST AND DECORRELATION METHOD

### 2.1 PROBLEM STATEMENT

To illustrate our approach, we first assume a multi layer perceptron (MLP) network with $L$ layers, including the output layer. Later on, we will demonstrate generalizations to other architectures. We label the input and layer activations of the network with $\mathbf{h}^{(k)} \in \mathbb{R}^{N^{(k)}}$, for $k = 0, \ldots, L$. Here, $k$ is the layer index, $N^{(k)}$ is the size of the layer $k$, $\mathbf{h}^{(0)} = \mathbf{x}$ is the input of the network, and $\mathbf{h}^{(L)}$ represents the output of the network.

The layer activations can be written as

$$\mathbf{h}^{(k)} = f^{(k)}(\mathbf{u}^{(k)}), \qquad \mathbf{u}^{(k)} = \mathbf{W}^{(k)}\mathbf{h}^{(k-1)} + \mathbf{b}^{(k)}, \tag{1}$$

for $k = 1, \ldots, L$, where $f^{(k)}$ are activation functions, $\mathbf{W}^{(k)}$ are synaptic weights and $\mathbf{b}^{(k)}$ are biases.

We assume that the performance criterion is the mean square of the output error ($\boldsymbol{\epsilon} = \mathbf{h}^{(L)} - \mathbf{y}$) between the final layer activations $\mathbf{h}^{(L)}$ and the desired output $\mathbf{y}$:

$$\mathbb{E}(\|\boldsymbol{\epsilon}\|^2) = \mathbb{E}(\|\mathbf{h}^{(L)} - \mathbf{y}\|_2^2). \tag{2}$$

## 2.2 Error broadcast and decorrelation loss functions

At the core of our approach lies the well-known orthogonality property of minimum mean square error (MMSE) estimators (Papoulis & Pillai, 2002) (see Appendix A for a brief summary):

*Let $\hat{\mathbf{y}}_*$ be the optimal nonlinear MMSE estimator of the desired vector $\mathbf{y}$ given the input $\mathbf{x}$, and let $\boldsymbol{\epsilon}_* = \mathbf{y} - \hat{\mathbf{y}}_*$ denote the corresponding estimation error. Then, for any properly measurable function $\mathbf{g}$ of the input $\mathbf{x}$, we have*

$$\mathbb{E}(\mathbf{g}(\mathbf{x})\boldsymbol{\epsilon}_*^T) = \mathbf{0}. \tag{3}$$

In other words, the estimation error of the optimal nonlinear MMSE estimator is orthogonal to any arbitrary nonlinear transformation of the input. In linear MMSE estimation, the orthogonality principle states that the estimation error is orthogonal to the observations and their linear functions. Mathematically, this is expressed equivalent to restricting $g(\cdot)$ to be a linear function. Using this orthogonality condition in reverse to derive linear estimators is a standard practice in the field (see, for example, Kailath et al. (2000)). Techniques such as Kalman Filtering are based on this principle, which is firmly grounded in the Hilbert space projection theorem.

For nonlinear MMSE estimation, the orthogonality condition in (3) is even stronger: the estimation error is orthogonal to any nonlinear function of the input. Exploiting this stronger condition to construct nonlinear MMSE estimators is an open problem, primarily because it raises questions about which nonlinear functions to choose and how many are needed.

In the proposed framework, we model the neural network as a parameterized nonlinear MMSE estimator and seek as many equations from the orthogonality principle as possible to determine these parameters. This is exactly the same principle as how the orthogonality condition is used in reverse to find parameters for linear estimators. To address the challenge of selecting nonlinear functions that yield informative equations for determining network parameters, we choose the activations of the hidden layers in the neural network as these functions. This choice is natural because these activations are directly related to the network's parameters through differentiation. Therefore, if the network converges to the optimal nonlinear MMSE estimator, the hidden layer activations should be orthogonal to the output errors. We formalize this observation with the following equations:

$$\mathbf{R}_{\mathbf{g}^{(k)}(\mathbf{h}^{(k)})\boldsymbol{\epsilon}} = \mathbb{E}(\mathbf{g}^{(k)}(\mathbf{h}^{(k)})\boldsymbol{\epsilon}_*^T) = \mathbf{0}, \text{ for } k = 0, \ldots, L, \tag{4}$$

where $\mathbf{g}^{(k)}$ is an arbitrary function of layer activations.

Figure 1c illustrates this phenomenon by showing the evolution of the average absolute correlation between layer activations and the error signal during backpropagation training of an MLP with three hidden layers on the CIFAR-10 dataset, based on the MSE criterion. The observed decrease in correlation between layer activations and output errors during MSE training is consistent with the orthogonality property stated in Equations (3) and (4).

Building upon this orthogonality property, we propose to define layer-specific surrogate loss functions. As shown in Section 2.3, these losses can be used to derive an alternative to backpropagation, where the output errors are broadcast directly to the network nodes, as depicted in Figure 1b. Specifically, based on the orthogonality condition in Equation (4), we propose minimizing the Frobenius norm of the cross-correlation matrices $\mathbf{R}_{\mathbf{g}^{(k)}(\mathbf{h}^{(k)}),\boldsymbol{\epsilon}}$ as a replacement for the standard MSE loss. To this end, we define the estimated cross-correlation matrix between a function $g^{(k)}$ of layer activations and the output error for batch $m$ and layer $k$ as

$$\hat{\mathbf{R}}_{\mathbf{g}^{(k)}(\mathbf{h}^{(k)})\boldsymbol{\epsilon}}[m] = \lambda\hat{\mathbf{R}}_{\mathbf{g}^{(k)}(\mathbf{h}^{(k)})\boldsymbol{\epsilon}}[m-1] + \frac{1-\lambda}{B}\mathbf{G}^{(k)}[m]\mathbf{E}[m]^T,$$

where $\lambda \in [0, 1]$ is the forgetting factor used in the autoregressive estimation, $B$ is the batch size, $\hat{\mathbf{R}}_{\mathbf{g}^{(k)}(\mathbf{h}^{(k)})\boldsymbol{\epsilon}}[0]$ is the initial value for the correlation matrix, which is a hyperparameter, and

$$\mathbf{G}^{(k)}[m] = \begin{bmatrix} \mathbf{g}^{(k)}(\mathbf{h}^{(k)}[mB+1]) & \mathbf{g}^{(k)}(\mathbf{h}^{(k)}[mB+2]) & \cdots & \mathbf{g}^{(k)}(\mathbf{h}^{(k)}[(m+1)B]) \end{bmatrix}, \tag{5}$$

is the matrix of nonlinearly transformed activations of layer $k$ for batch $m$, while

$$\mathbf{E}[m] = [\ \boldsymbol{\epsilon}[mB + 1] \quad \boldsymbol{\epsilon}[mB + 2] \quad \ldots \quad \boldsymbol{\epsilon}[(m + 1)B]\ ],$$

(6)

is the error matrix for batch $m$.

We then define the layer-specific loss function based on the orthogonality condition for layer $k$ as

$$J^{(k)}(\mathbf{h}^{(k)}, \boldsymbol{\epsilon})[m] = \frac{1}{2} \left\| \hat{\mathbf{R}}_{\mathbf{g}^{(k)}(\mathbf{h}^{(k)})\boldsymbol{\epsilon}}[m] \right\|_F^2,$$

(7)

where $\| \cdot \|_F$ denotes the Frobenius norm. This loss function captures the sum of the squared magnitudes of all cross-correlations between the components of the output error and the activations of layer $k$. Therefore, we refer to the minimization of this loss function as *decorrelation*.

## 2.3 ERROR BROADCAST AND DECORRELATION ALGORITHM

The set of functions in (7) defines individual loss functions for each hidden layer of the network, which are used to adjust the layer parameters. These loss functions can be minimized using a gradient descent-based algorithm.

To minimize the loss for layer $k$, we compute the gradient of the loss function $J^{(k)}(\mathbf{h}^{(k)}, \boldsymbol{\epsilon})$ with respect to the weight $W_{ij}^{(k)}$. The derivative can be decomposed into two terms:

$$\frac{\partial J^{(k)}(\mathbf{h}^{(k)}, \boldsymbol{\epsilon})}{\partial W_{ij}^{(k)}}[m] = \underbrace{\frac{1 - \lambda}{B} Tr \left( \hat{\mathbf{R}}_{\mathbf{g}^{(k)}(\mathbf{h}^{(k)})\boldsymbol{\epsilon}}[m] \mathbf{E}[m] \frac{\partial \mathbf{G}^{(k)}[m]^T}{\partial W_{ij}^{(k)}} \right)}_{[\Delta \mathbf{W}_1^{(k)}[m]]_{ij}}$$

$$+ \underbrace{\frac{1 - \lambda}{B} Tr \left( \hat{\mathbf{R}}_{\mathbf{g}^{(k)}(\mathbf{h}^{(k)})\boldsymbol{\epsilon}}[m] \frac{\partial \mathbf{E}[m]}{\partial W_{ij}^{(k)}} \mathbf{G}^{(k)}[m]^T \right)}_{[\Delta \mathbf{W}_2^{(k)}[m]]_{ij}}.$$

Similarly, the derivative with respect to the bias $b_i^{(k)}$ is given by:

$$\frac{\partial J^{(k)}(\mathbf{h}^{(k)}, \boldsymbol{\epsilon})}{\partial b_i^{(k)}}[m] = \underbrace{\frac{1 - \lambda}{B} Tr \left( \hat{\mathbf{R}}_{\mathbf{g}(\mathbf{h}^{(k)})\boldsymbol{\epsilon}}[m] \mathbf{E}[m] \frac{\partial \mathbf{G}^{(k)}[m]^T}{\partial b_i^{(k)}} \right)}_{[\Delta \mathbf{b}_1^{(k)}[m]]_i}$$

$$+ \underbrace{\frac{1 - \lambda}{B} Tr \left( \hat{\mathbf{R}}_{\mathbf{g}^{(k)}(\mathbf{h}^{(k)})\boldsymbol{\epsilon}}[m] \frac{\partial \mathbf{E}[m]}{\partial b_i^{(k)}} \mathbf{G}^{(k)}[m]^T \right)}_{[\Delta \mathbf{b}_2^{(k)}[m]]_i}.$$

Here $\Delta \mathbf{W}_1^{(k)}, \Delta \mathbf{b}_1^{(k)}[m]$ ($\Delta \mathbf{W}_2^{(k)}, \Delta \mathbf{b}_2^{(k)}[m]$) represent the components of the gradients containing derivatives of activations (output errors) with respect to the layer parameters. As derived in Appendix B.1, we obtain the closed-form expressions for $\Delta \mathbf{W}_1^{(k)}[m]$ and $\Delta \mathbf{b}_1^{(k)}[m]$:

$$[\Delta \boldsymbol{W}_1^{(k)}[m]]_{ij} = \frac{1 - \lambda}{B} \sum_{n=mB+1}^{(m+1)B} g_i'^{(k)}(h_i^{(k)}[n]) f'^{(k)}(u_i^{(k)}[n]) q_i^{(k)}[n] h_j^{(k-1)}[n],$$

(8)

$$[\Delta \mathbf{b}_1^{(k)}[m]]_i = \frac{1 - \lambda}{B} \sum_{n=mB+1}^{(m+1)B} g_i'^{(k)}(h_i^{(k)}[n]) f'^{(k)}(u_i^{(k)}[n]) q_i^{(k)}[n],$$

(9)

where $g_i'^{(k)}$ and $f'^{(k)}$ denote the derivatives of the nonlinearity $g^{(k)}$ and the activation function $f^{(k)}$, respectively. The term $\mathbf{q}^{(k)}[m]$ is defined as:

$$\mathbf{q}^{(k)}[m] = \hat{\mathbf{R}}_{\mathbf{g}^{(k)}(\mathbf{h}^{(k)}), \boldsymbol{\epsilon}}[m] \boldsymbol{\epsilon}[m],$$

representing the projection of the output error onto the layer activations, with the cross-correlation matrix $\hat{\mathbf{R}}_{\mathbf{g}^{(k)}(\mathbf{h}^{(k)}),\,\boldsymbol{\epsilon}}[m]$ as the transformation matrix. These projections are shown in Figure 1b.

The update terms $\Delta \boldsymbol{W}_1^{(k)}[m]$ and $\Delta \mathbf{b}_1^{(k)}[m]$ aim to adjust the activations so they gradually become orthogonal to $\boldsymbol{\epsilon}$ as they are based on the derivatives of the layer activations with respect to the layer parameters. Simultaneously, $\Delta \boldsymbol{W}_2^{(k)}[m]$ and $\Delta \mathbf{b}_2^{(k)}[m]$, derived from the derivatives of the output error with respect to the layer parameters, work to adjust the output errors, pushing them into a configuration more orthogonal to the activations. While both types of updates strive to decorrelate activations and output errors, there is a critical distinction: $\Delta \boldsymbol{W}_1^{(k)}[m]$ and $\Delta \mathbf{b}_1^{(k)}[m]$ depend only on the layer activations and the broadcast output error signals, whereas $\Delta \boldsymbol{W}_2^{(k)}[m]$ and $\Delta \mathbf{b}_2^{(k)}[m]$ rely on signals that propagate backward from the output layer to the current layer, resembling back-propagation (as shown in Appendix B.1).

By focusing solely on $\Delta \boldsymbol{W}_1^{(k)}[m]$ and $\Delta \mathbf{b}_1^{(k)}[m]$, we can eliminate the need for propagation terms, resulting in a completely localized update mechanism for training the neural network. Therefore, we prescribe the Error Broadcast and Decorrelation (EBD) update mechanism as:

$$\mathbf{W}^{(k)}[m+1] = \mathbf{W}^{(k)}[m] - \mu^{(k)}[m]\Delta \mathbf{W}_1^{(k)}[m],$$
$$\mathbf{b}^{(k)}[m+1] = \mathbf{b}^{(k)}[m] - \mu^{(k)}[m]\Delta \mathbf{b}_1^{(k)}[m],$$

for $k = 1, \ldots, L-1$, where $\mu^{(k)}[m]$ is the learning rate for layer $k$ at batch $m$. For the final layer $(k = L)$, we utilize the standard MMSE gradient update:

$$\mathbf{W}^{(L)}[m+1] = \mathbf{W}^{(L)}[m] - \mu^{(L)}[m]\frac{1}{B}\sum_{n=mB+1}^{(m+1)B} \left( f'^{(k)}(\mathbf{u}^{(L)}[n]) \odot \boldsymbol{\epsilon}[n] \right) \mathbf{h}^{(L-1)}[n]^T,$$

$$\mathbf{b}^{(L)}[m+1] = \mathbf{b}^{(L)}[m] - \mu^{(L)}[m]\frac{1}{B}\sum_{n=mB+1}^{(m+1)B} f'^{(k)}(\mathbf{u}^{(L)}[n]) \odot \boldsymbol{\epsilon}[n],$$

where $f'^{(L)}$ is the derivative of the activation function of the output layer.

## 2.4 FURTHER EBD ALGORITHM EXTENSIONS

We propose further extensions to the EBD framework to address potential activation collapse, which can arise when minimizing correlations is the sole objective. To prevent unit-level collapse, we introduce power regularization, while entropy regularization is employed to prevent dimensional collapse. Both regularizations can be implemented in ANNs as well as biologically plausible networks. Although CorInfoMax-EBD inherently includes entropy regularization, it can also benefit from the addition of power regularization for enhanced stability. Additionally, we introduce forward layer activation projections to improve the algorithm's versatility. We also extend the EBD formulations to more complex architectures, including Convolutional Neural Networks (CNNs) and Locally Connected (LC) networks. Detailed implementations and evaluations of these extensions are provided in Appendix C.

### 2.4.1 AVOIDING COLLAPSE

A critical challenge when applying the EBD algorithm to MLPs is the potential for activation collapse, where layer decorrelation losses defined in (7) are minimized by driving all layer activations to zero, even in the presence of non-zero output errors. This unintended minimization undermines the network's ability to learn meaningful representations, as all activations become inactive.

To counteract activation collapse, we introduce two complementary algorithmic remedies:

Power normalization A straightforward safeguard against total activation collapse is to regulate the power of layer activations through a power normalization loss function:

$$J_P^{(k)}(\mathbf{h}^{(k)}) = \sum_{l=1}^{N^{(k)}} \left( \frac{1}{B}\sum_{n=mB+1}^{(m+1)B} h_l^{(k)}[n]^2 - P^{(k)} \right)^2, \tag{10}$$

where $P^{(k)}$ is a hyperparameter representing the desired power level for activations in layer $k$. This loss ensures activations maintain a consistent power level, preventing collapse.

**Layer entropy** While power normalization prevents total collapse, it does not address the issue of activations collapsing into low-dimensional subspaces, which can restrict the network's expressiveness. To mitigate this dimensional degeneracy, we propose the incorporation of the layer-entropy objective, which has been utilized in self-supervised learning (Ozsoy et al., 2022) and principled biologically more realistic neural network formulations (Bozkurt et al., 2023):

$$J_E^{(k)}(\mathbf{h}^{(k)})[m] = \frac{1}{2} \log \det(\mathbf{R}_{\mathbf{h}^{(k)}}[m] + \varepsilon^{(k)}\mathbf{I}). \tag{11}$$

In this expression, $\mathbf{R}_{\mathbf{h}^{(k)}}[m]$ represents the layer auto-correlation matrix for layer $k$ at batch $m$, which is obtained through an auto-regressive update (as proposed in Ozsoy et al. (2022))

$$\mathbf{R}_{\mathbf{h}^{(k)}}[m] = \lambda_E \mathbf{R}_{\mathbf{h}^{(k)}}[m-1] + (1-\lambda_E)\frac{1}{B}\overline{\mathbf{H}^{(k)}[m]\mathbf{H}^{(k)}[m]^T}, \quad \text{where,}$$

$$\mathbf{H}^{(k)}[m] = \begin{bmatrix} \mathbf{h}^{(k)}[mB+1] & \mathbf{h}^{(k)}[mB+2] & \dots & \mathbf{h}^{(k)}[(m+1)B] \end{bmatrix}, \tag{12}$$

is the activation matrix, and $\lambda_E$ is the forgetting factor for the autoregressive averaging.

We note that, while the use of entropy and power regularizers may not be entirely novel, they play a significant role in preventing the collapse problem.

### 2.4.2 Forward broadcast

In the EBD algorithm (Section 2.3), output errors are broadcast to layers to adjust weights and reduce correlations with activations. To complement this, we introduce forward broadcasting, projecting hidden layer activations onto the output layer to optimize the decorrelation loss by adjusting the final layer's parameters. Details are provided in Appendix B.3.

### 2.4.3 Extensions to other network architectures

The EBD approach relies on the orthogonality of output errors to node activations , independent of network topology. We extend EBD to convolutional neural networks (CNNs) in Appendix C.1 and to locally connected (LC) networks in Appendix C.2.

## 3 EBD for biologically more realistic networks

In the previous section, we introduced the EBD algorithm within the context of MLP networks. While MLPs can resemble biologically plausible networks depending on the credit assignment mechanism, in this section, we extend the application of the EBD approach to neural networks that exhibit more biologically realistic dynamics and architectures. This extension is motivated by two key properties of the EBD framework: first, the error is broadcast directly to the layers, naturally eliminating the weight symmetry issue observed in the BP algorithm; second, the EBD update rules mimic the form of extended Hebbian updates with modulatory components. In the following subsections, we explore how EBD relates to the biologically plausible three-factor learning rule and demonstrate its integration with the biologically more realistic CorInfoMax networks (Bozkurt et al., 2023).

### 3.1 Three factor learning rule and EBD

The three-factor learning rule proposed for biological neural networks extends the traditional two-factor Hebbian rule by incorporating a modulatory signal into synaptic updates based on presynaptic and postsynaptic activity (Frémaux & Gerstner, 2016; Gerstner et al., 2018). Upon closer examination of the EBD update expression in (8), we observe that it conforms to the three-factor update form:

$$\Delta W_{ij}^{(k)} \propto \underbrace{g_i'^{(k)}(h_i^{(k)})f'^{(k)}(u_i^{(k)})}_{\text{Postsynaptic}} \underbrace{q_i^{(k)}}_{\text{Modulatory}} \underbrace{h_j^{(k-1)}}_{\text{Presynaptic}},$$

where the modulatory component $q_i^{(k)}$ is the linearly projected version of the error. This observation indicates that the EBD formulation inherently supports a variety of three-factor update rules,

depending on the choice of the nonlinearity $g^{(k)}$. For instance, selecting $g_i^{(k)}(h_i^{(k)}) = h_i^{(k)^2}$ leads to the error-modulated Hebbian update (Loewenstein & Seung, 2006; Bordelon & Pehlevan, 2022):

$$\Delta W_{ij}^{(k)} \propto \underbrace{h_i^{(k)} f'^{(k)}(u_i^{(k)})}_{\text{Postsynaptic}} \underbrace{q_i^{(k)}}_{\text{Modulatory}} \underbrace{h_j^{(k-1)}}_{\text{Presynaptic}}.$$

By supporting a variety of three-factor update rules through different nonlinear functions, EBD offers potential for modeling neural learning processes consistent with biological observations

### 3.2 CORINFOMAX-EBD: CORINFOMAX WITH THREE FACTOR LEARNING RULE

One of the significant advantages of the EBD framework is its flexibility to broadcast output errors into network nodes, which can be leveraged to transform time-contrastive, biologically plausible approaches into non-contrastive forms. To illustrate this property, we propose a modification of the recently introduced CorInfoMax framework (Bozkurt et al., 2023) (see Appendix D for a summary). The CorInfoMax approach uses correlative information flow between layers as its objective function:

$$\mathcal{J}_{CI}[m] = \sum_{k=1}^{L-1} \left( \hat{I}^{(\varepsilon_k)}(\mathbf{h}^{(k-1)}, \mathbf{h}^{(k)})[m] + \hat{I}^{(\varepsilon_k)}(\mathbf{h}^{(k)}, \mathbf{h}^{(k+1)})[m] \right), \quad \text{where,}$$

$$\hat{I}^{(\varepsilon_k)}(\mathbf{h}^{(k)}, \mathbf{h}^{(k+1)})[m] = \frac{1}{2} \log \det(\hat{\mathbf{R}}_{\mathbf{h}^{(k+1)}}[m] + \varepsilon_k \boldsymbol{I}) - \frac{1}{2} \log \det(\hat{\boldsymbol{R}}_{\overrightarrow{\mathbf{e}}^{(k+1)}}[m] + \varepsilon_k \boldsymbol{I}),$$

$$\hat{I}^{(\varepsilon_k)}(\mathbf{h}^{(k)}, \mathbf{h}^{(k+1)})[m] = \frac{1}{2} \log \det(\hat{\mathbf{R}}_{\mathbf{h}^{(k)}}[m] + \varepsilon_k \boldsymbol{I}) - \frac{1}{2} \log \det(\hat{\boldsymbol{R}}_{\overleftarrow{\mathbf{e}}^{(k)}}[m] + \varepsilon_k \boldsymbol{I}),$$

are alternative forms of correlative mutual information between nodes, defined in terms of the correlation matrices of layer activations, i.e., $\hat{\mathbf{R}}_{\mathbf{h}^{(k)}}$ and the correlation matrices of forward and backward prediction errors ($\hat{\boldsymbol{R}}_{\overrightarrow{\mathbf{e}}^{(k+1)}}$ and $\hat{\boldsymbol{R}}_{\overleftarrow{\mathbf{e}}^{(k)}}$). Here, forward/backward prediction errors are defined by

$$\overrightarrow{\mathbf{e}}^{(k+1)}[n] = \mathbf{h}^{(k+1)}[n] - \mathbf{W}^{(f,k)}[m]\mathbf{h}^{(k)}[n], \qquad \overleftarrow{\mathbf{e}}^{(k)}[n] = \mathbf{h}^{(k)}[n] - \mathbf{W}^{(b,k)}[m]\mathbf{h}^{(k+1)}[n],$$

respectively. Here, $\mathbf{W}^{(f,k)}[m]$ ($\mathbf{W}^{(b,k)}[m]$) is the forward (backward) prediction matrix for layer $k$.

This objective leads to network dynamics corresponding to a structure with feedforward and feedback prediction weights, and lateral connections $\mathbf{B}^{(k)}$ that maximize layer entropy. In the original work (Bozkurt et al., 2023), the two-phase EP approach (Scellier & Bengio, 2017) is proposed to train the network weights. As an alternative, we propose employing the EBD update rule to replace the two-phase EP adaptation. The proposed CorInfoMax-EBD algorithm is described by the following update equations defined in Algorithm 1:

---

**Algorithm 1** CorInfoMax-EBD Algorithm for Updating Weights in Layer $k$

**Input:** Batch size $B$, layer index $k$, iteration step $m$, learning rates $\mu^{(f,k)}$, $\mu^{(b,k)}$, $\mu^{(d_f,k)}$, $\mu^{(d_b,k)}$, $\mu^{(d_l,k)}$, factors $\lambda_d$, $\lambda_E$, $\gamma_E$, activations $\mathbf{H}^{(k)}$ in (12), the nonlinear function of layer activations $\mathbf{G}^{(k)}$ in (5), the derivative of the nonlinear function of layer activations $\mathbf{G}_d^{(k)}$ in (13), the derivative of activations $\mathbf{F}_d^{(k)}$ in (14), output error $\mathbf{E}$ in (6), prediction errors $\overleftarrow{\mathbf{E}}$ and $\overrightarrow{\mathbf{E}}^{(k)}$ in (33-34), lateral weight outputs $\mathbf{Z}^{(k)}$ in (35).

**Output:** Updated weights $\mathbf{W}^{(f,k)}$, $\mathbf{W}^{(b,k)}$, $\mathbf{B}^{(k)}$.

**Step 1: Update error projection weights:** $\mathbf{R}_{\mathbf{g}^{(k)}(\mathbf{h}^{(k)})_{\boldsymbol{\epsilon}}}[m] = \lambda_d \mathbf{R}_{\mathbf{g}^{(k)}(\mathbf{h}^{(k)})_{\boldsymbol{\epsilon}}}[m-1] + \frac{1-\lambda_d}{B}\mathbf{G}^{(k)}[m]\mathbf{E}[m]^T$

**Step 2: Project errors to layer $k$:** $\mathbf{Q}^{(k)}[m] = \mathbf{R}_{\mathbf{g}^{(k)}(\mathbf{h}^{(k)})_{\boldsymbol{\epsilon}}}^{(k)}[m]\mathbf{E}[m]$

**Step 3: Find the gradient of the nonlinear function of activations for layer $k$:**

$$\boldsymbol{\Phi}^{(k)}[m] = \mathbf{F}_d^{(k)}[m] \odot \mathbf{Q}^{(k)}[m] \odot \mathbf{G}_d^{(k)}[m]$$

**Step 4: Update forward, backward and lateral weights for layer $k$:**

$$\mathbf{W}^{(f,k)}[m] = \mathbf{W}^{(f,k)}[m-1] + \left( B^{-1}\mu^{(f,k)}[m]\overrightarrow{\mathbf{E}}^{(k)}[m] - B^{-1}\mu^{(d_f,k)}[m]\boldsymbol{\Phi}^{(k)}[m] \right) \mathbf{H}^{(k-1)}[m]^T$$

$$\mathbf{W}^{(b,k)}[m] = \mathbf{W}^{(b,k)}[m-1] + \left( B^{-1}\mu^{(b,k)}[m]\overleftarrow{\mathbf{E}}^{(k)}[m] - B^{-1}\mu^{(d_b,k)}[m]\boldsymbol{\Phi}^{(k)}[m] \right) \mathbf{H}^{(k+1)}[m]^T$$

$$\mathbf{B}^{(k)}[m] = \lambda_E^{-1}\mathbf{B}^{(k-1)}[m] - B^{-1}\gamma_E \mathbf{Z}^{(k)}[m]\mathbf{Z}^{(k)}[m]^T - B^{-1}\mu^{(d_l,k)}[m]\boldsymbol{\Phi}^{(k)}[m]\mathbf{H}^{(k)}[m]^T$$

---

Here, we assume layer activations $\mathbf{H}^{(k)}$, output error $\mathbf{E}^{(k)}$, forward (backward) prediction errors $\overrightarrow{\mathbf{E}}^{(k)}$ ($\overleftarrow{\mathbf{E}}^{(k)}$) and lateral weight outputs $\mathbf{Z}^{(k)}$ are computed by the CorInfoMax network dynamics specified in Bozkurt et al. (2023) (see also Appendix D). By integrating EBD, we enable a single-phase update per input, eliminating the less biologically plausible two-phase learning mechanism required by CorInfoMax-EP. The two-phase approach of EP—comprising separate label-free and label-connected phases—is considered less plausible because biological neurons are considered unlikely to alternate between distinct global phases for learning. Our method not only simplifies the learning process but also aligns more closely with biological learning processes. Additionally, we achieve comparable or even superior performance compared to the CorInfoMax-EP (see Section 4).

We also note that the CorInfoMax-EBD scheme proposed in this section is more biologically realistic than the MLP-based EBD approach in Section 2 due to several factors:

- The MLP-based EBD approach employs an entropy regularizer in (11), whose gradient involves the inverse of the layer-correlation matrix $\mathbf{B}^{(k)} = \mathbf{R}_{\mathbf{h}}^{(k)^{-1}}$, which in its direct form appears non-biologically plausible. The same entropy term is an integral part of the CorInfoMax objective. As described in Appendix D, in the online optimization of the CorInfoMax objective, the entropy gradient can be implemented via lateral connections in the CorInfoMax network. Specifically, the learning gradients for this entropy function can be implemented as rank-1 (anti-Hebbian) updates on the $\mathbf{B}$ matrix when a batch size of $B = 1$ is used. Note that the same lateral weights $\mathbf{B}^{(k)}$ are also updated by a three-factor rule due to the EBD update, as described in Algorithm 1.

- Similarly, for CorInfoMax with $B = 1$, the power normalization regularizer in (10) reduces to the form $(h_l^{(k)}[n]^2 - P^{(k)})^2$ for each neuron. The gradient of this expression corresponds to local updates, enhancing biological plausibility. Even when a single sample is used for power calculation, the power regularizer remains effective due to the averaging effect across samples over time.

- In addition to the biologically plausible implementations of power and entropy regularizations in the online CorInfoMax setting, the neuron models used in CorInfoMax networks involve more realistic neuron models with apical and basal dendrite alongside the soma compartment.

- Another aspect contributing to the biological plausibility is the existence of feedback connections (corresponding to the backward predictors) in the CorInfoMax network structure.

## 4 NUMERICAL EXPERIMENTS AND DISCUSSION OF RESULTS

In this section, we evaluate the performance of the proposed Error Broadcast and Decorrelation (EBD) approach on two benchmark datasets: MNIST (Deng, 2012) and CIFAR-10 (Krizhevsky et al., 2009). For experiments involving MLP, CNN and LC, we consider the same network architectures used in Clark et al. (2021). We also tested the proposed CorInfoMax-EBD model in comparison to the CorInfoMax-EP model of Bozkurt et al. (2023). More details about architectures, implementations, hyperparameter selections, and experimental outputs are provided in the Appendix E.

The test accuracy results of our EBD algorithm compared to BP and three error-broadcast methods: DFA without and with entropy regularization (DFA-E) (Nokland, 2016), nonnegative global error vector broadcasting (NN-GEVB) (Clark et al., 2021), and mixed-sign global error vector broadcasting (MS-GEVB) (Clark et al., 2021) —are summarized in Table 1 for MNIST and Table 2 for CIFAR-10. In addition, the test accuracy results for biologically more realistic CorInfoMax networks trained with EP and EBD learning methods are shown in Table 3. These results confirm that

Table 1: Accuracy (%) results for MLP, CNN, and LC networks on the MNIST dataset; best and second-best are bold and underlined. Columns marked with [*] are from Clark et al. (2021).

|      | DFA   | DFA+E (ours) | NN-GEVB [*] | MS-GEVB [*] | BP        | EBD (ours) |
|------|-------|--------------|-------------|-------------|-----------|------------|
| MLP  | 98.09 | 98.21        | 98.13       | 97.68       | **98.72** | 98.24 |
| CNN  | 99.06 | 99.07        | 97.67       | 98.17       | **99.46** | 99.08 |
| LC   | 98.92 | 98.90 | 98.22       | 98.16       | **99.13** | 98.92      |

Table 2: Accuracy (%) results for MLP, CNN, and LC networks on the CIFAR-10 dataset; best and second-best are bold and underlined. Columns marked with [*] are from Clark et al. (2021).

|       | DFA   | DFA+E (ours) | NN-GEVB [*] | MS-GEVB [*] | BP        | EBD (ours) |
|-------|-------|--------------|-------------|-------------|-----------|------------|
| MLP   | 52.09 | 52.22        | 52.38       | 51.14       | **56.37** | 55.47 |
| CNN   | 58.39 | 58.56        | 66.26       | 61.57       | **75.24** | 66.42 |
| LC    | 62.19 | 62.12        | 58.92       | 59.89       | **67.81** | 64.23 |

Table 3: Accuracy (%) results for EP and EBD CorInfoMax algorithms; best and second-best are bold and underlined. Column marked with [*] is from Bozkurt et al. (2023).

|          | CorInfoMax-EP [*] (batch size : 20) | CorInfoMax-EBD (Ours) (batch size : 20) | CorInfoMax-EBD (Ours) (batch size : 1) |
|----------|-------------------------------------|-----------------------------------------|----------------------------------------|
| MNIST    | **97.58**                           | 97.53                            | 94.7                                   |
| CIFAR-10 | 50.97                               | **55.79**                               | 53.4                            |

the networks trained with EBD approach achieves equivalent performance on the MNIST dataset and significantly better performance on the CIFAR-10 dataset. The improvements of EBD in Table 2 over DFA can be attributed to the fact that error projection weights are adaptable in EBD, and the improvement of CorInfoMax-EBD over CorInfoMax-EP in Table 3 can be attributed to the fact that CorInfoMax-EBD incorporates error decorrelation in updating lateral weights, whereas CorInfoMax-EP relies only on anti-Hebbian updates.

## 5 Conclusions, extensions and limitations

**Conclusions and Extensions.** In this article, we introduced the Error Broadcast and Decorrelation (EBD) framework as a biologically plausible alternative to traditional backpropagation. EBD addresses the credit assignment problem by minimizing correlations between layer activations and output errors, offering fresh insights into biologically realistic learning. This approach provides a theoretical foundation for existing error broadcast mechanisms in biological neural networks and facilitates flexible implementations in neuromorphic and artificial neural systems. EBD's error-broadcasting mechanism aligns with biological processes where global error signals modulate local synaptic updates, potentially bridging the gap between artificial learning algorithms and natural neural computations. Moreover, EBD's simplicity and parallelism make it suitable for efficient hardware implementations, such as neuromorphic computing systems that emulate the brain's architecture.

We believe that the MMSE orthogonality property underpinning the proposed EBD framework has great potential for developing new algorithms, deepening theoretical understanding, and analyzing neural networks in both artificial and biological contexts. We are currently unaware of similar theoretical properties for alternative loss functions. Notably, our numerical experiments in Appendix F.2 reveal that similar decorrelation behavior occurs for networks trained with backpropagation and categorical cross entropy loss, suggesting that decorrelation may be a general feature of the learning process and an intriguing avenue for further investigation.

**Limitations.** The current implementation of EBD involves several hyperparameters, including multiple learning rates for decorrelation and regularization functions, as well as forgetting factors for correlation matrices. Although these parameters offer flexibility, they add complexity to the tuning process. Additionally, the use of dynamically updated error projection matrices and the potential integration of entropy regularization may increase memory and computational demands. Future work could explore more efficient methods for managing these components, potentially automating or simplifying the tuning process to enhance usability. Furthermore, while the scalability of EBD is left out of the focus of the article, we acknowledge its importance. Launay et al. (2020) demonstrated that DFA scales to high-dimensional tasks like transformer-based language modeling. Since DFA is equivalent to EBD with frozen projection weights and without entropy regularization, we anticipate that EBD could scale similarly. However, this remains unvalidated empirically. Examining EBD's scalability and streamlining its components to improve usability are important tasks for future work.

# 6 REPRODUCIBILITY

To facilitate the reproducibility of our results, we have included the following:

    i. Detailed information on the derivation of the weight and bias updates of the Error Broadcast and Decorrelation (EBD) Algorithm for various networks in Appendix B for MLPs, C.1 for CNNs, C.2 for LCs,

    ii. Full list of hyperparameters used in the experiments in Appendix E.3.5, E.4.4, E.5.4, E.6.4,

    iii. Algorithm descriptions for CorInfoMax Error Broadcast and Decorrelation (CorInfoMax-EBD) Algorithm in pseudo-code format in Appendix E.3.2,

    iv. Python scripts, Jupyter notebooks, and bash scripts for replicating the individual experiments and reported results are included in the supplementary zip file.

# 7 ETHICS STATEMENT

We do not identify any immediate ethical concerns regarding the algorithmic framework proposed in this article. Furthermore, to the best of our knowledge, the datasets used in this work do not have any known or reported ethical issues.

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

APPENDIX

# A    PRELIMINARIES ON NONLINEAR MINIMUM MEAN SQUARE ERROR ESTIMATION

Let $\mathbf{y} \in \mathbb{R}^p$ and $\mathbf{x} \in \mathbb{R}^n$ represent two non-degenerate random vectors with a joint probability density function $f_{\mathbf{yx}}(\mathbf{y}, \mathbf{x})$ and conditional density $f_{\mathbf{y}|\mathbf{x}}(\mathbf{y}|\mathbf{x})$. The goal of nonlinear minimum mean square error (MMSE) estimation is to find an estimator function $\mathbf{b} : \mathbb{R}^n \to \mathbb{R}^p$ that minimizes the mean squared error (MSE), which is defined as:

$$MSE(\mathbf{b}) = \mathbb{E}(\|\mathbf{y} - \mathbf{b}(\mathbf{x})\|_2^2).$$

**Lemma A.1.** *The best linear MMSE estimate of $\mathbf{y}$ given $\mathbf{x}$ is:*

$$\mathbf{b}_{MMSE}(\mathbf{x}) = \mathbb{E}_{\mathbf{y}|\mathbf{x}}(\mathbf{y}|\mathbf{x}).$$

The proof of Lemma A.1 relies on the following fundamental result (see, for example, the textbook by Papoulis & Pillai (2002)), which is central to the development of the entire EBD framework in the current article:

**Lemma A.2.** *The estimation error for $\mathbf{b}_{MMSE}(\mathbf{x}) = \mathbb{E}_{\mathbf{y}|\mathbf{x}}[\mathbf{y}|\mathbf{x}]$, denoted as $\mathbf{e}_{MMSE} = \mathbf{y} - \mathbf{b}_{MMSE}(\mathbf{x})$, is orthogonal to any vector-valued function $\mathbf{g} : \mathbb{R}^n \to \mathbb{R}^k$ of $\mathbf{x}$. Formally, we have:*

$$\mathbb{E}(\mathbf{e}_{MMSE}\mathbf{g}(\mathbf{x})^T) = \mathbf{0}.$$

*Proof.* (Lemma A.2) The proof follows simple steps:

$$\begin{aligned}
\mathbb{E}(\mathbf{e}_{MMSE}\mathbf{g}(\mathbf{x})^T) &= \mathbb{E}_{\mathbf{x}}\left(\mathbb{E}_{\mathbf{y}|\mathbf{x}}\left((\mathbf{y} - \mathbb{E}_{\mathbf{y}|\mathbf{x}}(\mathbf{y}|\mathbf{x}))\mathbf{g}(\mathbf{x})^T|\mathbf{x}\right)\right) \\
&= \mathbb{E}_{\mathbf{x}}((\mathbb{E}_{\mathbf{y}|\mathbf{x}}(\mathbf{y}|\mathbf{x}) - \mathbb{E}_{\mathbf{y}|\mathbf{x}}(\mathbf{y}|\mathbf{x}))\mathbf{g}(\mathbf{x})^T) = \mathbf{0}.
\end{aligned}$$

$\square$

Using Lemma A.2, we can now prove Lemma Lemma A.1:

*Proof.* (Lemma A.1) Let $\mathbf{b} : \mathbb{R}^n \to \mathbb{R}^p$ be any arbitrary function. The corresponding MSE can be written as:

$$MSE(\mathbf{b}) = \mathbb{E}(\|\mathbf{y} - \mathbf{b}(\mathbf{x})\|_2^2).$$

By adding and subtracting $\mathbb{E}_{\mathbf{y}|\mathbf{x}}[\mathbf{y}|\mathbf{x}]$, we can decompose the error as:

$$\begin{aligned}
MSE(\mathbf{b}) &= \mathbb{E}(\|\mathbf{y} - \mathbb{E}_{\mathbf{y}|\mathbf{x}}(\mathbf{y}|\mathbf{x}) + \mathbb{E}_{\mathbf{y}|\mathbf{x}}(\mathbf{y}|\mathbf{x}) - \mathbf{b}(\mathbf{x})\|_2^2) \\
&= \mathbb{E}(\|\mathbf{y} - \mathbb{E}_{\mathbf{y}|\mathbf{x}}\|_2^2) + \mathbb{E}(\|\mathbb{E}_{\mathbf{y}|\mathbf{x}}(\mathbf{y}|\mathbf{x}) - \mathbf{b}(\mathbf{x})\|_2^2) \\
&\quad + 2\mathbb{E}((\mathbf{y} - \mathbb{E}_{\mathbf{y}|\mathbf{x}})^T(\mathbb{E}_{\mathbf{y}|\mathbf{x}}(\mathbf{y}|\mathbf{x}) - \mathbf{b}(\mathbf{x}))) \\
&= \mathbb{E}(\|\mathbf{y} - \mathbb{E}_{\mathbf{y}|\mathbf{x}}\|_2^2) + \mathbb{E}(\|\mathbb{E}_{\mathbf{y}|\mathbf{x}(\mathbf{y}|\mathbf{x})}(\mathbf{y}|\mathbf{x}) - \mathbf{b}(\mathbf{x})\|_2^2) \\
&\quad + 2\mathbb{E}(Tr((\mathbf{y} - \mathbb{E}_{\mathbf{y}|\mathbf{x}}(\mathbf{y}|\mathbf{x}))(\mathbb{E}_{\mathbf{y}|\mathbf{x}}(\mathbf{y}|\mathbf{x}) - \mathbf{b}(\mathbf{x}))^T)) \\
&= \mathbb{E}(\|\mathbf{y} - \mathbb{E}_{\mathbf{y}|\mathbf{x}}\|_2^2) + \mathbb{E}(\|\mathbb{E}_{\mathbf{y}|\mathbf{x}(\mathbf{y}|\mathbf{x})}(\mathbf{y}|\mathbf{x}) - \mathbf{b}(\mathbf{x})\|_2^2) \\
&\quad + 2Tr(\mathbb{E}(\mathbf{e}_{MMSE}(\mathbb{E}_{\mathbf{y}|\mathbf{x}}(\mathbf{y}|\mathbf{x}) - \mathbf{b}(\mathbf{x}))^T)).
\end{aligned}$$

The third term, representing the cross product, vanishes by Lemma A.2, leaving us with:

$$MSE(\mathbf{b}) = \mathbb{E}(\|\mathbf{y} - \mathbf{b}_{MMSE}(\mathbf{x})\|_2^2) + \mathbb{E}(\|\mathbf{b}_{MMSE}(\mathbf{x}) - \mathbf{b}(\mathbf{x})\|_2^2).$$

Since the second term is always non-negative, the MSE is minimized when $\mathbf{b}(\mathbf{x}) = \mathbf{b}_{MMSE}(\mathbf{x})$.

$\square$

# B   THE DERIVATION OF UPDATE TERMS

In this section, we present the detailed derivations for the EBD algorithm and its variations, as introduced in Section 2.3.

## B.1   $\Delta\mathbf{W}_1$ AND $\Delta\mathbf{b}_1$ CALCULATION

In Section 2.3, we defined the weight update elemet $[\Delta\mathbf{W}_1]_{ij}$ as follows:

$$\frac{1-\lambda}{B}Tr\left(\hat{\mathbf{R}}_{\mathbf{g}^{(k)}(\mathbf{h}^{(k)})\boldsymbol{\epsilon}}[m]\mathbf{E}[m]\frac{\partial\mathbf{G}^{(k)}[m]^T}{\partial W_{ij}^{(k)}}\right).$$

The derivative term in this expression can be expanded as

$$\frac{\partial\mathbf{G}^{(k)}[m]}{\partial W_{ij}^{(k)}}=\mathbf{e}_i\begin{bmatrix}g'^{(k)}_i(h_i^{(k)}[mB+1])f'^{(k)}(u_i^{(k)}[mB+1])h_j^{(k-1)}[mB+1]\\g'^{(k)}_i(h_i^{(k)}[mB+2])f'^{(k)}(u_i^{(k)}[mB+2])h_j^{(k-1)}[mB+2]\\\vdots\\g'^{(k)}_i(h_i^{(k)}[(m+1)B])f'^{(k)}(u_i^{(k)}[(m+1)B])h_j^{(k-1)}[(m+1)B]\end{bmatrix}^T,$$

where $\mathbf{e}_i$ represents the standard basis vector with all elements set to zero, except for the element at index $i$, which is equal to 1.

By defining the matrix

$$\boldsymbol{Q}^{(k)}[m]=\hat{\mathbf{R}}_{\mathbf{g}^{(k)}(\mathbf{h}^{(k)})\boldsymbol{\epsilon}}[m]\mathbf{E}[m]=\begin{bmatrix}\mathbf{q}^{(k)}[mB+1]&\dots&\mathbf{q}^{(k)}[(m+1)B]\end{bmatrix},$$

which represents the projection of the output error onto layer $k$, we can express the weight update as:

$$[\Delta\boldsymbol{W}_1^{(k)}[m]]_{ij}=\frac{1-\lambda}{B}\sum_{n=mB+1}^{(m+1)B}g'^{(k)}_i(h_i^{(k)}[n])f'^{(k)}(u_i^{(k)}[n])q_i^{(k)}[n]h_j^{(k-1)}[n].$$

To further simplify this expression, we introduce the matrices:

$$\mathbf{G}_d^{(k)}[m]=\begin{bmatrix}\mathbf{g}^{(k)'}(\mathbf{h}^{(k)}[mB+1])&\mathbf{g}^{(k)'}(\mathbf{h}^{(k)}[mB+2])&\dots&\mathbf{g}^{(k)'}(\mathbf{h}^{(k)}[(m+1)B])\end{bmatrix},\quad(13)$$

$$\mathbf{F}_d^{(k)}[m]=\begin{bmatrix}\mathbf{f}^{(k)'}(\mathbf{u}^{(k)}[mB+1])&\mathbf{f}^{(k)'}(\mathbf{u}^{(k)}[mB+2])&\dots&\mathbf{f}^{(k)'}(\mathbf{u}^{(k)}[(m+1)B])\end{bmatrix},\quad(14)$$

and $\boldsymbol{Z}^{(k)}[m]=\mathbf{G}_d^{(k)}[m]\odot\mathbf{F}_d^{(k)}[m]\odot\mathbf{Q}^{(k)}[m]$, which allows us to express the weight update in a more compact form:

$$\Delta\boldsymbol{W}_1^{(k)}[m]=\frac{1-\lambda}{B}\boldsymbol{Z}^{(k)}[m]\boldsymbol{H}^{(k-1)}[m]^T.$$

Following a similar procedure, the bias update is given by:

$$\Delta\mathbf{b}_1^{(k)}[m]=\frac{1-\lambda}{B}\boldsymbol{Z}^{(k)}[m]\mathbf{1}_{L\times N_{k-1}}.$$

## B.2   $\Delta\mathbf{W}_2$ AND $\Delta\mathbf{b}_2$ CALCULATION

In Section 2.3, we defined the weight update element $[\Delta\mathbf{W}_2]_{ij}$ involving the derivative of the output error as

$$\frac{1-\lambda}{B}Tr\left(\hat{\mathbf{R}}_{\mathbf{g}^{(k)}(\mathbf{h}^{(k)})\boldsymbol{\epsilon}}[m]\frac{\partial\mathbf{E}[m]}{\partial W_{ij}^{(k)}}\mathbf{G}^{(k)}[m]^T\right).$$

To begin, we consider the derivative term:

$$\frac{\partial\boldsymbol{\epsilon}}{\partial W_{ij}^{(k)}},$$

which can be expanded as

$$\frac{\partial \boldsymbol{\epsilon}}{\partial W_{ij}^{(k)}} = \underbrace{\frac{\partial \boldsymbol{\epsilon}}{\partial \mathbf{h}^{(L)}}}_{\mathbf{I}} \underbrace{\frac{\partial \mathbf{h}^{(L)}}{\partial \mathbf{u}^{(L)}}}_{\text{diag}(f'^{(k)}(\mathbf{u}^{(L)}))} \underbrace{\frac{\partial \mathbf{u}^{(L)}}{\partial \mathbf{h}^{(L-1)}}}_{\mathbf{W}^{(L)}} \underbrace{\frac{\partial \mathbf{h}^{(L-1)}}{\partial \mathbf{u}^{(L-1)}}}_{\text{diag}(f'^{(k)}(\mathbf{u}^{(L-1)}))} \cdots$$

$$\cdots \underbrace{\frac{\partial \mathbf{h}^{(k+1)}}{\partial \mathbf{u}^{(k+1)}}}_{\text{diag}(f'^{(k)}(\mathbf{u}^{(k+1)}))} \underbrace{\frac{\partial \mathbf{u}^{(k+1)}}{\partial \mathbf{h}^{(k)}}}_{\mathbf{W}^{(k+1)}} \underbrace{\frac{\partial \mathbf{h}^{(k)}}{\partial \mathbf{u}^{(k)}}}_{\text{diag}(f'^{(k)}(\mathbf{u}^{(k)}))} \underbrace{\frac{\partial \mathbf{u}^{(k)}}{\partial W_{ij}^{(k)}}}_{\mathbf{e}_i h_j^{(k-1)}}$$

This expression reflects propagation terms, from the output back to the layer $k$. Defining $\Phi^{(L)}[n] = \text{diag}(f^{(L)'}(\mathbf{u}^{(L)}[n]))$, and

$$\Phi^{(k)}[n] = \Phi^{(k+1)}[n]\mathbf{W}^{(k+1)}[m]\text{diag}(f'^{(k)}(\mathbf{u}^{(k)}[n])),$$

we obtain

$$\frac{\partial \boldsymbol{\epsilon}[n]}{\partial W_{ij}^{(k)}} = \Phi^{(k)}[n]h_j^{(k-1)}[n]\mathbf{e}_i.$$

Thus, the derivative of the error at time step $n$ with respect to $W_{ij}^{(k)}$ can be written as:

$$\frac{1-\lambda}{B}Tr\left(\mathbf{R}_{\mathbf{g}^{(k)}(\mathbf{h}^{(k)})\boldsymbol{\epsilon}}[m]\frac{\partial \mathbf{E}[m]}{\partial W_{ij}^{(k)}}\mathbf{G}^{(k)}[m]^T\right) =$$

$$\frac{1-\lambda}{B}Tr\left(\mathbf{R}_{\mathbf{g}^{(k)}(\mathbf{h}^{(k)})\boldsymbol{\epsilon}}[m]\sum_{n=mB+1}^{(m+1)B}\frac{\partial \boldsymbol{\epsilon}[n]}{\partial W_{ij}^{(k)}}\mathbf{g}^{(k)}(h^{(k)}[n])^T\right).$$

Substituting the definition $\tilde{\mathbf{g}}^{(k)}[n] = \mathbf{R}_{\mathbf{g}^{(k)}(\mathbf{h}^{(k)})\boldsymbol{\epsilon}}[m]^T\mathbf{g}^{(k)}(h^{(k)}[n])$, we obtain:

$$\frac{1-\lambda}{B}Tr\left(\mathbf{R}_{\mathbf{g}(\mathbf{h}^{(k)})\boldsymbol{\epsilon}}[m]\frac{\partial \mathbf{E}[m]}{\partial W_{ij}^{(k)}}\mathbf{G}^{(k)}[m]^T\right),$$

$$= \frac{1-\lambda}{B}Tr\left(\sum_{n=mB+1}^{(m+1)B}h_j^{(k-1)}[n]\Phi^{(k)}[n]\mathbf{e}_i\tilde{\mathbf{g}}^{(k)}[n]^T\right),$$

$$= \frac{1-\lambda}{B}\sum_{n=mB+1}^{(m+1)B}\mathbf{e}_j^T\mathbf{h}^{(k-1)}[n]\tilde{\mathbf{g}}^{(k)}[n]^T\Phi^{(k)}[n]\mathbf{e}_i,$$

$$= \mathbf{e}_i^T\left(\frac{1-\lambda}{B}\sum_{n=mB+1}^{(m+1)B}\Phi^{(k)}[n]^T\tilde{\mathbf{g}}^{(k)}[n]\mathbf{h}^{(k-1)^T}\right)\mathbf{e}_j.$$

Now, defining:

$$\psi^{(k)}[n] = \Phi^{(k)}[n]^T\tilde{\mathbf{g}}^{(k)}[n],$$

and assembling these into the matrix:

$$\boldsymbol{\Psi}^{(k)}[m] = \left[\begin{array}{cccc} \psi^{(k)}[mB+1] & \psi^{(k)}[mB+2] & \cdots & \psi^{(k)}[(m+1)B] \end{array}\right],$$

we can compactly express the weight and bias updates as:

$$\Delta \mathbf{W}_2^{(k)}[m] = \frac{1-\lambda}{B}\boldsymbol{\Psi}^{(k)}[m]\mathbf{H}^{(k-1)}[m]^T,$$

$$\Delta \mathbf{b}_2^{(k)}[m] = \frac{1-\lambda}{B}\boldsymbol{\Psi}^{(k)}[m]\mathbf{1}_{L \times N_{k-1}}.$$

### B.3   ON EBD WITH FORWARD PROJECTIONS

In the EBD algorithm introduced in Section 2.3 , output errors are broadcast to individual layers to modify their weights, thereby reducing the correlation between hidden layer activations and output errors. To enhance this mechanism, we introduce forward broadcasting, where hidden layer activations are projected onto the output layer. This projection facilitates the optimization of the decorrelation loss by adjusting the parameters of the final layer more effectively.

The purpose of forward broadcasting is to enhance the network's ability to minimize the decorrelation loss by directly influencing the final layer's weights using the activations from the hidden layers. By projecting the hidden layer activations forward onto the output layer, we establish a direct pathway for these activations to impact the adjustments of the final layer's weights. This mechanism allows the final layer to update its parameters in a way that reduces the correlation between the output errors and the hidden layer activations. Consequently, the errors at the output layer are steered toward being orthogonal to the hidden layer activations.

This mechanism could potentially be effective because the final layer is responsible for mapping the network's internal representations to the output space. By incorporating information from earlier layers, we enable the final layer to align its parameters more closely with the features that are most relevant for reducing the overall error.

While the proposed forward broadcasting mechanism is primarily motivated by performance optimization, it can conceptually be related to the long-range (Leong et al., 2016) and bottom-up (Ibrahim et al., 2021) synaptic connections in the brain, which allow certain neurons to influence distant targets. These long-range bottom-up connections are actively being researched, and incorporating similar mechanisms into computational models could enhance their alignment with biological neural processes. By integrating mechanisms that mirror these neural pathways, forward broadcasting may be useful for modeling how information is transmitted across different neural circuits.

#### B.3.1   GRADIENT DERIVATION FOR THE EBD WITH FORWARD PROJECTIONS

We derive the gradients of the layer decorrelation losses with respect to the parameters of the final layer. The partial derivative of the objective function $J^{(k)}(\mathbf{h}^{(k)}, \boldsymbol{\epsilon})$ with respect to the final layer weights can be written as:

$$
\begin{aligned}
\frac{\partial J^{(k)}(\mathbf{h}^{(k)}, \boldsymbol{\epsilon})}{\partial W_{ij}^{(L)}}[m] &= \frac{1-\lambda}{B} Tr\left( \hat{\mathbf{R}}_{\mathbf{g}(\mathbf{h}^{(k)})\boldsymbol{\epsilon}}[m] \frac{\partial(\mathbf{E}[m]\mathbf{G}^{(k)}[m]^T)}{\partial W_{ij}^{(L)}} \right) \\
&= \underbrace{\frac{1-\lambda}{B} Tr\left( \hat{\mathbf{R}}_{\mathbf{g}(\mathbf{h}^{(k)})\boldsymbol{\epsilon}}[m] \frac{\partial \mathbf{E}[m]}{\partial W_{ij}^{(L)}} \mathbf{G}^{(k)}[m]^T \right)}_{[\Delta \mathbf{W}^{(L,k),f}[m]]_{ij}}, \\
&= \frac{1-\lambda}{B} Tr\left( \mathbf{R}_{\mathbf{g}(\mathbf{h}^{(k)})\boldsymbol{\epsilon}}[m] \sum_{n=mB+1}^{(m+1)B} \frac{\partial \boldsymbol{\epsilon}[n]}{\partial W_{ij}^{(L)}} \mathbf{g}(h^{(k)}[n])^T \right).
\end{aligned}
$$

Substituting the definition $\tilde{\mathbf{g}}^{(k)}[n] = \mathbf{R}_{\mathbf{g}(\mathbf{h}^{(k)})\boldsymbol{\epsilon}}[m]^T\mathbf{g}(h^{(k)}[n])$, we can further express the partial derivative as:

$$
\begin{aligned}
\frac{\partial J^{(k)}(\mathbf{h}^{(k)}, \boldsymbol{\epsilon})}{\partial W_{ij}^{(L)}}[m] &= \frac{1-\lambda}{B}Tr\left(\sum_{n=mB+1}^{(m+1)B} h_j^{(L-1)}[n]\Phi^{(L)}[n]\mathbf{e}_i\tilde{\mathbf{g}}^{(k)}[n]^T\right), \\
&= \frac{1-\lambda}{B}\sum_{n=mB+1}^{(m+1)B} \mathbf{e}_j^T\mathbf{h}^{(L-1)}[n]\tilde{\mathbf{g}}^{(k)}[n]^T\Phi^{(L)}[n]\mathbf{e}_i, \\
&= \mathbf{e}_i^T\left(\frac{1-\lambda}{B}\sum_{n=mB+1}^{(m+1)B} \Phi^{(L)}[n]^T\tilde{\mathbf{g}}^{(k)}[n]\mathbf{h}^{(L-1)^T}\right)\mathbf{e}_j, \\
&= \mathbf{e}_i^T\left(\frac{1-\lambda}{B}\sum_{n=mB+1}^{(m+1)B} (f'(\mathbf{u}^{(L)}[n])\odot\tilde{\mathbf{g}}^{(k)}[n])\mathbf{h}^{(L-1)^T}\right)\mathbf{e}_j.
\end{aligned}
$$

Next, defining the following terms:

$$
\psi^{(k,L)}[n] = f'(\mathbf{u}^{(L)}[n])\odot\tilde{\mathbf{g}}^{(k)}[n],
$$

and assembling them into the matrix:

$$
\boldsymbol{\Psi}^{(k,L)}[m] = \begin{bmatrix} \psi^{(k,L)}[mB+1] & \psi^{(k,L)}[mB+2] & \dots & \psi[(m+1)B] \end{bmatrix},
$$

we can write the weight update as:

$$
\Delta\mathbf{W}^{(L,k),f}[m] = \frac{1-\lambda}{B}\boldsymbol{\Psi}^{(k,L)}[m]\mathbf{H}^{(k-1)}[m]^T.
$$

Following a similar procedure, the bias update can be written as:

$$
\Delta\mathbf{b}^{(L,k),f}[m] = \frac{1-\lambda}{B}\boldsymbol{\Psi}^{(k,L)}[m]\mathbf{1}_{L\times N_{k-1}}.
$$

Based on these expressions, we can write

$$
[\Delta\mathbf{W}^{(L,k),f}[m]]_{ij} = \frac{1-\lambda}{B}\sum_{n=mB+1}^{(m+1)B} f^{(L)'}(u_i^{(L)}[n])\tilde{g}_i^{(k)}[n]\mathbf{h}_j^{(L-1)}
$$

$$
[\Delta\mathbf{b}^{(L,k),f}[m]]_i = \frac{1-\lambda}{B}\sum_{n=mB+1}^{(m+1)B} f^{(L)'}(u_i^{(L)}[n])\tilde{g}_i^{(k)}[n].
$$

## C  ADDITIONAL EXTENSIONS OF EBD APPROACH

### C.1  EXTENSIONS TO CONVOLUTIONAL NEURAL NETWORKS (CNNS)

Let $\mathbf{H}^{(k)} \in \mathbb{R}^{P^{(k)} \times M^{(k)} \times N^{(k)}}$ represent the output of the $k^{th}$ layer of a Convolutional Neural Network (CNN), where $P^{(k)}$ is the number of channels and the layer's output is $M^{(k)} \times N^{(k)}$ dimensional. Furthermore, we use $\mathbf{W}^{(k,p)} \in \mathbb{R}^{P^{(k-1)} \times \Omega^{(k)} \times \Omega^{(k)}}$ and $\mathbf{b}^{(k,p)} \in \mathbb{R}$ to represent the filter tensor weights and bias coefficient respectively for the channel-$p$ of the $k^{th}$ layer, and $\Omega^{(k)}$ is the symmetric convolution kernel size. Then a convolutional layer can be defined as

$$\mathbf{H}^{(k,p)} = f(\mathcal{U}^{(k,p)}), \tag{15}$$

$$\mathcal{U}^{(k,p)} = (\mathbf{H}^{(k-1)} * \mathbf{W}^{(k,p)}) + \mathbf{b}^{(k,p)}, \tag{16}$$

where the symbol "$*$" represents the convolution [1] operation that acts upon both the spatial and channel dimensions to generate the $p^{th}$ channel of $k^{th}$ layer output $\mathbf{H}^{(k,p)}$, and $f$ is the nonlinearity acted on the convolution output.

### C.1.1  ERROR BROADCAST AND DECORRELATION FORMULATION

Similar to equation 4, we have the cross-correlation between output errors $\epsilon$ and the arbitrary function of the $k^{th}$ layer activation of the $p^{th}$ channel denoted as $\mathbf{g}^{(k)}(\mathbf{H}^{(k,p)})$, for each layer and spatial indexes $r \in \mathbb{Z} : 1 \leq r \leq M^{(k)}$ and $s \in \mathbb{Z} : 1 \leq s \leq N^{(k)}$ as

$$\mathbf{R}_{\mathbf{g}^{(k)}(\mathbf{H}^{(k,p)})\epsilon}[q,r,s] = \mathbb{E}(\mathbf{g}^{(k)}(\mathbf{H}^{(k,p)}[r,s])\epsilon_q) = \mathbf{0}. \tag{17}$$

Then this cross-correlation must ideally be zero due to the orthogonality condition. We can then write the loss for layer-$k$ at batch-$m$ as:

$$J^{(k)}(\mathbf{H}^{(k,p)}, \boldsymbol{\epsilon})[m] = \frac{1}{2} \sum_{q=1}^{n_c} \left\| \hat{\mathbf{R}}_{\mathbf{g}^{(k)}(\mathbf{H}^{(k,p)})\epsilon}[m,q,:,:] \right\|_F^2, \tag{18}$$

where $\hat{\mathbf{R}}_{\mathbf{g}(\mathbf{H}^{(k),p})\epsilon}$ is the recurrently estimated cross-correlation using the training batches. Then we can optimize the network by taking the derivative of the loss function with respect to the weight $\mathbf{W}_{hij}^{(k,p)}$ corresponding to input channel $h$ and weight spatial indexes $i,j \in \mathbb{Z} : 1 \leq i,j \leq \Omega^{(k)}$ as

$$\frac{\partial J^{(k)}(\mathbf{H}^{(k,p)}, \boldsymbol{\epsilon})[m]}{\partial \mathbf{W}_{hij}^{(k,p)}}$$

$$= \frac{1-\lambda}{B} \sum_{q=1}^{n_c} \sum_{n=mB+1}^{(m+1)B} \epsilon_q[n] \cdot Tr\left( (\hat{\mathbf{R}}_{\mathbf{g}^{(k)}(\mathbf{H}^{(k,p)})\epsilon}[m,q,:,:]^T \frac{\partial \mathbf{g}^{(k)}(\mathbf{H}^{(k,p)}[n,:,:])}{\partial \mathbf{W}_{hij}^{(k,p)}} \right) \tag{19}$$

$$= \frac{1-\lambda}{B} \sum_{q=1}^{n_c} \sum_{n=mB+1}^{(m+1)B} \sum_{r,s} \epsilon_q[n] \left[ (\hat{\mathbf{R}}_{\mathbf{g}^{(k)}(\mathbf{H}^{(k,p)})\epsilon}[m,q,:,:] \odot \frac{\partial \mathbf{g}^{(k)}(\mathbf{H}^{(k,p)}[n,:,:])}{\partial \mathbf{W}_{hij}^{(k,p)}} \right]_{[r,s]},$$

in which $n_c$ is the error dimension, $N^{(k)}$ and $M^{(k)}$ are the width and height of the $k^{th}$ layer, and the derivative with respect to the $\epsilon$ term is neglected. The inner partial derivative term can be written as

$$\frac{\partial \mathbf{g}^{(k)}(\mathbf{H}^{(k,p)}[n,:,:])}{\partial \mathbf{W}_{hij}^{(k,p)}} = \mathbf{g}'^{(k)}(\mathbf{H}^{(k,p)}[n,:,:]) \odot \frac{\partial \mathbf{H}^{(k,p)}[n,:,:]}{\partial \mathbf{W}_{hij}^{(k,p)}}, \tag{20}$$

and using the Equations (15) and (16),

$$\frac{\partial \mathbf{H}^{(k,p)}[n,:,:]}{\partial \mathbf{W}_{hij}^{(k,p)}} = f'(\mathcal{U}^{(k,p)}[n,:,:]) \odot (\mathcal{E}_{hij}^{(k)} * \mathbf{H}^{(k-1)}[n,:,:]). \tag{21}$$

---

[1] Although we call it as convolution, in CNNs, the actual operation used is the correlation operation where the kernel is unflipped.

where $\mathcal{E}_{hij}^{(k)} \in \mathbb{R}^{P^{(k-1)} \times \Omega^{(k)} \times \Omega^{(k)}}$ is a Kronecker delta tensor that occurs as the gradient of $\mathbf{W}^{(k,p)}$ with respect to $\mathbf{W}_{hij}^{(k,p)}$. Combining the expressions, we have

$$\phi[n,p,:,:] = \sum_{q=1}^{n_c} \epsilon_q[n] \cdot \left( \hat{\mathbf{R}}_{\mathbf{g}^{(k)}(\mathbf{H}^{(k,p)})\boldsymbol{\epsilon}}[n,q,:,:] \odot \mathbf{g}^{(k)}(\mathbf{H}^{(k,p)}[n,:,:]) \odot f'(\mathcal{U}^{(k,p)}[n,:,:]) \right).$$

(22)

Then, combining the Equations (19), (20), (21), and then writing the convolution explicitly, we have

$$\frac{\partial J^{(k)}(\mathbf{H}^{(k,p)}, \boldsymbol{\epsilon})[m]}{\partial \mathbf{W}_{hij}^{(k,p)}} = \frac{1-\lambda}{B} \sum_{n=mB+1}^{(m+1)B} \sum_{r,s} \left[ \phi[n,p,:,:] \odot (\mathcal{E}_{hij}^{(k)} * \mathbf{H}^{(k-1)}[n,:,:]) \right]_{[r,s]}$$

$$= \frac{1-\lambda}{B} \sum_{n=mB+1}^{(m+1)B} \sum_{r,s} \phi[n,p,r,s] \cdot \left( \sum_{h',i',j'} \mathcal{E}_{hij}^{(k)}[h',i',j'] \cdot \mathbf{H}^{(k-1,h')}[n,r+i',s+j'] \right).$$

By the definition of the delta function $\mathcal{E}_{hij}^{(k)}$ and writing the resulting expression as a 2D convolution between $\mathbf{H}^{(k-1)}$ and $\phi$ respectively, we have

$$= \frac{1-\lambda}{B} \sum_{n=mB+1}^{(m+1)B} \sum_{r,s} \phi[n,p,r,s] \cdot \mathbf{H}^{(k-1,h)}[n,r+i,s+j]$$

$$= \frac{1-\lambda}{B} \sum_{n=mB+1}^{(m+1)B} \left[ \phi[n,p,:,:] * \mathbf{H}^{(k-1,h)}[n,:,:] \right]_{[i,j]}.$$

The resulting expression for the weight update is:

$$\frac{\partial J^{(k)}(\mathbf{H}^{(k,p)}, \boldsymbol{\epsilon})[m]}{\partial \mathbf{W}_h^{(k,p)}} = \frac{1-\lambda}{B} \sum_{n=mB+1}^{(m+1)B} (\phi[n,p,:,:] * \mathbf{H}^{(k-1,h)}[n,:,:]).$$

(23)

Similarly, it can be shown that the bias update:

$$\frac{\partial J^{(k)}(\mathbf{H}^{(k,p)}, \boldsymbol{\epsilon})[m]}{\partial \mathbf{b}^{(k,p)}} = \frac{1-\lambda}{B} \sum_{n=mB+1}^{(m+1)B} \sum_{r=1}^{N^{(k)}} \sum_{s=1}^{M^{(k)}} \phi[n,p,r,s].$$

The convolutional layer parameters can be trained using these gradient formulas for each layer separately, and can be calculated by utilizing the batched convolution operation.

### C.1.2 WEIGHT ENTROPY OBJECTIVE

The layer entropy objective is computationally cumbersome for a convolutional layer that has multiple dimensions. Therefore, we propose the weight-entropy objective to avoid dimensional collapse

$$J_E^{(k)}(\mathbf{W}^{(k)}) = \frac{1}{2} \log \det(\mathbf{R}_{\overline{\mathbf{W}}^{(k)}} + \varepsilon^{(k)} \mathbf{I}),$$

where we define $\overline{\mathbf{W}}^{(k)} \in \mathbb{R}^{P^{(k)} \times P^{(k-1)}.\Omega^{(k)}.\Omega^{(k)}}$ as the unraveled version of the full size weight tensor $\mathbf{W}^{(k)}$, and the covariance matrix $\mathbf{R}_{\overline{\mathbf{W}}^{(k)}}$ is conditionally defined as:

$$\mathbf{R}_{\overline{\mathbf{W}}^{(k)}} = \begin{cases} \overline{\mathbf{W}}^{(k)T} \overline{\mathbf{W}}^{(k)}, & \text{if } P^{(k)} \geq P^{(k-1)}.\Omega^{(k)}.\Omega^{(k)}, \\ \overline{\mathbf{W}}^{(k)} \overline{\mathbf{W}}^{(k)T}, & \text{otherwise}, \end{cases}$$

to decrease its dimensions and reduce the computational costs for further steps. Then, the derivative of this objective can be written as:

$$\Delta J_E^{(k)}(\mathbf{W}^{(k)}) = \begin{cases} \overline{\mathbf{W}}^{(k)} \mathbf{R}_{\overline{\mathbf{W}}^{(k)}}^{-1}, & \text{if } P^{(k)} \geq P^{(k-1)}.\Omega^{(k)}.\Omega^{(k)}, \\ \mathbf{R}_{\overline{\mathbf{W}}^{(k)}}^{-1} \overline{\mathbf{W}}^{(k)}, & \text{otherwise}. \end{cases}$$

Therefore, $\frac{\partial J_E(\mathbf{W}^{(k)})}{\partial W_{hij}^{(k,p)}}$ can be obtained by reshaping $\Delta J_E^{(k)}(\mathbf{W}^{(k)})$ as the weight tensor $\mathbf{W}^{(k)}$.

### C.1.3 ACTIVATION SPARSITY REGULARIZATION

To further regularize the model, we enforce the layer activation sparsity loss that is given as

$$J_{\ell_1}^{(k)}(\mathbf{H}^{(k,p)}) = \frac{\|\mathbf{H}^{(k,p)}\|_1}{\|\mathbf{H}^{(k,p)}\|_2}. \tag{24}$$

The gradient of the sparsity loss with respect to the hidden layer can be written as:

$$\Delta J_{\ell_1}^{(k)}(\mathbf{H}^{(k,p)}) = \frac{1}{\|\mathbf{H}^{(k,p)}\|_2}\text{sign}(\mathbf{H}^{(k,p)}) - \frac{\|\mathbf{H}^{(k,p)}\|}{\|\mathbf{H}^{(k,p)}\|_2^3}\mathbf{H}^{(k,p)}. \tag{25}$$

Then, the gradient of the loss with respect to the model weights can be calculated in a similar manner as the Equation (23):

$$\frac{\partial J_{\ell_1}^{(k)}(\mathbf{H}^{(k,p)})[m]}{\partial \mathbf{W}_h^{(k,p)}} = \frac{1}{B}\sum_{n=mB+1}^{(m+1)B}\left(\Delta J_{\ell_1}^{(k)}(\mathbf{H}^{(k,p)})[n,p,:,:] * \mathbf{H}^{(k-1,h)}[n,:,:]\right). \tag{}$$

### C.2 EXTENSIONS TO LOCALLY CONNECTED (LC) NETWORKS

Let $\mathbf{H}^{(k)} \in \mathbb{R}^{P^{(k)} \times M^{(k)} \times N^{(k)}}$ represent the output of the $k^{th}$ layer of a Locally Connected Network (LC), where $P^{(k)}$ is the number of channels and the layer's output is $M^{(k)} \times N^{(k)}$ dimensional. We use $\mathbf{W}^{(k,p,r,s)} \in \mathbb{R}^{P^{(k-1)} \times \Omega^{(k)} \times \Omega^{(k)}}$ and $\mathbf{b}^{(k,p,r,s)} \in \mathbb{R}$ to represent the filter tensor weights and bias coefficient at spatial locations $r \in \mathbb{Z}: 1 \le r \le M^{(k)}$ and $s \in \mathbb{Z}: 1 \le s \le N^{(k)}$, for channel-$p$ of the $k^{th}$ layer, where $\Omega^{(k)}$ is the local receptive field size. Then a locally connected layer can be defined as

$$\mathbf{H}^{(k,p)} = f(\mathcal{U}^{(k,p)}), \tag{26}$$
$$\mathcal{U}^{(k,p)} = (\mathbf{H}^{(k-1)} \circledast \mathbf{W}^{(k,p)}) + \mathbf{b}^{(k,p)}, \tag{27}$$

where the symbol "$\circledast$" represents the locally connected operation which acts upon both the spatial and channel dimensions, but without weight sharing across spatial locations, generating the $p^{th}$ channel of the $k^{th}$ layer output $\mathbf{H}^{(k,p)}$, and $f$ is the nonlinearity applied to the result.

### C.2.1 ERROR BROADCAST AND DECORRELATION FORMULATION

For the LC network, the orthogonality condition and the corresponding loss $J^{(k)}(\mathbf{H}^{(k,p)}, \boldsymbol{\epsilon})[m]$ for layer-$k$ at batch-$m$ can be written equivalently as Equations (17) and (18) respectively. Then the optimization can be performed by taking the derivative of the loss function with respect to $\mathbf{W}_{hij}^{(k,p,r,s)}$ corresponding to input channel $h$, weight spatial indexes $i,j \in \mathbb{Z}: 1 \le i,j \le \Omega^{(k)}$ as

$$\frac{\partial J^{(k)}(\mathbf{H}^{(k,p)}, \boldsymbol{\epsilon})[m]}{\partial \mathbf{W}_{hij}^{(k,p,r,s)}}$$

$$= \frac{1-\lambda}{B}\sum_{q=1}^{n_c}\sum_{n=mB+1}^{(m+1)B}\epsilon_q[n] \cdot Tr\left((\hat{\mathbf{R}}_{\mathbf{g}^{(k)}(\mathbf{H}^{(k,p)})\boldsymbol{\epsilon}}[m,q,:,:]^T\frac{\partial \mathbf{g}^{(k)}(\mathbf{H}^{(k,p)}[n,:,:])}{\partial \mathbf{W}_{hij}^{(k,p,r,s)}}\right) \tag{28}$$

$$= \frac{1-\lambda}{B}\sum_{q=1}^{n_c}\sum_{n=mB+1}^{(m+1)B}\sum_{r,s}\epsilon_q[n]\left[(\hat{\mathbf{R}}_{\mathbf{g}^{(k)}(\mathbf{H}^{(k,p)})\boldsymbol{\epsilon}}[m,q,:,:] \odot \frac{\partial \mathbf{g}^{(k)}(\mathbf{H}^{(k,p)}[n,:,:])}{\partial \mathbf{W}_{hij}^{(k,p,r,s)}}\right]_{[r,s]}.$$

The inner partial derivative term can be written as

$$\frac{\partial \mathbf{g}^{(k)}(\mathbf{H}^{(k,p)}[n,:,:])}{\partial \mathbf{W}_{hij}^{(k,p,r,s)}} = \mathbf{g}'^{(k)}(\mathbf{H}^{(k,p)}[n,:,:]) \odot \frac{\partial \mathbf{H}^{(k,p)}[n,:,:]}{\partial \mathbf{W}_{hij}^{(k,p,r,s)}}, \tag{29}$$

and using Equations (26) and (27), we obtain:

$$\frac{\partial \mathbf{H}^{(k,p)}[n,:,:]}{\partial \mathbf{W}_{hij}^{(k,p,r,s)}} = f'(\mathcal{U}^{(k,p)}[n,:,:]) \odot (\mathcal{E}_{hij}^{(k)} \circledast \mathbf{H}^{(k-1)}[n,:,:]). \tag{30}$$

Here, $\mathcal{E}_{hij}^{(k,r,s)} \in \mathbb{R}^{P^{(k-1)} \times \Omega^{(k)} \times \Omega^{(k)}} \times M^{(k)} \times N^{(k)}$ is a Kronecker delta tensor that occurs as the gradient of $\mathbf{W}^{(k,p)}$ with respect to $\mathbf{W}_{hij}^{(k,p,r,s)}$. Combining the expressions in (28), (29), (30), and the expression for $\phi$ as in (22) which is equivalent for both CNNs and LCs, and then writing the locally connected operation explicitly, we have

$$\frac{\partial J^{(k)}(\mathbf{H}^{(k,p)}, \boldsymbol{\epsilon})[m]}{\partial \mathbf{W}_{hij}^{(k,p,r,s)}} = \frac{1-\lambda}{B} \sum_{n=mB+1}^{(m+1)B} \sum_{r,s} \left[ \phi[n,p,:,:] \odot (\mathcal{E}_{hij}^{(k,r,s)} \circledast \mathbf{H}^{(k-1)}[n,:,:]) \right]_{[r,s]}$$

$$= \frac{1-\lambda}{B} \sum_{n=mB+1}^{(m+1)B} \phi[n,p,r,s] \cdot \left( \sum_{\substack{h',i',j' \\ r',s'}} \mathcal{E}_{hij}^{(k,r,s)}[h',i',j',r',s'] \cdot \mathbf{H}^{(k-1,h')}[n,r'+i',s'+j'] \right).$$

Then, by the definition of the Kronecker delta, the resulting expression for the weight update is:

$$\frac{\partial J^{(k)}(\mathbf{H}^{(k,p)}, \boldsymbol{\epsilon})[m]}{\partial \mathbf{W}_{hij}^{(k,p,r,s)}} = \frac{1-\lambda}{B} \sum_{n=mB+1}^{(m+1)B} \left( \phi[n,p,r,s] \cdot \mathbf{H}^{(k-1,h)}[n,r+i,s+j] \right). \tag{31}$$

Similarly, it can be shown that the bias update is:

$$\frac{\partial J^{(k)}(\mathbf{H}^{(k,p)}, \boldsymbol{\epsilon})[m]}{\partial \mathbf{b}^{(k,p,r,s)}} = \frac{1-\lambda}{B} \sum_{n=mB+1}^{(m+1)B} \phi[n,p,r,s].$$

### C.2.2 WEIGHT ENTROPY OBJECTIVE

Similar to CNNs, we propose the weight-entropy objective to avoid dimensional collapse in LCs

$$J_E^{(k)}(\mathbf{W}^{(k)}) = \frac{1}{2} \log \det(\mathbf{R}_{\overline{\mathbf{W}}^{(k)}} + \varepsilon^{(k)}\mathbf{I}),$$

where we define $\overline{\mathbf{W}}^{(k)} \in \mathbb{R}^{P^{(k)} \times P^{(k-1)} \cdot M^{(k)} \cdot N^{(k)} \cdot \Omega^{(k)} \cdot \Omega^{(k)}}$ as the unraveled version of the full size weight tensor $\mathbf{W}^{(k)}$, then the covariance matrix $\mathbf{R}_{\overline{\mathbf{W}}^{(k)}}$ is defined as:

$$\mathbf{R}_{\overline{\mathbf{W}}^{(k)}} = \overline{\mathbf{W}}^{(k)T} \overline{\mathbf{W}}^{(k)}.$$

Then, the derivative of this objective can be written as:

$$\Delta J_E^{(k)}(\mathbf{W}^{(k)}) = \overline{\mathbf{W}}^{(k)} \mathbf{R}_{\overline{\mathbf{W}}^{(k)}}^{-1}$$

$\frac{\partial J_E(\mathbf{W}^{(k)})}{\partial W_{hij}^{(k,p,r,s)}}$ can be obtained by reshaping $\Delta J_E^{(k)}(\mathbf{W}^{(k)})$ as the weight tensor $\mathbf{W}^{(k)}$.

### C.2.3 ACTIVATION SPARSITY REGULARIZATION

The layer activation sparsity loss for the LC is the same as the one given for the CNN in (24), with its gradient with respect to the activations as in (25). Then, the gradient of the loss with respect to the model weights can be calculated in a similar manner as the expression (31):

$$\frac{\partial J_{\ell_1}^{(k)}(\mathbf{H}^{(k)})[m]}{\partial \mathbf{W}_{hij}^{(k,p,r,s)}} = \frac{1}{B} \sum_{n=mB+1}^{(m+1)B} \left( \Delta J_{\ell_1}^{(k)}(\mathbf{H}^{(k)})[n,p,r,s] \circledast \mathbf{H}^{(k-1,h)}[n,r+i,s+j] \right).$$

## D BACKGROUND ON ONLINE CORRELATIVE INFORMATION MAXIMIZATION BASED BIOLOGICALLY PLAUSIBLE NEURAL NETWORKS

Bozkurt et al. (2023) recently proposed a framework, which we refer as CorInfoMax-EP, to address weight symmetry problem corresponding to backpropagation algorithm. In this section, we provide a brief summary of this framework.

The CorInfoMax-EP framework utilizes an online optimization setting to maximize correlative information between two consecutive layers:

$$\sum_{k=0}^{L-1} \hat{I}^{(\epsilon)}(\mathbf{h}^{(k)}, \mathbf{h}^{(k+1)})[m] - \frac{\beta}{2}\|\mathbf{y}[m] - \mathbf{h}^{(L)}[m]\|_2^2,$$

where $\hat{I}^{(\epsilon)}(\mathbf{h}^{(k)}, \mathbf{h}^{(k+1)})[m]$ is the correlative mutual information between layers $k$ and $k+1$, and the term on the left corresponds to the mean square error between the network output $\mathbf{h}^{(L)}[m]$ and the training label $\mathbf{y}[m]$. This framework utilizes two alternative but equivalent forms for the correlative mutual information

$$\hat{\overrightarrow{I}}^{(\varepsilon_k)}(\mathbf{h}^{(k)}, \mathbf{h}^{(k+1)})[m] = \frac{1}{2}\log\det(\hat{\mathbf{R}}_{\mathbf{h}^{(k+1)}}[m] + \varepsilon_k \boldsymbol{I}) - \frac{1}{2}\log\det(\hat{\boldsymbol{R}}_{\overrightarrow{\mathbf{e}}_*^{(k+1)}}[m] + \varepsilon_k \boldsymbol{I}),$$

$$\hat{\overleftarrow{I}}^{(\varepsilon_k)}(\mathbf{h}^{(k)}, \mathbf{h}^{(k+1)})[m] = \frac{1}{2}\log\det(\hat{\mathbf{R}}_{\mathbf{h}^{(k)}}[m] + \varepsilon_k \boldsymbol{I}) - \frac{1}{2}\log\det(\hat{\boldsymbol{R}}_{\overleftarrow{\mathbf{e}}_*^{(k)}}[m] + \varepsilon_k \boldsymbol{I}),$$

defined in terms of the correlation matrices of layer activations, i.e., $\hat{\mathbf{R}}_{\mathbf{h}^{(k)}}$ and the correlation matrices of forward and backward prediction errors ($\hat{\boldsymbol{R}}_{\overrightarrow{\mathbf{e}}_*^{(k+1)}}$ and $\hat{\boldsymbol{R}}_{\overleftarrow{\mathbf{e}}_*^{(k)}}$) between two consequitive layers. Here, forward/backward prediction errors are defined by

$$\overrightarrow{\mathbf{e}}_*^{(k+1)}[n] = \mathbf{h}^{(k+1)}[n] - \mathbf{W}^{(f,k)}[m]\mathbf{h}^{(k)}[n], \qquad \overleftarrow{\mathbf{e}}_*^{(k)}[n] = \mathbf{h}^{(k)}[n] - \mathbf{W}^{(b,k)}[m]\mathbf{h}^{(k+1)}[n],$$

respectively. Here, $\mathbf{W}^{(f,k)}[m]$ ($\mathbf{W}^{(b,k)}[m]$) is the forward (backward) prediction matrix for layer $k$.

In order to enable online implementation, the exponentially weighted correlation matrices for hidden layer activations and prediction errors are defined as follows:

$$\hat{\mathbf{R}}_{\mathbf{h}^{(k)}}[m] = \frac{1-\lambda}{1-\lambda^m}\sum_{i=1}^{m}\lambda^{m-i}\mathbf{h}^{(k)}[m]\mathbf{h}^{(k)}[m]^T,$$

$$\hat{\mathbf{R}}_{\overrightarrow{\mathbf{e}}^{(k)}}[m] = \frac{1-\lambda}{1-\lambda^m}\sum_{i=1}^{m}\lambda^{m-i}\overrightarrow{\mathbf{e}}^{(k)}[m]\overrightarrow{\mathbf{e}}^{(k)}[m]^T,$$

$$\hat{\mathbf{R}}_{\overleftarrow{\mathbf{e}}^{(k)}}[m] = \frac{1-\lambda}{1-\lambda^m}\sum_{i=1}^{m}\lambda^{m-i}\overleftarrow{\mathbf{e}}^{(k)}[m]\overleftarrow{\mathbf{e}}^{(k)}[m]^T.$$

Through the trace approximation of $\log\det(\cdot)$ function, we obtain:

$$\log\det\left(\hat{\boldsymbol{R}}_{\overrightarrow{\mathbf{e}}^{(k+1)}}[m] + \varepsilon\boldsymbol{I}\right)$$

$$\approx \frac{1}{\varepsilon_k}\sum_{i=1}^{t}\lambda^{t-i}\|\mathbf{h}^{(k+1)}[i] - \mathbf{W}_{ff,*}^{(k)}[m]\mathbf{h}^{(k)}[i]\|_2^2 + \varepsilon_k\|\boldsymbol{W}_{ff,*}^{(k)}[m]\|_F^2 + N_{k+1}\log(\varepsilon_k)$$

$$\log\det\left(\hat{\boldsymbol{R}}_{\overleftarrow{\mathbf{e}}^{(k)}}[m] + \varepsilon_k\boldsymbol{I}\right)$$

$$\approx \frac{1}{\varepsilon_k}\sum_{i=1}^{t}\lambda^{t-i}\|\mathbf{h}^{(k)}[i] - \mathbf{W}_{fb,*}^{(k)}[m]\mathbf{h}^{(k+1)}[i]\|_2^2 + \varepsilon_k\|\boldsymbol{W}_{fb,*}^{(k)}[m]\|_F^2 + N_k\log(\varepsilon_k),$$

### D.1 THE DERIVATION OF THE CORINFOMAX NETWORK

Based on the definitions above, the following layerwise objectives can be defined:

$$\hat{J}_k(\mathbf{h}^{(k)})[m] = \hat{\overrightarrow{I}}^{(\epsilon_{k-1})}(\mathbf{h}^{(k-1)}, \mathbf{h}^{(k)})[m] + \hat{\overleftarrow{I}}^{(\varepsilon_k)}(\mathbf{h}^{(k)}, \mathbf{h}^{(k+1)})[m], \text{ for } k = 1, \ldots, L-1,$$

i.e., correlative information maximization objectives for the hidden layers, and the mixture of correlation maximization and MSE objectives for the final layer

$$\hat{J}_L(\mathbf{h}^{(L)})[m] = \hat{I}^{(\overrightarrow{\epsilon_{L-1}})}(\mathbf{h}^{(L-1)}, \mathbf{h}^{(L)})[m] - \frac{\beta}{2}\|\mathbf{h}^{(L)}[m] - \mathbf{y}[m]\|_2^2.$$

The gradient of the hidden layer objective functions with respect to the corresponding layer activations can be written as:

$$\nabla_{\mathbf{h}^{(k)}}\hat{J}_k(\mathbf{h}^{(k)})[m] = 2\gamma \boldsymbol{B}_{\mathbf{h}^{(k)}}[m]\mathbf{h}^{(k)}[m] - \frac{1}{\epsilon_{k-1}}\overrightarrow{\mathbf{e}}^{(k)}[m] - \frac{1}{\varepsilon_k}\overleftarrow{\mathbf{e}}^{(k)}[m], \tag{32}$$

where $\gamma = \frac{1-\lambda}{\lambda}$, and $\boldsymbol{B}_{\mathbf{h}^{(k)}}[m] = (\hat{\mathbf{R}}_{\mathbf{h}^{(k)}}[m] + \epsilon_{k-1}\boldsymbol{I})^{-1}$, i.e., the inverse of the layer correlation matrix.

For the output layer, we can write the gradient as

$$\nabla_{\mathbf{h}^{(L)}}\hat{J}_L(\mathbf{h}^{(L)})[m] = \gamma \boldsymbol{B}_{\mathbf{h}^{(L)}}[m]\mathbf{h}^{(L)}[m] - \frac{1}{\epsilon_{L-1}}\overrightarrow{\mathbf{e}}^{(L)}[m] - \beta(\mathbf{h}^{(L)}[m] - \boldsymbol{y}[m]).$$

The gradient ascent updates corresponding to these expressions can be organized to obtain CorInfoMax network dynamics:

$$\tau_{\mathbf{u}}\frac{d\mathbf{u}^{(k)}[m;s]}{ds} = -g_{lk}\mathbf{u}^{(k)}[m;s] + \frac{1}{\varepsilon_k}\boldsymbol{M}^{(k)}[m]\mathbf{h}^{(k)}[m;s] - \frac{1}{\epsilon_{k-1}}\overrightarrow{\mathbf{e}}_u^{(k)}[m;s] - \frac{1}{\epsilon_k}\overleftarrow{\mathbf{e}}_u^{(k)}[m;s],$$

$$\overrightarrow{\mathbf{e}}_u^{(k)}[m;s] = \mathbf{u}^{(k)}[m;s] - \boldsymbol{W}_{ff}^{(k-1)}[t]\mathbf{h}^{(k-1)}[m;s],$$

$$\overleftarrow{\mathbf{e}}_u^{(k)}[m;s] = \mathbf{u}^{(k)}[m;s] - \boldsymbol{W}_{fb}^{(k)}[m]\mathbf{h}^{(k+1)}[m;s],$$

$$\mathbf{h}^{(k)}[m;s] = \sigma_+(\mathbf{u}^{(k)}[m;s]),$$

where $m$ is the sample index, $s$ is the time index for the network dynamics, $\tau_{\mathbf{u}}$ is the update time constant, $\boldsymbol{M}^{(k)}[t] = \varepsilon_k(2\gamma\boldsymbol{B}_{\mathbf{h}^{(k)}}[t] + g_{lk}\boldsymbol{I})$, and $\sigma_+ = \min(1, \max(u, 0))$ represents the elementwise clipped-ReLU function, which is the projection operation corresponding to the combination of the nonnegativity constraint $\mathbf{h}^{(k)} \geq 0$ and the boundedness constraint $\|\mathbf{h}^{(k)}\|_\infty \leq 1$ on the activations of the network.

Note that Bozkurt et al. (2023) takes one more step to organize the network dynamics into a form that fits into the form of a network with three compartment (soma, basal dendrite and appical dendrite compartments) neuron model.

## D.2  CorInfoMax-EP Learning Dynamics

The CorInfoMax-EP framework in Bozkurt et al. (2023) employs equilibrium propagation(EP) to update feedforward and feedback weights of the CorInfoMax network.

### D.2.1  Feedforward and feedback weights

In the CorInfoMax objective, feedforward and feedback weights correspond to forward and backward predictors corresponding to the regularized least squares objectives

$$C_{ff}(\boldsymbol{W}_{ff}^{(k)}[m]) = \varepsilon_k\|\boldsymbol{W}_{ff}^{(k)}[m]\|_F^2 + \|\overrightarrow{\mathbf{e}}^{(k+1)}[m]\|_2^2,$$

and

$$C_{fb}(\boldsymbol{W}_{fb}^{(k)}[m]) = \varepsilon_k\|\boldsymbol{W}_{ff}^{(k)}[m]\|_F^2 + \|\overleftarrow{\mathbf{e}}^{(k)}[m]\|_2^2,$$

respectively. The derivatives of these functions with respect to forward and backward synaptic weights can be written as

$$\frac{\partial C_{ff}(\boldsymbol{W}_{ff}^{(k)}[m])}{\partial \boldsymbol{W}_{ff}^{(k)}[m]} = 2\varepsilon_k\boldsymbol{W}_{ff}^{(k)}[m] - 2\overrightarrow{\mathbf{e}}^{(k+1)}[m]\mathbf{h}^{(k)}[m]^T,$$

and

$$\frac{\partial C_{fb}(\boldsymbol{W}_{fb}^{(k)}[m])}{\partial \boldsymbol{W}_{fb}^{(k)}[m]} = 2\varepsilon_k \boldsymbol{W}_{fb}^{(k)}[m] - 2\overleftarrow{\mathbf{e}}^{(k)}[m]\mathbf{h}^{(k+1)}[m]^T.$$

The EP based updates of the feedforward and feedback weights are obtained by evaluating these gradients in two different phases: the nudge phase ($\beta = \beta' > 0$), and the free phase ($\beta = 0$):

$$\delta \boldsymbol{W}_{ff}^{(k)}[m] \propto \frac{1}{\beta'} \left( (\overrightarrow{\mathbf{e}}^{(k+1)}[m]\mathbf{h}^{(k)}[m]^T)\Big|_{\beta=\beta'} - (\overrightarrow{\mathbf{e}}^{(k+1)}[m]\mathbf{h}^{(k)}[m]^T)\Big|_{\beta=0} \right),$$

$$\delta \boldsymbol{W}_{fb}^{(k)}[m] \propto \frac{1}{\beta'} \left( (\overleftarrow{\mathbf{e}}^{(k)}[m]\mathbf{h}^{(k+1)}[m]^T)\Big|_{\beta=\beta'} - (\overleftarrow{\mathbf{e}}^{(k)}[m]\mathbf{h}^{(k+1)}[m]^T)\Big|_{\beta=0} \right).$$

### D.2.2 LATERAL WEIGHTS

The lateral weight updates derived from the weight correlation matrices of the layer activations, using the matrix inversion lemma (Kailath et al., 2000):

$$\boldsymbol{B}^{(k)}[m+1] = \lambda_{\mathbf{r}}^{-1}(\boldsymbol{B}^{(k)}[m] - \gamma \mathbf{z}^{(k)}[m]\mathbf{z}^{(k)}[m]^T), \text{ where } \mathbf{z}^{(k)}[m] = \boldsymbol{B}^{(k)}[m]\mathbf{r}^{(k)}[m]\Big|_{\beta=\beta'}.$$

### D.3 CORINFOMAX-EP

Although the CorInfoMax-EP algorithm derivation above is based on single input sample based updates, it can be extendable to batch updates. Assuming a batch size of $B$, and we define the following matrices:

$$\mathbf{H}^{(k)}[m] = \left[ \begin{array}{cccc} \mathbf{h}^{(k)}[mB+1] & \mathbf{h}^{(k)}[mB+2] & \dots & \mathbf{h}^{(k)}[(m+1)B] \end{array} \right],$$

as the activation matrix for the layer-$k$,

$$\overleftarrow{\mathbf{E}}^{(k)}[m] = \left[ \begin{array}{cccc} \overleftarrow{\mathbf{e}}^{(k)}[mB+1] & \overleftarrow{\mathbf{e}}^{(k)}[mB+2] & \dots & \overleftarrow{\mathbf{e}}^{(k)}[(m+1)B] \end{array} \right], \tag{33}$$

as the backward prediction matrix for the layer-$k$,

$$\overrightarrow{\mathbf{E}}^{(k)}[m] = \left[ \begin{array}{cccc} \overrightarrow{\mathbf{e}}^{(k)}[mB+1] & \overrightarrow{\mathbf{e}}^{(k)}[mB+2] & \dots & \overrightarrow{\mathbf{e}}^{(k)}[(m+1)B] \end{array} \right], \tag{34}$$

as the forward prediction matrix for the layer-$k$,

$$\mathbf{Z}^{(k)}[m] = \left[ \begin{array}{cccc} \mathbf{z}^{(k)}[mB+1] & \mathbf{z}^{(k)}[mB+2] & \dots & \mathbf{z}^{(k)}[(m+1)B] \end{array} \right], \tag{35}$$

as the lateral weights' output matrix for the layer-$k$, and

$$\mathbf{E} = \left[ \begin{array}{cccc} \boldsymbol{\epsilon}[mB+1] & \boldsymbol{\epsilon}[mB+2] & \dots & \boldsymbol{\epsilon}[(m+1)B] \end{array} \right],$$

as the output error matrix.

In terms of these definitions, Algorithm 2 lays out the details of the CorInfoMax-EP algorithm:

---

**Algorithm 2** CorInfoMax Equilibrium Propagation (CorInfoMax-EP) Update for Layer $k$

---

**Require:** Learning rate parameters $\lambda_E$, $\mu^{(f,k)}[m]$, $\mu^{(b,k)}[m]$

**Require:** Previous synaptic weights $\mathbf{W}^{(f,k)}[m-1]$ (forward), $\mathbf{W}^{(b,k)}[m-1]$ (backward), $\mathbf{B}^{(k)}$ (lateral)

**Require:** Batch size $B$

**Require:** Layer activations $\mathbf{H}^{(k)}[m]$, preactivations $\mathbf{U}^{(k)}[m]$, output errors $\mathbf{E}^{(k)}[m]$, lateral weight outputs $\mathbf{Z}^{(k)}[m]$, forward prediction errors $\overrightarrow{\mathbf{E}}^{(k)}[m]$ and backward prediction errors $\overleftarrow{\mathbf{E}}^{(k)}[m]$ computed by CorInfoMax network dynamics described in Bozkurt et al. (2023)

**Ensure:** Updated weights $\mathbf{W}^{(f,k)}[m]$, $\mathbf{W}^{(b,k)}[m]$ ,$\mathbf{B}^{(k)}[m]$

1 $\gamma_E \leftarrow \frac{1-\lambda_E}{\lambda_E}$

**Update forward weights for layer $k$:**

2 $\Delta\mathbf{W}_{\text{EP}}^{(f,k)}[m] \leftarrow -\frac{\mu^{(d_f,k)}[m]}{B\beta'} \left( \left. (\overrightarrow{\mathbf{E}}^{(k+1)}[m]\mathbf{H}^{(k)}[m]^T) \right|_{\beta=\beta'} - \left. (\overrightarrow{\mathbf{E}}^{(k+1)}[m]\mathbf{H}^{(k)}[m]^T) \right|_{\beta=0} \right)$

3 $\mathbf{W}^{(f,k)}[m] \leftarrow \mathbf{W}^{(f,k)}[m-1] + \Delta\mathbf{W}_{\text{EP}}^{(f,k)}[m]$

**Update backward weights for layer $k$:**

4 $\Delta\mathbf{W}_{\text{EP}}^{(b,k)}[m] \leftarrow -\frac{\mu^{(d_b,k)}[m]}{B\beta'} \left( \left. (\overleftarrow{\mathbf{E}}^{(k)}[m]\mathbf{H}^{(k)}[m]^T) \right|_{\beta=\beta'} - \left. (\overleftarrow{\mathbf{E}}^{(k)}[m]\mathbf{H}^{(k)}[m]^T) \right|_{\beta=0} \right)$

5 $\mathbf{W}^{(b,k)}[m] \leftarrow \mathbf{W}^{(b,k)}[m-1] + \Delta\mathbf{W}_{\text{EP}}^{(b,k)}[m]$

**Update Lateral weights for layer $k$:**

6 $\Delta\mathbf{B}_{\text{E}}^{(k)}[m] \leftarrow -\frac{\gamma_E}{B}\mathbf{Z}^{(k)}[m]\mathbf{Z}^{(k)}[m]^T$

7 $\mathbf{B}^{(k)}[m] \leftarrow \frac{1}{\lambda_E}\mathbf{B}^{(k)}[m] + \Delta\mathbf{B}_{\text{E}}^{(k)}[m]$

---

# E  SUPPLEMENTARY ON NUMERICAL EXPERIMENTS

The models were trained on an NVIDIA Tesla V100 GPU, using the hyperparameters detailed in the sections below. Each experiment was conducted five times under identical settings, and the reported results reflect the average performance. We used the standard train/test splits for the datasets, with MNIST comprising 60,000 training examples and CIFAR-10 comprising 50,000, while both datasets included 10,000 test examples. Rather than utilizing automatic differentiation tools, we manually implemented the gradient calculations for the EBD algorithm, utilizing batched operations to ensure computational efficiency. As a side note, the $(1-\lambda)$ factors present in the derived update expressions are absorbed into the learning rate constants and thus eliminated. In our experiments, we trained the MLP models for **120** epochs and the CNN and LC models for **100** epochs on MNIST and **200** epochs on CIFAR-10. Also, we trained the CorInfoMax-EBD model for **60** epochs.

## E.1  ARCHITECTURES

The architectural details of MLP, CNN and LC networks for the MNIST and CIFAR-10 datasets are shown in Tables 4 and 5, respectively. The structure of the models are the same as in the reference Clark et al. (2021). In all architectures, we used ReLU as the nonlinear functions except the last layer.

Table 4: MNIST architectures. **FC:** fully connected layer. **Conv:** convolutional layer. **LC:** locally connected layer. For fully connected layers, `layer_size` is shown which corresponds to the size of the hidden layer. For convolutional and locally connected layers, (`num_channels`, `kernel_size`, `stride`, `padding`) are shown.

| MLP | | Convolutional | | Locally connected | |
|---|---|---|---|---|---|
| FC1 | 1024 | Conv1 | $64, 3 \times 3, 1, 1$ | LC1 | $32, 3 \times 3, 1, 1$ |
| FC2 | 512 | AvgPool | $2 \times 2$ | AvgPool | $2 \times 2$ |
| | | Conv2 | $32, 3 \times 3, 1, 1$ | LC2 | $32, 3 \times 3, 1, 1$ |
| | | AvgPool | $2 \times 2$ | AvgPool | $2 \times 2$ |
| | | FC1 | 1024 | FC1 | 1024 |

Table 5: CIFAR-10 architectures. Conventions are the same as in Table 4.

| MLP | | Convolutional | | Locally connected | |
|---|---|---|---|---|---|
| FC1 | 1024 | Conv1 | $128, 5 \times 5, 1, 2$ | LC1 | $64, 5 \times 5, 1, 2$ |
| FC2 | 512 | AvgPool | $2 \times 2$ | AvgPool | $2 \times 2$ |
| FC3 | 512 | Conv2 | $64, 5 \times 5, 1, 2$ | LC2 | $32, 5 \times 5, 1, 2$ |
| | | AvgPool | $2 \times 2$ | AvgPool | $2 \times 2$ |
| | | Conv3 | $64, 2 \times 2, 2, 0$ | LC3 | $32, 2 \times 2, 2, 0$ |
| | | FC1 | 1024 | FC1 | 512 |

The architectural details of the biologically more realistic CorInfoMax network for the MNIST and CIFAR-10 datasets are shown in Table 6. These techniques are the same as the examples in Appendix J.5 of Bozkurt et al. (2023).

Table 6: CorInfoMax architectures. Conventions are the same as in Table 4.

| MNIST | | CIFAR-10 | |
|---|---|---|---|
| FC1 | 500 | FC1 | 1000 |
| FC2 | 500 | FC2 | 500 |

## E.2 ACCURACY AND LOSS CURVES

Figures 2 and 3 present the training/test accuracy and MSE loss curves over epochs for the CIFAR-10 and MNIST datasets. Solid lines represent test curves; dashed lines denote training curves.

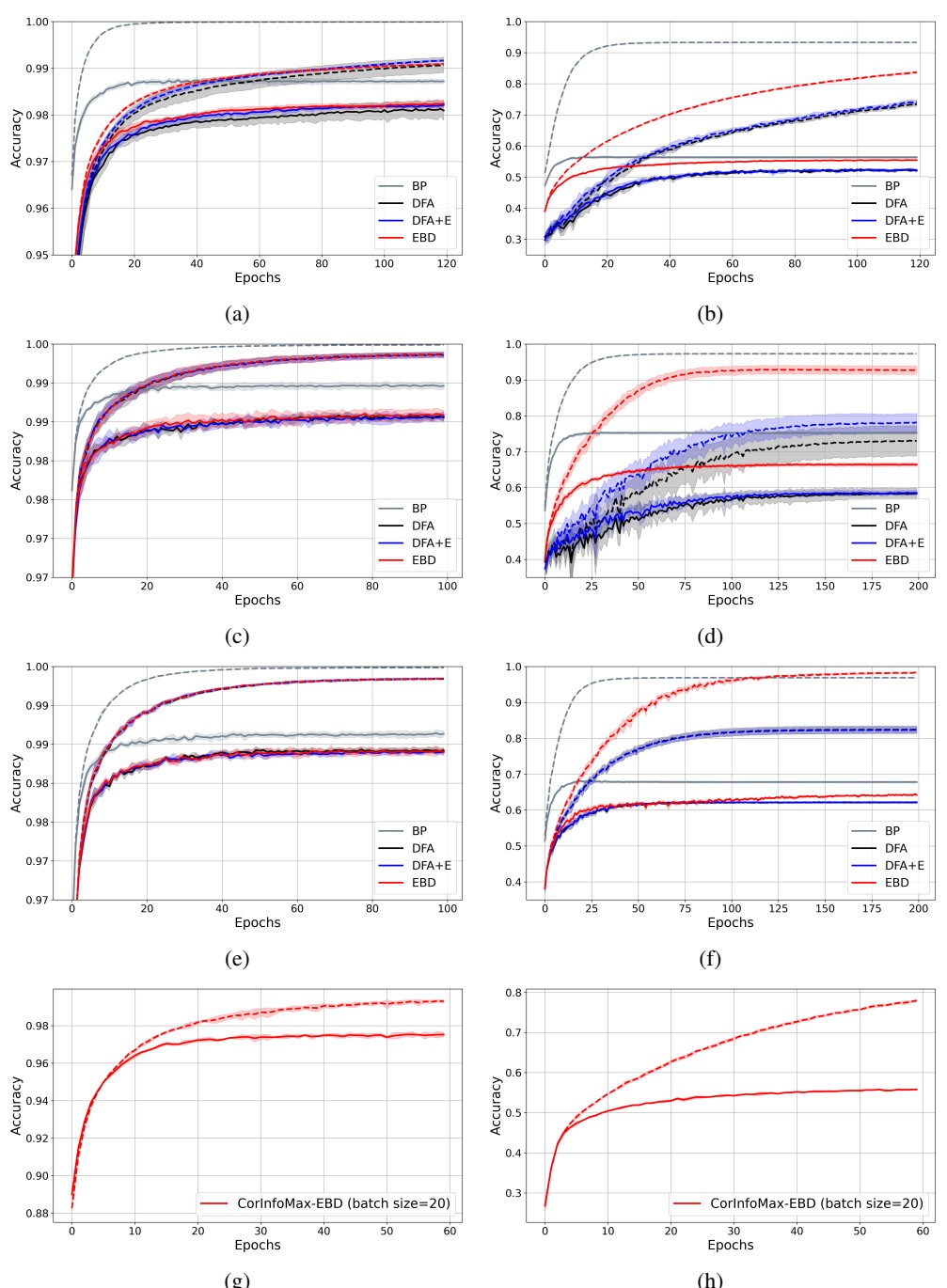

Figure 2: Train and test accuracies plotted as a function of algorithm epochs for training with various update rules (averaged over $n = 5$ runs associated with the corresponding $\pm$ std envelopes) for the (a) MLP on MNIST dataset (b) MLP on CIFAR-10 dataset (c) CNN on MNIST dataset (d) CNN on CIFAR-10 dataset (e) LC on MNIST dataset (f) LC on CIFAR-10 dataset (g) CorInfoMax-EBD on MNIST dataset (h) CorInfoMax-EBD on CIFAR-10 dataset

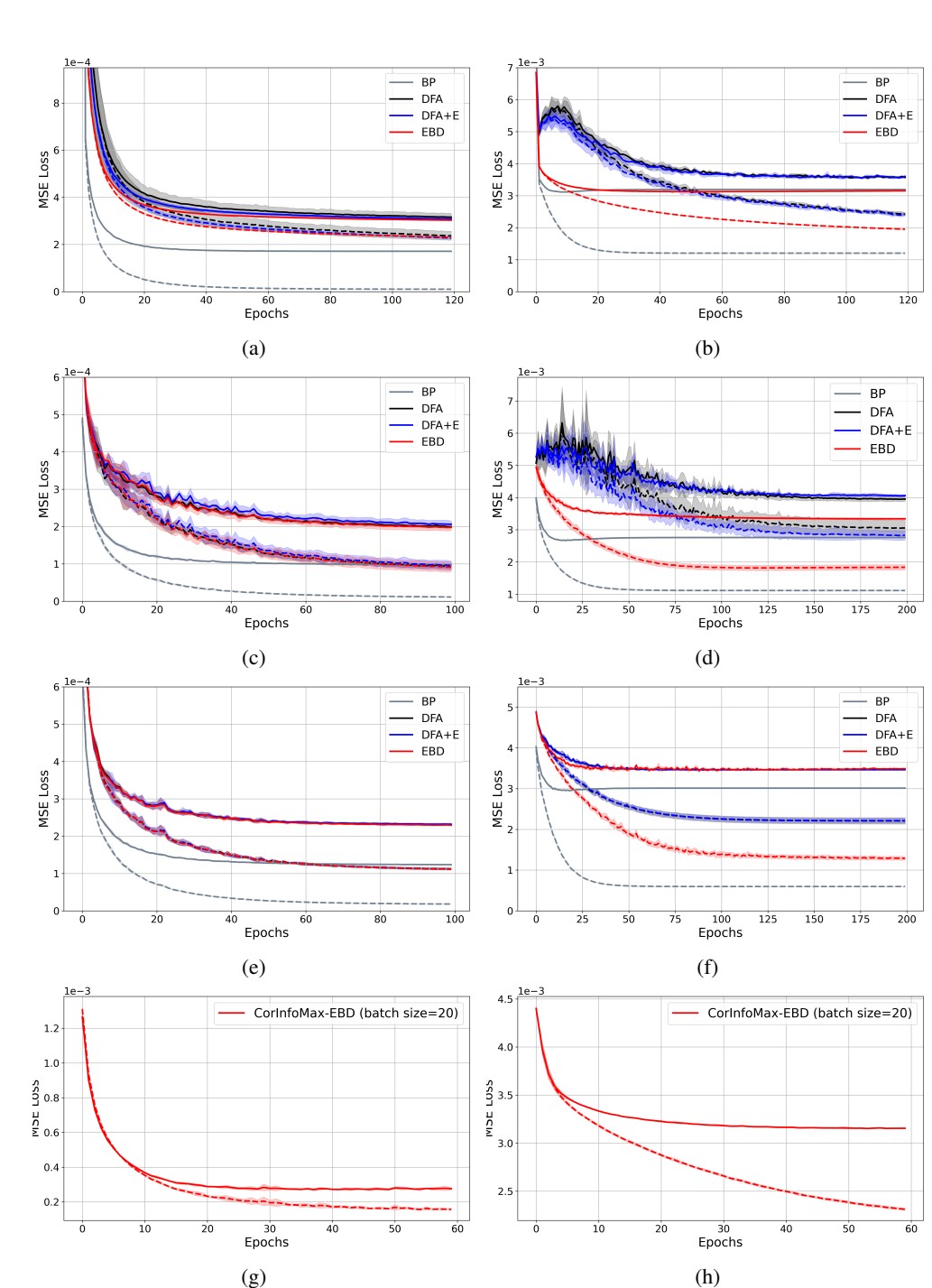

Figure 3: Train and test mean squared errors (MSE) plotted as a function of algorithm epochs for training with various update rules (averaged over $n = 5$ runs associated with the corresponding $\pm$ std envelopes) for the (a) MLP on MNIST dataset (b) MLP on CIFAR-10 dataset (c) CNN on MNIST dataset (d) CNN on CIFAR-10 dataset (e) LC on MNIST dataset (f) LC on CIFAR-10 dataset (g) CorInfoMax-EBD on MNIST dataset (h) CorInfoMax-EBD on CIFAR-10 dataset

### E.3 CorInfoMax-EBD

In this section, we offer additional details regarding the numerical experiments conducted with the CorInfoMax Error Broadcast and Decorrelation (CorInfoMax-EBD) algorithm. Appendix E.3.1 elaborates on the general implementation details. Appendix E.3.2 presents the fundamental learning steps of the algorithm, which are based on the EBD method. Appendices E.3.3 and E.3.4 discuss the initialization of the algorithm's variables and describe the hyperparameters. Finally, Appendix E.3.5 (batch size=20) and E.3.6 (batch size=1) details the specific hyperparameter configurations used in our numerical experiments for the MNIST and CIFAR-10 datasets. In Appendix E.2 we present the accuracy and loss learning curves for the CorInfoMaxEBD, shown in Figures 2.(g)-(h) and Figures 3.(g)-(h), respectively.

### E.3.1 Implementation details

We implemented the CorInfoMax-EBD algorithm based on the repository available at:

`https://github.com/BariscanBozkurt/Supervised-CorInfoMax`

This repository was referenced in Bozkurt et al. (2023). The following modifications were made to the original code:

- **Reduction to a Single Phase:** We simplified the algorithm by reducing it to a single phase. Specifically, we removed the nudge phase, during which the label is coupled to the network dynamics. In this modified version, the network operates solely in the free phase, where the label is decoupled from the network. This change aligns with the removal of time-contrastive updates from the CorInfoMax-EP algorithm.

- **Algorithmic Updates:** We incorporated the updates outlined in Algorithm 3.

- **Hyperparameters:** We maintained the same hyperparameters for the neural dynamics as in the original code. Additionally, new hyperparameters specific to the learning dynamics were introduced, which are detailed in Appendix E.3.4.

In the CorInfoMax-EBD implementation the following loss and regularization functions are used

- EBD loss: $J^{(k)}$,

- Power normalization loss: $J_P^{(k)}$,

- $\ell_2$ weight regularization (weight decay): $J_{\ell_2}^{(k)}$,

- Activation sparsity regularization: $J_{\ell_1}^{(k)} = \|\mathbf{H}^{(k)}\|_1$.

E.3.2 ALGORITHM

The CorInfoMax-EBD algorithm follows the same neural dynamics framework detailed in Bozkurt et al. (2023) for computing neuron activations. Consequently, we only outline the steps specific to the learning process, which distinguishes it from the original CorInfoMax-EP algorithm described in Bozkurt et al. (2023). The full iterative process for updating weights in the CorInfoMax-EBD algorithm is provided in Algorithm 3.

---

**Algorithm 3** CorInfoMax Error Broadcast and Decorrelation (CorInfoMax-EBD) Update for Layer $k$

---

**Require:** Learning rate parameters $\lambda_d, \lambda_E$, $\mu^{(d,k)}[m]$, $\mu^{(f,k)}[m]$, $\mu^{(b,k)}[m]$

**Require:** Previous synaptic weights $\mathbf{W}^{(f,k)}[m-1]$ (forward), $\mathbf{W}^{(b,k)}[m-1]$ (backward), $\mathbf{B}^{(k)}[m-1]$ (lateral)

**Require:** Previous error projection weights $\mathbf{R}_{g(\mathbf{h}^{(k)})\epsilon}[m-1]$

**Require:** Batch size $B$

**Require:** Layer activations $\mathbf{H}^{(k)}[m]$ in (12), the derivative of activations $\mathbf{F}_d^{(k)}$ in (14), in (6), prediction errors $\overset{\leftarrow}{\mathbf{E}}$ and $\overset{\rightarrow}{\mathbf{E}}{}^{(k)}$ in (33-34), lateral weight outputs $\mathbf{Z}^{(k)}$ in (35) computed by CorInfoMax network dynamics described in Bozkurt et al. (2023) (and Appendix D)

**Require:** The nonlinear function of layer activations $\mathbf{G}^{(k)}$ in (5) and the derivative of the nonlinear function of layer activations $\mathbf{G}_d^{(k)}$ in (13)

**Ensure:** Updated weights $\mathbf{W}^{(f,k)}[m]$, $\mathbf{W}^{(b,k)}[m]$, $\mathbf{B}^{(k)}[m]$

**Error projection weight update for layer $k$:**

1  $\hat{\mathbf{R}}_{\mathbf{g}^{(k)}(\mathbf{h}^{(k)})\epsilon}[m] \leftarrow \lambda_d \hat{\mathbf{R}}_{\mathbf{g}^{(k)}(\mathbf{h}^{(k)})\epsilon}[m-1] + \dfrac{1-\lambda_d}{B} \mathbf{G}^{(k)}[m] \mathbf{E}^{(k)}[m]^T$

**Project errors to layer $k$:**

2  $\mathbf{Q}^{(k)}[m] \leftarrow \hat{\mathbf{R}}_{\mathbf{g}^{(k)}(\mathbf{h}^{(k)})\epsilon}[m] \mathbf{E}^{(k)}[m]$

**Find the gradient of the nonlinear function of activations for layer $k$:**

3  $\mathbf{\Phi}^{(k)}[m] = \mathbf{F}_d^{(k)}[m] \odot \mathbf{Q}^{(k)}[m] \odot \mathbf{G}_d^{(k)}[m]$

**Update forward weights for layer $k$:**

4  $\Delta\mathbf{W}_{\text{EBD}}^{(f,k)}[m] \leftarrow -\dfrac{\mu^{(d_f,k)}[m]}{B} \mathbf{\Phi}^{(k)}[m]\mathbf{H}^{(k-1)}[m]^{\top}$

5  $\Delta\mathbf{W}_{\text{Pred}}^{(f,k)}[m] \leftarrow \dfrac{\mu^{(f,k)}[m]}{B} \overset{\rightarrow}{\mathbf{E}}{}^{(k)}[m] \left(\mathbf{H}^{(k-1)}[m]\right)^{\top}$

6  $\mathbf{W}^{(f,k)}[m] \leftarrow \mathbf{W}^{(f,k)}[m-1] + \Delta\mathbf{W}_{\text{EBD}}^{(f,k)}[m] + \Delta\mathbf{W}_{\text{Pred}}^{(f,k)}[m]$

**Update backward weights for layer $k$:**

7  $\Delta\mathbf{W}_{\text{EBD}}^{(b,k)}[m] \leftarrow -\dfrac{\mu^{(d_b,k)}[m]}{B} \mathbf{\Phi}^{(k)}[m] \mathbf{H}^{(k+1)}[m]^{\top}$

8  $\Delta\mathbf{W}_{\text{Pred}}^{(b,k)}[m] \leftarrow \dfrac{\mu^{(b,k)}[m]}{B} \overset{\leftarrow}{\mathbf{E}}{}^{(k)}[m] \mathbf{H}^{(k+1)}[m]^{\top}$

9  $\mathbf{W}^{(b,k)}[m] \leftarrow \mathbf{W}^{(b,k)}[m-1] + \Delta\mathbf{W}_{\text{EBD}}^{(b,k)}[m] + \Delta\mathbf{W}_{\text{Pred}}^{(b,k)}[m]$

**Update Lateral weights for layer $k$:**

10  $\Delta\mathbf{B}_{\text{EBD}}^{(k)}[m] \leftarrow -\dfrac{\mu^{(d_l,k)}[m]}{B} \mathbf{\Phi}^{(k)}[m] \mathbf{H}^{(k)}[m]^{\top}$

11  $\Delta\mathbf{B}_{\text{E}}^{(k)}[m] \leftarrow -\dfrac{\gamma_E}{B} \mathbf{Z}^{(k)}[m]\mathbf{Z}^{(k)}[m]^T$

12  $\mathbf{B}^{(k)}[m] \leftarrow \dfrac{1}{\lambda_E}\mathbf{B}^{(k)}[m] + \Delta\mathbf{B}_{\text{E}}^{(k)}[m] + \Delta\mathbf{B}_{\text{EBD}}^{(k)}[m]$

---

### E.3.3 INITIALIZATION OF ALGORITHM VARIABLES

We initialize the variables $\mathbf{W}^{(f,k)}$, $\mathbf{W}^{(b,k)}$, and $\mathbf{R}_{\mathbf{h}^{(k)}\boldsymbol{\epsilon}}$ using PyTorch's Xavier uniform initialization with its default parameters for the MNIST dataset. For the CIFAR-10 dataset is initialized with gain 0.25. For the lateral weights $\mathbf{B}^{(k)}$, we first generate a random matrix $\mathbf{J}^{(k)}$ of the same dimensions, also using the Xavier uniform distribution, with gain= 1 for the MNIST dataset and with gain= 0.5 for the CIFAR-10 dataset. We then compute $\mathbf{B}^{(k)}[0] = \mathbf{J}^{(k)}\mathbf{J}^{(k)T}$, ensuring that $\mathbf{B}^{(k)}[0]$ is a positive definite symmetric matrix.

### E.3.4 DESCRIPTION OF HYPERPARAMETERS

Table 7 presents a description of the hyperparameters used in the CorInfoMax-EBD implementation.

Table 7: Detailed explanation of hyperparameter notations for the CorInfoMax-EBD algorithm

| Hyperparameter | Description |
| --- | --- |
| $\alpha[m]$ | Learning rate dynamic scaling factor |
| $\alpha_2[m]$ | Learning rate dynamic scaling factor 2 |
| $\mu^{(d_f,k)}$ | Learning rate for decorrelation loss (forward weights) |
| $\mu^{(d_b,k)}$ | Learning rate for decorrelation loss (backward weights) |
| $\mu^{(d_l,k)}$ | Learning rate for decorrelation loss (lateral weights) |
| $\mu^{(f,k)}$ | Learning rate for forward prediction |
| $\mu^{(b,k)}$ | Learning rate for backward prediction |
| $\mu^{(p,k)}$ | Learning rate for power normalization loss |
| $p^{(k)}$ | Target power level |
| $\mu^{(k)}_{f,\ell_1}$ | Learning rate for activation sparsity (forward weights) |
| $\mu^{(k)}_{b,\ell_1}$ | Learning rate for activation sparsity (backward weights) |
| $\mu^{(k)}_{f,w-\ell_2}$ | Forward weight $\ell_2$-regularization coefficent |
| $\mu^{(k)}_{b,w-\ell_2}$ | Backward weight $\ell_2$-regularization coefficent |
| $\lambda_E$ | Layer correlation matrix update forgetting factor |
| $\lambda_d$ | Error-layer activation cross-correlation forgetting factor |
| $m^{(d)}$ | Momentum factor for decorrelation forward weight gradient |
| $B$ | Batch size |

### E.3.5 HYPERPARAMETERS FOR MNIST AND CIFAR-10 DATASETS FOR 20 BATCH SIZE

Table 8 and 9 list the hyperparameters used in the CorInfoMax-EBD numerical experiments for the MNIST and CIFAR-10 datasets with a batch size of 20, where $m$ denotes the iteration index in both.

Table 8: CorInfoMax-EBD hyperparameters: MNIST dataset, 20-batch size.

| Hyperparameter | Value |
|---|---|
| $\alpha[m]$ | $\frac{1}{3 \times 10^{-3} \times \lfloor \frac{m}{10} \rfloor + 1}$ |
| $\alpha_2[m]$ | $\frac{1}{3 \times \lfloor \frac{m}{10} \rfloor + 1}$ |
| $\mu^{(d_f,k)}[m]$ | $\begin{bmatrix} 96 & 60 & 1e5 \end{bmatrix} \alpha[m]$ |
| $\mu^{(d_b,k)}[m]$ | $\begin{bmatrix} 96 & 60 & 1e5 \end{bmatrix} \alpha[m]$ |
| $\mu^{(d_l,k)}[m]$ | $\begin{bmatrix} 0.25 & 0.25 & 0.25 \end{bmatrix} \alpha[m]$ for epoch$= 0$ |
|  | $\begin{bmatrix} 0.5 & 0.5 & 0.5 \end{bmatrix} \alpha[m]$ for epoch$> 0$ |
| $\mu^{(f,k)}[m]$ | $\begin{bmatrix} 0.11e-18 & 0.06e-18 & 0.035e-18 \end{bmatrix} \alpha[m]$ |
| $\mu^{(b,k)}[m]$ | $\begin{bmatrix} 1.125e-18 & 0.375e-18 \end{bmatrix} \alpha[m]$ |
| $\mu^{(p,k)}[m]$ | $\begin{bmatrix} 4.4e-3 & 6e-3 & 3.5e-12 \end{bmatrix} \alpha_2[m]$ |
| $p^{(k)}$ | $\begin{bmatrix} 2.5 & 2.5 & 0.1 \end{bmatrix}$ |
| $\mu_{f,\ell_1}^{(k)}[m]$ | $\begin{bmatrix} 0.008 & 0.135 & 0 \end{bmatrix} \alpha_2[m]$ |
| $\mu_{b,\ell_1}^{(k)}[m]$ | $\begin{bmatrix} 0 & 0.35 & 0.05 \end{bmatrix} \alpha_2[m]$ |
| $\mu_{f,w-\ell_2}^{(k)}[m]$ | $\frac{8e-2}{10^{-2} \times \lfloor \frac{m}{10} \rfloor + 1}$ |
| $\mu_{b,w-\ell_2}^{(k)}[m]$ | $\frac{8e-2}{10^{-2} \times \lfloor \frac{m}{10} \rfloor + 1}$ |
| $\lambda_E$ | $0.999999$ |
| $\lambda_d$ | $0.99999$ |
| $m^{(d)}[m]$ | $0.99 \frac{1}{\lfloor \frac{m}{10} \rfloor + 1} + 0.999(1 - \frac{1}{\lfloor \frac{m}{10} \rfloor + 1})$ |
| $B$ | $20$ |

Table 9: CorInfoMax-EBD hyperparameters: CIFAR-10 dataset, 20-batch size.

| Hyperparameter | Value |
|---|---|
| $\alpha[m]$ | $\frac{1}{3 \times 10^{-3} \times \lfloor \frac{m}{10} \rfloor + 1}$ |
| $\alpha_2[m]$ | $\frac{1}{3 \times \lfloor \frac{m}{10} \rfloor + 1}$ |
| $\mu^{(d_f,k)}[m]$ | $\begin{bmatrix} 80 & 50 & 1e5 \end{bmatrix} \alpha[m]$ for epoch$= 0$ |
|  | $\begin{bmatrix} 320 & 400 & 1e5 \end{bmatrix} \alpha[m]$ for epoch$> 0$. |
| $\mu^{(d_b,k)}[m]$ | $\begin{bmatrix} 0 & 0 & 0 \end{bmatrix} \alpha[m]$ |
| $\mu^{(d_l,k)}[m]$ | $\begin{bmatrix} 0.5 & 0.5 & 0.5 \end{bmatrix} \alpha[m]$ for epoch$= 0$ |
|  | $\begin{bmatrix} 2.0 & 2.0 & 2.0 \end{bmatrix} \alpha[m]$ for epoch$> 0$ |
| $\mu^{(f,k)}[m]$ | $\begin{bmatrix} 0.11e-18 & 0.06e-18 & 0.035e-18 \end{bmatrix} \alpha[m]$ |
| $\mu^{(b,k)}[m]$ | $\begin{bmatrix} 1.125e-18 & 0.375e-18 \end{bmatrix} \alpha[m]$ |
| $\mu^{(p,k)}[m]$ | $\begin{bmatrix} 4.4e-3 & 6e-3 & 3.5e-12 \end{bmatrix} \alpha_2[m]$ |
| $p^{(k)}$ | $\begin{bmatrix} 2.5 & 2.5 & 0.1 \end{bmatrix}$ |
| $\mu_{f,\ell_1}^{(k)}[m]$ | $\begin{bmatrix} 0.008 & 0.135 & 0 \end{bmatrix} \alpha_2[m]$ |
| $\mu_{b,\ell_1}^{(k)}[m]$ | $\begin{bmatrix} 0 & 0.35 & 0.05 \end{bmatrix} \alpha_2[m]$ |
| $\mu_{f,w-\ell_2}^{(k)}[m]$ | $\frac{8e-2}{10^{-2} \times \lfloor \frac{m}{10} \rfloor + 1}$ |
| $\mu_{b,w-\ell_2}^{(k)}[m]$ | $\frac{8e-2}{10^{-2} \times \lfloor \frac{m}{10} \rfloor + 1}$ |
| $\lambda_E$ | $0.999999$ |
| $\lambda_d$ | $0.99999$ |
| $m^{(d)}[m]$ | $0.99 \frac{1}{\lfloor \frac{m}{10} \rfloor + 1} + 0.999(1 - \frac{1}{\lfloor \frac{m}{10} \rfloor + 1})$ |
| $B$ | $20$ |

### E.3.6 HYPERPARAMETERS FOR MNIST AND CIFAR-10 DATASETS FOR 1 BATCH SIZE

Table 10 and 11 list the hyperparameters used in the CorInfoMax-EBD numerical experiments for the MNIST and CIFAR-10 datasets with a batch size of 1, where $m$ denotes the iteration index in both.

Table 10: CorInfoMax-EBD hyperparameters: MNIST dataset, 1-batch size.

| Hyperparameter | Value |
|---|---|
| $\alpha[m]$ | $\frac{1}{3\times10^{-3}\times\lfloor\frac{m}{10}\rfloor+1}$ |
| $\alpha_2[m]$ | $\frac{1}{3\times\lfloor\frac{m}{10}\rfloor+1}$ |
| $\mu^{(d_f,k)}[m]$ | $\begin{bmatrix} 4.8 & 3.0 & 5e3 \end{bmatrix}\alpha[m]$ |
| $\mu^{(d_b,k)}[m]$ | $\begin{bmatrix} 4.8 & 3.0 & 5e3 \end{bmatrix}\alpha[m]$ |
| $\mu^{(d_l,k)}[m]$ | $\begin{bmatrix} 0.0125 & 0.0125 & 0.0125 \end{bmatrix}\alpha[m]$ for epoch$=0$ |
| | $\begin{bmatrix} 0.025 & 0.025 & 0.025 \end{bmatrix}\alpha[m]$ for epoch$>0$ |
| $\mu^{(f,k)}[m]$ | $\begin{bmatrix} 0.11e-18 & 0.06e-18 & 0.035e-18 \end{bmatrix}\alpha[m]$ |
| $\mu^{(b,k)}[m]$ | $\begin{bmatrix} 1.125e-18 & 0.375e-18 \end{bmatrix}\alpha[m]$ |
| $\mu^{(p,k)}[m]$ | $\begin{bmatrix} 2.2e-4 & 3e-4 & 3.5e-12 \end{bmatrix}\alpha_2[m]$ |
| $p^{(k)}$ | $\begin{bmatrix} 2.5 & 2.5 & 0.1 \end{bmatrix}$ |
| $\mu^{(k)}_{f,\ell_1}[m]$ | $\begin{bmatrix} 0.0004 & 0.00675 & 0 \end{bmatrix}\alpha_2[m]$ |
| $\mu^{(k)}_{b,\ell_1}[m]$ | $\begin{bmatrix} 0 & 0.0175 & 0.0025 \end{bmatrix}\alpha_2[m]$ |
| $\mu^{(k)}_{f,w-\ell_2}[m]$ | $\frac{8e-2}{10^{-2}\times\lfloor\frac{m}{10}\rfloor+1}$ |
| $\mu^{(k)}_{b,w-\ell_2}[m]$ | $\frac{8e-2}{10^{-2}\times\lfloor\frac{m}{10}\rfloor+1}$ |
| $\lambda_E$ | 0.99999995 |
| $\lambda_d$ | 0.99999 |
| $m^{(d)}[m]$ | $0.99\frac{1}{\lfloor\frac{m}{10}\rfloor+1}+0.999(1-\frac{1}{\lfloor\frac{m}{10}\rfloor+1})$ |
| $B$ | 1 |

Table 11: CorInfoMax-EBD hyperparameters: CIFAR-10 dataset, 1-batch size.

| Hyperparameter | Value |
|---|---|
| $\alpha[m]$ | $\frac{1}{3\times10^{-3}\times\lfloor\frac{m}{10}\rfloor+1}$ |
| $\alpha_2[m]$ | $\frac{1}{3\times\lfloor\frac{m}{10}\rfloor+1}$ |
| $\mu^{(d_f,k)}[m]$ | $\begin{bmatrix} 4 & 2.5 & 5e3 \end{bmatrix}\alpha[m]$ for epoch$=0$ |
| | $\begin{bmatrix} 16 & 20 & 5e3 \end{bmatrix}\alpha[m]$ for epoch$>0$. |
| $\mu^{(d_b,k)}[m]$ | $\begin{bmatrix} 0 & 0 & 0 \end{bmatrix}\alpha[m]$ |
| $\mu^{(d_l,k)}[m]$ | $\begin{bmatrix} 0.025 & 0.025 & 0.025 \end{bmatrix}\alpha[m]$ for epoch$=0$ |
| | $\begin{bmatrix} 0.1 & 0.1 & 0.1 \end{bmatrix}\alpha[m]$ for epoch$>0$ |
| $\mu^{(f,k)}[m]$ | $\begin{bmatrix} 0.11e-18 & 0.06e-18 & 0.035e-18 \end{bmatrix}\alpha[m]$ |
| $\mu^{(b,k)}[m]$ | $\begin{bmatrix} 1.125e-18 & 0.375e-18 \end{bmatrix}\alpha[m]$ |
| $\mu^{(p,k)}[m]$ | $\begin{bmatrix} 2.2e-4 & 3e-4 & 3.5e-12 \end{bmatrix}\alpha_2[m]$ |
| $p^{(k)}$ | $\begin{bmatrix} 0.125 & 0.125 & 0.005 \end{bmatrix}$ |
| $\mu^{(k)}_{f,\ell_1}[m]$ | $\begin{bmatrix} 0.0004 & 0.000675 & 0 \end{bmatrix}\alpha_2[m]$ |
| $\mu^{(k)}_{b,\ell_1}[m]$ | $\begin{bmatrix} 0 & 0.0175 & 0.0025 \end{bmatrix}\alpha_2[m]$ |
| $\mu^{(k)}_{f,w-\ell_2}[m]$ | $\frac{8e-2}{10^{-2}\times\lfloor\frac{m}{10}\rfloor+1}$ |
| $\mu^{(k)}_{b,w-\ell_2}[m]$ | $\frac{8e-2}{10^{-2}\times\lfloor\frac{m}{10}\rfloor+1}$ |
| $\lambda_E$ | 0.99999995 |
| $\lambda_d$ | 0.99999 |
| $m^{(d)}[m]$ | $0.99\frac{1}{\lfloor\frac{m}{10}\rfloor+1}+0.999(1-\frac{1}{\lfloor\frac{m}{10}\rfloor+1})$ |
| $B$ | 1 |

### E.4 MULTI-LAYER PERCEPTRON

In this section, we provide additional details about the numerical experiments conducted to train Multi-layer Perceptrons (MLPs) using the EBD algorithm (MLP-EBD). Appendix E.4.1 outlines the implementation details of these experiments, while Appendix E.4.2 discusses the initialization of algorithm variables. Information about hyperparameters and their values for the MNIST and CIFAR-10 datasets can be found in Appendices E.4.3-E.4.4. In Appendix E.2 we present the accuracy and loss learning curves for the MLP architecture, shown in Figures 2.(a)-(b) and Figures 3.(a)-(b), respectively.

#### E.4.1 IMPLEMENTATION DETAILS

For the MLP experiments using the proposed EBD approach, we adopted the same network architecture as described in Clark et al. (2021), detailed in Tables 4 and 5.

In the MLP-EBD implementation, the following loss and regularization functions were employed:

- EBD loss: $J^{(k)}$,

- Power normalization loss: $J_P^{(k)}$,

- Entropy objective: $J_E^{(k)}$,

- $\ell_2$ weight regularization (weight decay): $J_{\ell_2}^{(k)}$,

- Activation sparsity regularization: $J_{\ell_1}^{(k)} = \|\mathbf{H}^{(k)}\|_1$.

Additionally, we imposed a weight-sparsity constraint by setting $WS$ percent of the weights to zero during the initialization phase and maintaining these weights at zero throughout training.

#### E.4.2 INITIALIZATION OF ALGORITHM VARIABLES

We use the Pytorch framework's Xavier uniform initialization with gain value $10^{-2}$ on the $\mathbf{R}_{\mathbf{h}^{(k)}\boldsymbol{\epsilon}}$ variables, and Kaiming uniform distribution with gain $0.75$ for synaptic weights $\mathbf{W}^{(k)}$.

#### E.4.3 DESCRIPTION OF HYPERPARAMETERS

Table 12 provides the description of the hyperparameters for the MLP-EBD implementation.

Table 12: Description of the hyperparameter notations for MLP-EBD.

| Hyperparameter | Description |
|---|---|
| $\alpha[m]$ | Learning rate dynamic scaling factor |
| $\alpha_2[m]$ | Learning rate dynamic scaling factor 2 |
| $\mu^{(d,b,k)}$ | Learning rate for (backward projection) decorrelation loss |
| $\mu^{(d,f,k)}$ | Learning rate for (forward projection) decorrelation loss |
| $\mu^{(E,k)}$ | Learning rate for entropy objective |
| $\mu^{(p,k)}$ | Learning rate for power normalization loss |
| $p^{(k)}$ | Target power level |
| $\mu_{\ell_1}^{(k)}$ | Learning rate for activation sparsity |
| $\mu_{w-\ell_2}^{(k)}$ | Weight $\ell_2$-regularization coefficent |
| $\lambda_E$ | Layer autocorrelation matrix update forgetting factor |
| $\lambda_d$ | Error-layer activation cross-correlation forgetting factor |
| $m^{(d)}$ | Momentum factor for decorrelation gradient |
| $B$ | Batch size |
| $WS$ | Weight Sparsity |

### E.4.4 HYPERPARAMETERS FOR MNIST AND CIFAR-10 DATASETS

Table 13 and 14 list the hyperparameters used in the MLP-EBD numerical experiments for the MNIST and CIFAR-10 datasets respectively, where $m$ denotes the iteration index in both.

Table 13: Hyperparameters for MLP-EBD for the MNIST dataset.

| Hyperparameter | Value |
|---|---|
| $\alpha[m]$ | $\frac{1}{1.5 \times \lfloor \frac{m}{10} \rfloor + 1}$ |
| $\alpha_2[m]$ | $\lfloor \frac{m}{10} \rfloor / 3e4 + 1$ |
| $\mu^{(d,b,k)}[m]$ | $18000\alpha[m]\alpha_2[m]$ for $k = 0, 1$ |
| | $20000\alpha[m]\alpha_2[m]$ for $k = 2$ |
| $\mu^{(d,f,k)}[m]$ | $0.005\alpha[m]\alpha_2[m]$ for $k = 0, 1$ |
| $\mu^{(E,k)}[m]$ | $\begin{bmatrix} 2.5e-4 & 1.5e-3 & 0 \end{bmatrix}\alpha[m]$ |
| $\mu^{(p,k)}[m]$ | $\begin{bmatrix} 4e-3 & 6e-3 & 1e-10 \end{bmatrix}\alpha[m]$ |
| $p^{(k)}[m]$ | $\begin{bmatrix} 2.5e-1 & 2.5e-1 & 0.1 \end{bmatrix}\alpha[m]$ |
| $\mu_{\ell_1}^{(k)}$ | $\begin{bmatrix} 8e-1 & 3e-1 & 0 \end{bmatrix}\alpha[m]$ |
| $\mu_{w-\ell_2}^{(k)}$ | $1.6e-4\alpha[m]$ for all layers |
| $\lambda_E$ | 0.99999 |
| $\lambda_d$ | 0.999999 |
| $m^{(d)}$ | 0.9999 for all layers |
| $B$ | 20 |
| $WS$ | 55 |

Table 14: Hyperparameters for MLP-EBD for the CIFAR-10 dataset.

| Hyperparameter | Value |
|---|---|
| $\alpha[m]$ | $\frac{1}{1.5 \times \lfloor \frac{m}{10} \rfloor + 1}$ |
| $\alpha_2[m]$ | $\lfloor \frac{m}{10} \rfloor / 3e4 + 1$ |
| $\mu^{(d,b,k)}[m]$ | $\begin{bmatrix} 4000 & 2000 & 2000 & 3500 \end{bmatrix}\alpha[m]\alpha_2[m]$ |
| $\mu^{(d,f,k)}[m]$ | $0.005\alpha[m]\alpha_2[m]$ for $k = 0, 1$ |
| $\mu^{(E,k)}[m]$ | $\begin{bmatrix} 2.5e-4 & 1.5e-3 & 1.5e-3 & 0 \end{bmatrix}\alpha[m]$ |
| $\mu^{(p,k)}[m]$ | $\begin{bmatrix} 4e-3 & 6e-3 & 6e-3 & 1e-10 \end{bmatrix}\alpha[m]$ |
| $p^{(k)}[m]$ | $\begin{bmatrix} 2.5e-1 & 2.5e-1 & 2.5e-1 & 0.1 \end{bmatrix}\alpha[m]$ |
| $\mu_{\ell_1}^{(k)}$ | $\begin{bmatrix} 8e-1 & 3e-1 & 3e-1 & 0 \end{bmatrix}\alpha[m]$ |
| $\mu_{w-\ell_2}^{(k)}$ | $1.6e-4\alpha[m]$ for all layers |
| $\lambda_E$ | 0.99999 |
| $\lambda_d$ | 0.999999 |
| $m^{(d)}$ | 0.9999 for all layers |
| $B$ | 20 |
| $WS$ | 40 |

## E.5 CONVOLUTIONAL NEURAL NETWORK

In this section, we offer additional details regarding the numerical experiments for training Convolutional Neural Neural Networks (CNNs) using EBD algorithm (CNN-EBD). Section E.5.1 provides information about implemetation details. Appendices E.5.2 and E.5.3 discuss the initialization of the algorithm's variables and describe the hyperparameters. Finally, Appendix E.5.4 detail the specific hyperparameter configurations used in our numerical experiments for the MNIST and CIFAR-10 datasets. In Appendix E.2 we present the accuracy and loss learning curves for the CNN, shown in Figures 2.(c)-(d) and Figures 3.(c)-(d), respectively.

### E.5.1 IMPLEMENTATION DETAILS

The architectures we utilized for the CNN networks can be found in tables 4 and 5 respectively for the MNIST and CIFAR10 datasets. In the training, we used the Adam optimizer with hyperparameters $\beta_1 = 0.9$, $\beta_2 = 0.999$, and $\epsilon = 10^{-8}$ (Kingma & Ba, 2015). Also, the model biases are not utilized. In the CNN-EBD implementation the following loss and regularization functions as detailed in section C.1 are used:

- EBD loss: $J^{(k)}$,

- Entropy objective: $J_E^{(k)}$,

- Activation sparsity regularization: $J_{\ell_1}^{(k)}$.

### E.5.2 INITIALIZATION OF ALGORITHM VARIABLES

We use the Kaiming normal initialization for the weights, with a common standard deviation scaling parameter $\sigma_{\mathbf{W}}$, on both the linear and convolutional layers. Furthermore, the estimated cross-correlation variable $\mathbf{R}_{\mathbf{h}^{(k)}\epsilon}$ (linear layers) and $\mathbf{R}_{\mathbf{g}^{(k)}(\mathbf{H}^{(k,p)})\epsilon}$ (convolutional layers) are initialized with zero mean normal distributions with standard deviations $\sigma_{\mathbf{R}_{lin}}$ and $\sigma_{\mathbf{R}_{conv}}$ respectively.

### E.5.3 DESCRIPTION OF HYPERPARAMETERS

Table 15 describes the notation for the hyperparameters used to train CNNs using the Error Broadcast and Decorrelation (EBD) approach.

Table 15: Description of the hyperparameter notations for CNN-EBD.

| Hyperparameter | Description |
| --- | --- |
| $\alpha[i]$ | Learning rate dynamic scaling factor where $i$ is the epoch index |
| $\mu^{(d,b,k)}$ | Learning rate for (backward projection) decorrelation loss |
| $\mu^{(E,k)}$ | Learning rate for entropy objective |
| $\mu_{\ell_1}^{(k)}$ | Learning rate for activation sparsity |
| $\sigma_{\mathbf{W}}$ | Standard deviation of the weight initialization. |
| $\sigma_{\mathbf{R}_{lin}}$ | Std. dev. of $\mathbf{R}_{\mathbf{h}^{(k)}\epsilon}$ initialization in linear layers |
| $\sigma_{\mathbf{R}_{conv}}$ | Std. dev. of $\mathbf{R}_{\mathbf{g}^{(k)}(\mathbf{H}^{(k,p)})\epsilon}$ initialization in convolutional layers |
| $\sigma_{\mathbf{R}_{local}}$ | Gain parameter for $\mathbf{R}_{\mathbf{g}^{(k)}(\mathbf{H}^{(k,p)})\epsilon}$ initialization in locally connected layers |
| $\lambda_E$ | Layer autocorrelation matrix update forgetting factor |
| $\lambda_d$ | Error-layer activation cross-correlation forgetting factor |
| $\lambda_R$ | Convergence parameter for $\lambda$ as in Equations (36), (37) |
| $\epsilon_L$ | Entropy objective epsilon parameter for linear layers |
| $\epsilon$ | Entropy objective epsilon parameter for conv. or locally con. layers |
| $\beta$ | Adam Optimizer weight decay parameter |
| $B$ | Batch size |

We also introduce a convergence parameter $\lambda_R$ which increases the estimation parameter for the decorrelation loss $\lambda_d$, together with the estimation parameter for the layer entropy objective $\lambda_E$, to converge to 1 as the training proceeds with the following Equations (36), (362) where $i$ is the epoch index:

$$\lambda_d^{(i+1)} = \lambda_d^{(i)} + \lambda_R \cdot \left(1 - \lambda_d^{(i)}\right), i \geq 0. \tag{36}$$

$$\lambda_E^{(i+1)} = \lambda_E^{(i)} + \lambda_R \cdot \left(1 - \lambda_E^{(i)}\right), i \geq 0. \tag{37}$$

### E.5.4   HYPERPARAMETERS FOR MNIST AND CIFAR-10 DATASETS

Table 16, lists the hyperparameters as defined in Table 15, used in the CNN-EBD training experiments.

Table 16: Hyperparameters for CNN-EBD for both the MNIST and CIFAR-10 datasets, where $i$ denotes the epoch index.

| Hyperparameter | MNIST | CIFAR-10 |
|---|---|---|
| $\alpha[i]$ | $10^{-4} \cdot 0.97^{-i}$ | $10^{-4} \cdot 0.97^{-i}$ |
| $\mu^{(d,b,k)[i]}$ | $0.1\alpha[i]$ for $k = 0, 1, 2, 3$ | $0.1\alpha[i]$ for $k = 0, 1, 2, 3$ |
| | $10\alpha[i]$ for $k = 4$ | $10\alpha[i]$ for $k = 4$ |
| $\mu^{(E,k)}[i]$ | $\begin{bmatrix} 1 & 1 & 1 & 10 & 0 \end{bmatrix} 10^{-7}\alpha[i]$ | $\begin{bmatrix} 1 & 1 & 1 & 1 & 1 \end{bmatrix} 10^{-6}\alpha[i]$ |
| $\mu_{\ell_1}^{(k)}[i]$ | $\begin{bmatrix} 1 & 1 & 1 & 10 & 0 \end{bmatrix} 10^{-11}\alpha[i]$ | $\begin{bmatrix} 1 & 1 & 1 & 10^2 & 0 \end{bmatrix} 10^{-10}\alpha[i]$ |
| $\sigma_{\mathbf{W}}$ | $\sqrt{\frac{1}{6}}$ | $\sqrt{\frac{1}{6}}$ |
| $\sigma_{\mathbf{R}_{lin}}$ | $1e-2$ | $1e-2$ |
| $\sigma_{\mathbf{R}_{conv}}$ | $1e-2$ | $1e-2$ |
| $\lambda_d$ | $0.99999$ | $0.99999$ |
| $\lambda_E$ | $0.99999$ | $0.99999$ |
| $\lambda_R$ | $2e-2$ | $2e-2$ |
| $\beta$ | $1e-8$ | $1e-5$ |
| $\epsilon_L$ | $1e-8$ | $1e-8$ |
| $\epsilon$ | $1e-5$ | $1e-5$ |
| $B$ | 16 | 16 |

### E.6 LOCALLY CONNECTED NETWORK

In this section, we offer additional details regarding the numerical experiments for the training of Locally Connected Networks (LCs) using EBD algorithm (LC-EBD). Appendix E.6.1 provides information about implemetation details. Appendices E.6.2 and E.6.3 discuss the initialization of the algorithm's variables and describe the hyperparameters. Finally, Appendix E.6.4 detail the specific hyperparameter configurations used in our numerical experiments for the MNIST and CIFAR-10 datasets. In Appendix E.2 we present the accuracy and loss learning curves for the LCs, shown in Figures 2.(e)-(f) and Figures 3.(e)-(f), respectively.

#### E.6.1 IMPLEMENTATION DETAILS

The training procedure mirrors the CNN approach described in Section E.5.1 for CNNs. In the LC-EBD implementation, the loss and regularization functions detailed in section C.2 are used:

- EBD loss: $J^{(k)}$,

- Entropy objective: $J_E^{(k)}$,

- Activation sparsity regularization: $J_{\ell_1}^{(k)}$.

#### E.6.2 INITIALIZATION OF ALGORITHM VARIABLES

We use the Kaiming uniform initialization for the weights, with a common standard deviation scaling parameter $\sigma_{\mathbf{W}}$, on both the linear and locally connected layers. The estimated cross-correlation variable $\mathbf{R}_{\mathbf{h}^{(k)}\boldsymbol{\epsilon}}$ (linear layers) is initialized with a normal distribution with zero mean and standard deviation $\sigma_{\mathbf{R}_{lin}}$. Also, the parameter $\mathbf{R}_{\mathbf{g}^{(k)}(\mathbf{H}^{(k,p)})\boldsymbol{\epsilon}}$ (locally connected layers) is initialized with Pytorch framework's Xavier uniform initialization with the gain parameter equal to $\sigma_{\mathbf{R}_{local}}$.

#### E.6.3 DESCRIPTION OF HYPERPARAMETERS

Table 15 in the CNN section, again describes the notation for the hyperparameters used to train LCs using the Error Broadcast and Decorrelation (EBD) approach. The convergence parameter $\lambda_R$ introduced in equations (36) and (36) is used as well.

#### E.6.4 HYPERPARAMETERS FOR MNIST AND CIFAR-10 DATASETS

Table 17, lists the hyperparameters as defined in Table 15, used in the LC-EBD training experiments.

Table 17: Hyperparameters for LC-EBD for both the MNIST and CIFAR-10 datasets, where $i$ denotes the epoch index.

| Hyperparameter | MNIST | CIFAR-10 |
|---|---|---|
| $\alpha[i]$ | $10^{-4} \cdot 0.96^{-i}$ | $10^{-4} \cdot 0.98^{-i}$ |
| $\mu^{(d,b,k)}[i]$ | $0.1\alpha[i]$ for $k = 0, 1, 2, 3$ | $0.5\alpha[i]$ for $k = 0, 1, 2, 3$ |
| | $10\alpha[i]$ for $k = 4$ | $5\alpha[i]$ for $k = 4$ |
| $\mu^{(E,k)}[i]$ | $\begin{bmatrix} 1 & 1 & 1 & 10^2 & 0 \end{bmatrix} 10^{-9}\alpha[i]$ | $\begin{bmatrix} 1 & 1 & 1 & 10 & 10^3 \end{bmatrix} 10^{-11}\alpha[i]$ |
| $\mu_{\ell_1}^{(k)}[i]$ | $\begin{bmatrix} 1 & 1 & 1 & 10 & 0 \end{bmatrix} 10^{-11}\alpha[i]$ | $\begin{bmatrix} 1 & 1 & 1 & 10 & 0 \end{bmatrix} 10^{-13}\alpha[i]$ |
| $\sigma_{\mathbf{W}}$ | $\sqrt{\frac{1}{6}}$ | $\sqrt{\frac{1}{6}}$ |
| $\sigma_{\mathbf{R}_{lin}}$ | 1 | $1e-3$ |
| $\sigma_{\mathbf{R}_{local}}$ | 1 | $1e-1$ |
| $\lambda_d$ | 0.99999 | 0.99999 |
| $\lambda_E$ | 0.99999 | 0.99999 |
| $\lambda_R$ | $3e-2$ | $3e-2$ |
| $\beta$ | $1e-8$ | $1e-6$ |
| $\epsilon_L$ | $1e-8$ | $1e-8$ |
| $\epsilon$ | $1e-5$ | $1e-5$ |
| $B$ | 16 | 16 |

### E.7 IMPLEMENTATION DETAILS FOR DIRECT FEEDBACK ALIGNMENT AND BACKPROPAGATION BASED TRAINING

This section presents further details on the numerical experiments comparing Direct Feedback Alignment (DFA) and Backpropagation (BP) methods, conducted under the same training conditions and number of epochs as those used for our proposed EBD algorithm. The results of these experiments are provided in Tables 1 and 2. We also include the DFA+E method, which extends DFA by incorporating correlative entropy regularization similar to the EBD. Note that, when the update on the $\mathbf{R}_{\mathbf{h}^{(k)}\epsilon}$ is fixed to its initialization, the EBD algorithm reduces to standard DFA.

For BP-based models trained on MNIST and CIFAR-10, we used the Adam optimizer with hyperparameters $\beta_1 = 0.9$, $\beta_2 = 0.999$, and $\epsilon = 10^{-8}$ (Kingma & Ba, 2015). For DFA and DFA+E models, we again used the Adam optimizer for CNN and LC models, while MLP models were trained using SGD with momentum.

In Tables 18 and 19, we detail the hyperparameters for models trained with BP, DFA, and DFA+E update rules on MNIST and CIFAR-10 respectively. Some of the learning rate and the learning rate decay values or methodologies are linked to the tables corresponding to the hyperparameter details of its EBD counterpart, where the same method is also utilized for its DFA or DFA+E counterpart. Unlinked values denote a constant value applied to each layer, or the step decay multiplier applied per epoch. Additionally, sparsity inducing losses are not utilized for BP, DFA and DFA+E models.

| Model | Method | Learning Rate ($\mu^{(d,b,k)}$) | L2 Reg. Coef. | LR Decay | Epochs |
|-------|--------|---------------------------------|---------------|----------|--------|
|       | BP     | $5e-5$                          | $1e-5$        | 0.96     | 120    |
| **MLP** | DFA  | Table-13                        | Table-13      | Table-13 | 120    |
|       | DFA+E  | Table-13                        | Table-13      | Table-13 | 120    |
|       | BP     | $5e-5$                          | $1e-8$        | 0.97     | 100    |
| **CNN** | DFA  | Table-16                        | $1e-8$        | 0.97     | 100    |
|       | DFA+E  | Table-16                        | $1e-8$        | 0.97     | 100    |
|       | BP     | $5e-5$                          | $1e-8$        | 0.96     | 100    |
| **LC** | DFA   | Table-17                        | $1e-8$        | 0.96     | 100    |
|       | DFA+E  | Table-17                        | $1e-8$        | 0.96     | 100    |

Table 18: Hyperparameter details for models trained on the MNIST dataset, including learning rate, L2 regularization coefficient, learning rate decay, and number of epochs for MLP, CNN, and LC models using BP, DFA, and DFA+E methods.

| Model | Method | Learning Rate ($\mu^{(d,b,k)}$) | L2 Reg. Coef. | LR Decay | Epochs |
|-------|--------|---------------------------------|---------------|----------|--------|
|       | BP     | $5e-5$                          | $1e-5$        | 0.85     | 120    |
| **MLP** | DFA  | Table-14                        | 0             | Table-14 | 120    |
|       | DFA+E  | Table-14                        | 0             | Table-14 | 120    |
|       | BP     | $5e-5$                          | $1e-5$        | 0.92     | 200    |
| **CNN** | DFA  | Table-16                        | $1e-5$        | 0.97     | 200    |
|       | DFA+E  | Table-16                        | $1e-5$        | 0.97     | 200    |
|       | BP     | $1e-4$                          | $1e-6$        | 0.90     | 200    |
| **LC** | DFA   | Table-17                        | $1e-6$        | 0.96     | 200    |
|       | DFA+E  | Table-17                        | $1e-6$        | 0.96     | 200    |

Table 19: Hyperparameter details for models trained on the CIFAR-10 dataset, including learning rate, L2 regularization coefficient, learning rate decay, and number of epochs for MLP, CNN, and LC models using BP, DFA, and DFA+E methods.

### E.8 RUNTIME COMPARISON FOR THE UPDATE RULES

In this section, we present the relative average runtimes from the simulations, normalized to BP for the MNIST and CIFAR-10 models in Tables 20 and 21 respectively, for the models that we implemented and demonstrated their performance in Tables 1 and 2.

The results show that entropy regularization in both EBD and DFA+E more than doubles the average runtime. However, these runtimes could be significantly improved by optimizing the implementation of the entropy gradient terms, specifically by avoiding repeated matrix inverse calculations. A more efficient approach would involve directly updating the inverses of the correlation matrices instead of recalculating both the matrices and their inverses at each step. This strategy aligns with the CorInfoMax-(EP/EBD) network structure. Nonetheless, we chose not to pursue this optimization, as CorInfoMax networks already employ it effectively.

The efficiency of the DFA, DFA+E, and EBD methods can be further enhanced through low-level optimizations and improved implementations.

Table 20: Average Runtimes in MNIST (relative to BP)

| Model | DFA | DFA+E | BP | EBD |
|-------|------|-------|-----|------|
| MLP   | 3.40 | 7.68  | 1.0 | 8.06 |
| CNN   | 1.68 | 2.95  | 1.0 | 3.85 |
| LC    | 1.61 | 3.57  | 1.0 | 3.54 |

Table 21: Average Runtimes in CIFAR-10 (relative to BP)

| Model | DFA | DFA+E | BP | EBD |
|-------|------|-------|-----|------|
| MLP   | 2.85 | 6.94  | 1.0 | 7.61 |
| CNN   | 2.10 | 3.24  | 1.0 | 4.11 |
| LC    | 1.35 | 2.01  | 1.0 | 2.41 |

### E.9 ON THE SCALING OF THE ORTHOGONALITY CONDITIONS

The orthogonality principle in our method is defined as the uncorrelatedness of a given hidden layer neuron's activation with all components of the output error. Specifically, for the $i^{th}$ neuron of layer-$k$ and the $j^{\text{th}}$ component of the output error, $\epsilon_j$, the orthogonality condition is expressed as,

$$R_{h_i^{(k)}\epsilon_j} = E(h_i^{(k)}\epsilon_j) = 0, \quad j = 1, \dots m, \quad i = 1, \dots n \tag{38}$$

where $m$ is the number of output components, and $n$ is the size of activations for layer $k$.

Based on Equation 38 above, for each hidden layer, there are $m$ x $n$ orhogonality constraints. Therefore, even if the hidden layer dimensions and/or the network depth increase, the total number of constraints also increases, making our system less underdetermined. We use constraints for different neurons separately to adjust their corresponding weight/bias parameters. In other words, the constraints in (Eq. A) are used to update the $i^{\text{th}}$ row of the $j^{\text{th}}$ column of $\mathbf{W}^{(k)}$, i.e. $\mathbf{W}_{i,j}^{(k)}$ and the bias compoent $b_i$.

Note that, more generality of the orthogonality condition for nonlinear estimators offers potential to increase the number of constraints per hidden layer neuron: We can increase the number of the orthogonality conditions per neuron even further by considering the fact that uncorrelatedness requirement is for any function $g$ of hidden layer neuron activations, i.e.,

$$R_{g(h_i^{(k)})\epsilon_j} = E(g(h_i^{(k)})\epsilon_j) = 0, \quad j = 1, \dots m \quad i = 1, \dots n$$

Therefore, the number of uncorrelatedness (orthgonality) constraints per hidden layer/output neuron can be increased by introducing multiple $g$ functions. However, in our numerical experiments we haven't pursued this path.

Although the orthogonality conditions scale with the increasing parameter size, the total number of parameters in the network is in general larger than the number of decorrelation conditions. This results in fewer constraints than parameters, leading to an overparameterized system, where a unique optimal estimator cannot be determined solely based on these conditions. But our results show that the learning rule converges effectively within practical timeframes. Particularly in the case of using Locally Connected Networks (LC), which are highly overparameterized, the improved performance and generalization observed strongly validate the practicality of our approach to successfully train in the overparametrized regime.

Importantly, this issue of overparameterization also exists in standard backpropagation, where the number of parameters often exceeds the number of training samples, leading to an overparameterized and underdetermined system. In both cases, this overparameterization does not hinder learning; rather, it is a fundamental characteristic of deep learning. Research has demonstrated that the implicit bias in gradient descent introduces a regularization effect, steering the optimization process toward solutions that generalize well to unseen data (Soudry et al., 2018).

Similarly, in our method, while the number of orthogonality constraints is smaller than the total number of network parameters, the system is guided by the statistical properties of the error and activations. While we cannot claim to fully characterize the implicit regularization effect in our method, we suggest that these statistical constraints play a role similar to the implicit regularization observed in regular backpropagation. This helps ensure that the learned parameters are not arbitrary but are shaped by the decorrelation principles inherent to our framework, contributing to the model's generalization capabilities. We believe that investigating the inherent implicit bias in Error Broadcast and Decorrelation (EBD) opens the door to further understanding how this framework naturally regularizes the learning process.

To further adress limited-data problems, our method incorporates several regularization techniques: entropy regularization (encourages the network to utilize the full feature space by spreading activations), sparsity regularization (enforces sparse activations to reduce redundancy), weight decay (prevents overfitting by penalizing large weights). These regularizers supplement the orthogonality-based learning rule, particularly in the limited-data regime, improving generalization and stability.

# F    CALCULATION OF THE CORRELATION BETWEEN LAYER ACTIVATIONS AND OUTPUT ERROR.

Figure 1c illustrates the decrease in the average absolute correlation between hidden activations and output error during backpropagation, using a Multi-layer Perceptron (MLP) model with the architecture outlined in Table 5, on the CIFAR-10 dataset. Additionally, the figure 1c shows the correlation throughout training on the MNIST dataset, employing the MLP model detailed in Table 4. Details for the MSE based training and the Cross-Entropy based training are explained in Appendices F.1 and F.2 respectively.

## F.1    CORRELATION IN MEAN SQUARED ERROR (MSE) CRITERION-BASED TRAINING

The MLP models are trained using the Stochastic Gradient Descent (SGD) optimizer with a small learning rate of $10^{-4}$ and the MSE criterion. In both plots, the initial value represents the correlation before training begins. The reduction in correlation observed during training provides insight into the core principle of the EBD algorithm.

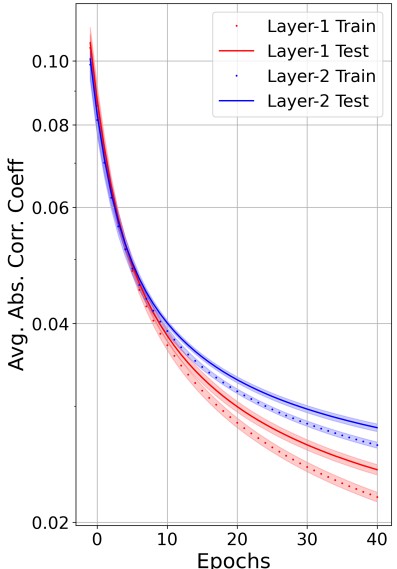

Figure 4: The evolution of the average absolute correlation between layer activations and the error signal during backpropagation training of an MLP with two hidden layers (using the MSE criterion) on the MNIST dataset, showing the correlation decrease over epochs, on both the training and test sets.

To compute these correlations, we apply a batched version of Welford's algorithm (Chan et al., 1982), which efficiently calculates the Pearson correlation coefficient between hidden activations and errors in a memory-efficient way by using streaming statistics.

Welford's algorithm works by accumulating the necessary statistics (e.g., sums and sums of squares) across batches of data and finalizing the correlation computation only after all data has been processed, avoiding the need to store all hidden activations simultaneously.

Given the hidden activations $\mathbf{h}^{(k)} \in \mathbb{R}^{b \times N^{(k)}}$, where $b$ is the batch size and $N^{(k)}$ is the number of hidden units, and the errors $\boldsymbol{\epsilon} \in \mathbb{R}^{b \times k}$, where $k$ is the number of output dimensions (e.g., classes); the goal is to compute the Pearson correlation coefficient between activations $h_i$ for each hidden unit $i$ and the corresponding error values across all samples as:

$$\rho^{(k)} = \frac{\mathrm{Cov}(\mathbf{h}^{(k)}, \boldsymbol{\epsilon})}{\sqrt{\sigma^2_{\mathbf{h}^{(k)}}} \sqrt{\sigma^2_{\boldsymbol{\epsilon}}} + \epsilon}$$

where $\epsilon$ is a small constant for numerical stability. Finally, we compute the average of the absolute values of the correlation coefficients for each hidden layer $k$ to generate the corresponding plots.

### F.2 CORRELATION IN CROSS-ENTROPY CRITERION-BASED TRAINING

Although the orthogonality property is specifically associated with the MSE loss, we also explored the dynamics of cross-correlation between layer activations and output errors when cross-entropy is used as the training criterion.

With the same experimental setup as described in Appendix F.1, but replacing the MSE loss with cross-entropy, we obtained the correlation evolution curves shown in Figure 5a for CIFAR-10 and in Figure 5b for MNIST dataset. Notably, the correlation between layer activations and output errors still decreases over epochs, despite the change in the loss function.

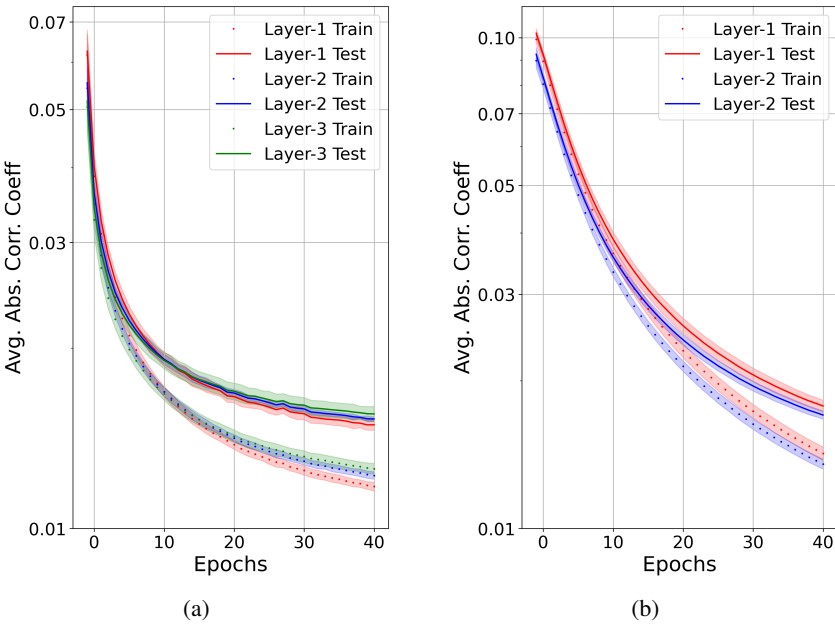

(a)                                              (b)

Figure 5: Evolution of the average absolute correlation between layer activations and output errors during backpropagation training of an MLP with three hidden layers, trained using cross-entropy loss. (a) CIFAR-10 dataset and (b) MNIST dataset. Despite the use of cross-entropy, correlation decreases similarly to the MSE criterion.

