# OpenReview forum: "Error Broadcast and Decorrelation as a Potential Artificial and Natural Learning Mechanism"
_ICLR.cc/2025/Conference — Submitted to ICLR 2025_

### Official Review · Reviewer_VBft · 2024-10-21

**Soundness:** 2
**Presentation:** 2
**Contribution:** 2
**Rating:** 5
**Confidence:** 4

**Summary:**

This work proposes a novel optimization method called the “Error Broadcast and Decorrelation” (EBD) algorithm. This algorithm attempts to obtain the optimal minimum mean square error (MMSE) estimator by meeting its core criteria: that at optimum, there is zero correlation between an error signal and non-linearly transformed encodings of data. This enables a new perspective on error broadcasting which attempts to capture correlations between errors and neural network layer activations and uses this correlation (covariance) structure to propagate and thereafter minimise correlation. This effectively results in a decorrelation between error and layer-wise activations once converged. This method is combined with a number of additions to stabilize network activity norms and encourage activation entropy within network layers, as well as being integrated into the CorInfoMax framework. This method is finally tested by training of multilayer perceptrons and convolutional networks with the MNIST and CIFAR10 tasks.

**Strengths:**

- This work contributes a new perspective for measurement of a final optimum of network training based upon the MMSE estimator in a principled manner based upon the orthogonality of error and layer-activations.
- The paper describes this method and its drawbacks within the methods section at length and covers a set of failure modes and extensions.
- Detailed descriptions of the mathematical steps and hyperparameters of models tested are given.

**Weaknesses:**

- Of major and significant concern are the claims of this paper in comparison to existing work by Clark et al. 2021. Specifically, the results of this paper are presented in direct comparison to trained models in earlier work (Clark et al.) while claiming outperformance. However, this paper integrates a number of elements to the training scheme that are not present in the original comparison work. For example, the code and tables of this paper suggest that among other additions this work makes use of a learning rate scheduler, as well as potentially using many more epochs of training (unclear). In comparison, the original work by Clark et al. has no learning rate scheduler and far fewer training hyperparameters in general. This suggests that the comparison is entirely inappropriate. To provide a genuine comparison, I would encourage the authors to carry out a reimplementation and rigorous test against existing methods (at least against BP) in which the same degree of parameter searching/sweeping is carried out for all methods compared. Otherwise comparison is uninformative at best, and misleading at worst. For these reasons, the results in Tables 1 and 2 cannot in their current form be trusted to provide understanding of the relative efficacy of training methods.
- This paper claims to provide a novel biologically plausible learning mechanism (even in the title), however to make this method work, a power normalizing and entropy encouraging mechanism is added to network dynamics. It is not discussed whether these are reasonable mechanisms within a biologically plausible context.
- The current set of simulations results are sufficiently limited that it is not clear whether this method would scale. In particular, biologically-plausible rules can succeed at tasks such as MNIST or  CIFAR-10 level but fail completely at large scale (see Bartunov et al. 2018, Neurips). Currently, there are no explanations of how well this method might do when scaled to harder datasets, or even how well it scales when network width or depth is modified. Without measures of performance across network scale and task complexity, it is not possible to know whether this method’s performance is robust to depth/task-complexity.
- The description of this work’s method is comprehensive but requires a reader to go back and forth to understand it well. For example, the added extensions to EBD, which are used during training, are described with some distance after the main method (in Section 4) making it difficult to understand all moving parts of the simulation in a single read. Furthermore, the paper in general is too heavy on the methods aspects leaving zero room for interpretation of results and discussion. A refactoring of the paper in these respects would greatly help its readability and contribution as well as enabling a more complete discussion on the implications of the work.

**Questions:**

The weaknesses section above contains most of my concerns which should be addressed for an increased score. Here a few additional questions are posed.
- How might the mechanisms for power normalization and layer entropy be a plausible addition to biological neural networks?
- Can this framework be extended beyond the MSE loss case? In practice, loss functions in the deep neural network literature are often very different than an MSE loss. In Appendix E.2, correlation curves are shown for the Categorical Cross Entropy loss, however it is unclear if this was used in practice to train networks. Clarity would be appreciated.
- The computational complexity of the method and the proposed additional learning of correlation structures is not much discussed. How much might such a method cost in this regard?

---

> ### Author Response · Authors · 2024-11-18
> **Response to Reviewer VBft's Comments**
>
> >strengths: contributes a new perspective, covers a set of failure modes and extensions.
>
> We would like to thank the reviewer for the possitive assesments about the novel perspective and the contributions of this article.
>
> > Weaknesses 1.  significant concern  about comparison to Clark et al. 2021. provide a genuine comparison, the same degree of parameter searching/sweeping, at least against BP
>
>
> We understand the importance of ensuring that comparisons between different training methods are conducted under equivalent conditions to provide a fair and accurate assessment of their relative performance.
>
> To address your concerns, we have taken the following steps:
>
> - We re-implemented the existing methods, specifically Backpropagation (BP) and Direct Feedback Alignment (DFA), under the same training conditions as our proposed EBD algorithm.
> - This includes using the same learning rate schedulers, number of epochs, regularizations, and performing a comparable degree of hyperparameter optimization for all methods.
> - In our original submission, we noted that Clark et al., 2021 trained all their models for 190 epochs. In our experiments, we trained the MLP models for 120 epochs and the CNN and LC models for 100 epochs on MNIST and 200 epochs on CIFAR-10. For the revised experiments, we ensured that all methods (EBD, BP, and DFA) were trained for the same number of epochs specific to each architecture and dataset (for MLPs 120 epochs and for CNN/LC 100 epochs for MNIST and 200 epochs for CIFAR-10).
> - We performed extensive hyperparameter tuning for BP and DFA, similar to what was done for EBD.
> - Detailed hyperparameter settings and learning curves are provided in the appendix of the revised manuscript.
> - The results of these new experiments are included in the updated Tables 1 and 2 of our manuscript, which are also available below:
>
>
> MNIST Dataset:
> [x]: values from Clark et al., 2021
> [ours]: our numerical experiments
> DFA+E: DFA with correlative entropy regularization
> |     | DFA [x] | DFA [ours] | DFA+E [ours] | NN-GEVB [x] | MS-GEVB [x]| BP[x] | BP [ours] | EBD [ours] |
> |:--------:|:--------:|:--------:|:-----:|:-----:|:-----:|:------:|:----:|:---:|
> | MLP     |  97.91   |  98.09     | 98.21  | 98.13 | 97.69 |98.71 |98.72  | **98.24**  |
> | CNN     |  98.36   | 99.06     |  99.07 | 97.67 | 98.17  | 99.35 |99.46 | **99.08**  |
> | LC     | 98.52    |  98.90     | 98.90  | 98.22  |98.16  | 98.93 | 99.13  |  98.92 |
>
> CIFAR-10 Dataset:
> [x]: values from Clark et al., 2021
> [ours]: our numerical experiments
> DFA+E: DFA with correlative entropy regularization
> |     | DFA [x] | DFA [ours] | DFA+E [ours] | NN-GEVB [x] | MS-GEVB [x]| BP[x] | BP [ours] | EBD [ours] |
> |:--------:|:--------:|:--------:|:-----:|:-----:|:-----:|:------:|:----:|:---:|
> | MLP    |   50.46  | 52.09  | 52.22 | 52.38 | 51.14 | 55.31  | 56.37 | 55.17  |
> | CNN    |   55.93  | 58.39  | 58.56 | 66.26 | 61.57 | 71.2   | 75.24 | 66.42  |
> | LC     |   60.59  | 62.19  | 62.12 | 58.92 | 59.89 | 67.68  | 67.81 | 64.29  |
>
>
> - Based on these results, we observed significant improvements in the performance of BP and DFA compared to the results reported in Clark et al., 2021, due to the optimized training conditions.
> - Despite these improvements, our EBD algorithm still demonstrates competitive performance:
> On the MNIST dataset, EBD achieves slightly better accuracy than DFA under the same conditions.
> On the CIFAR-10 dataset, EBD shows a more pronounced improvement over DFA.
>
> We hope these new numerical experiments answers the concerns of the reviewer in terms of setting the optimization conditions of the algorithms to the same grounds.
>
> >Weakness 3.  scalability of simulations
>
> Our main contribution lies in offering a fresh theoretical backing for the error broadcast mechanism, which is believed to be a key process in biological learning, especially through the involvement of neuromodulators. While it is true that most biologically plausible frameworks face scaling issues, recent work by Launay et al. (2020) demonstrates that algorithmic enhancements can enable error broadcast approaches like Direct Feedback Alignment (DFA) to scale to large-scale, complex problems.
>
> Our framework is essentially similar to DFA, with the crucial distinction that our broadcasting weights are adaptive rather than fixed. Given these similarities with DFA and the demonstrated scalability in Launay et al.'s work, we believe that our proposed method can likewise be extended to train more complex network structures effectively.
>
> Ref:
> Launay, Julien, et al. "Direct feedback alignment scales to modern deep learning tasks and architectures." Advances in Neural Information Processing Systems 33 (2020): 9346-9360.

---

> > ### Author Response · Authors · 2024-11-18
> > **Response to Reviewer VBft's Comments (Part 2)**
> >
> > >Weakness 2 and Question 1: bio plausibility of power normalization and entropy regularization
> >
> > We would like to clarify and elaborate on how our work addresses these concerns by incorporating an online, more biologically plausible method:
> >
> >  - In our article, we employ batch optimization-based learning in numerical experimens for MLP, CNN and LC  implementations. The entropy and power normalization regularizers used in these numerical experiments are computed using batch calculations.
> >  - However , In Section 3, titled "Error-Based Learning in a Biologically More Realistic Network", we introduce our **biologically more plausible network framework**:
> >      -  We adopt the CorInfoMax network architecture from Bozkurt et al. (2023), but with a crucial modification: we replace the equilibrium propagation-based training with our proposed EBD training algorithm.
> >      -   In Bozkurt et al. (2023), the CorInfoMax network is derived from the **online optimization** of the correlative information objective, which is naturally using online learning and aligns well with biological reality.
> >      -   The **layer entropies** are part of this   optimization obtjective, and these entropies are optimized online through lateral (recurrent) connections among layer units, eliminating the need for batch-based entropy calculations.
> >      -   The power normalization can be enforced with batch of size 1 by adjusting the learning rate.
> >      -   We have provided new numerical experiment results for CorInfoMax-EBD with batch size of 1.
> >      -   Feedforward and feedback weights are updated using Hebbian learning rules, which are widely considered biologically plausible. Synaptic weights of recurrent (lateral) connections are updated using an anti-Hebbian learning rule, further enhancing the biological realism of the network.
> >      -   All updates in Algorithm 1 (CorInfoMax-EBD) are designed to be executed online, using a single input per training update (i.e., a batch size of $1$). This means that the algorithm does not require retaining and normalizing entire batches at each layer, directly addressing the reviewer's concern about batch learning undermining biological plausibility. While the algorithm is extendable to batch data if needed, its fundamental design supports online learning, which is more consistent with biological learning mechanisms.
> >      -   CorInfoMax-EP approach in Bozkurt et al. (2023) uses equilibrium propagation learning rule which relies on a two-phase update, time-contrastive process that raises questions about its biological realism. Our version, CorInfoMax-EBD, uses EBD learning rule, which allows for single-phase updates without the need for separate phases.
> >
> > In summary, the CorInfoMax-EBD approach presented in (Section 3 of) our paper offers an online, biologically plausible framework that does not depend on batch learning. By utilizing Hebbian and anti-Hebbian learning rules and eliminating the need for batch-based regularizers, we address the feasibility of an online, biologically plausible approach as suggested by the reviewer. (See also General Response to Reviewer's Comments-1 above)
> >
> > > Weakness 4. separation of  EBD and its extensions
> >
> > Thank you for this comment. We indeed revised our article by takin this comment into account: we moved the extensions placed in Section 4 as a subsection for Section 2. Now the flow is more smooth as the extensions of the MLP based method are provided in the same section. We also modified the article in a way to make it more accessible in terms of understanding and interpretations.
> >
> > >Question 2: Beyond MSE loss
> >
> > Our proposed framework exploits the nonlinear orthogonality property specific to MMSE (Minimum Mean Square Error) estimators. While the MMSE objective is a sensible choice for especially for regression problems, we acknowledge that different tasks potentially require different loss functions—such as cross-entropy.
> >
> > We are not currently aware of similarly powerful theoretical properties for loss functions like cross-entropy. This presents an intriguing opportunity for future research to explore whether analogous properties exist for other loss functions used in network training.
> >
> > However, our numerical experiments detailed in Appendix F.2 ("Correlation in Cross-Entropy Criterion-Based Training") show that even when training with the cross-entropy loss, we observe a similar decrease in correlation between errors and layer activations. This observation suggests that the decorrelation phenomenon might be a more general and fundamental aspect of the learning process, extending beyond the MMSE objective.
> >
> > Therefore, while the MMSE estimator may not always be the best objective for every network and task, the underlying decorrelation feature might still play a crucial role across different loss functions. We believe that further investigation into this area could yield valuable insights into the fundamental mechanisms of learning in neural networks.

---

> > > ### Author Response · Authors · 2024-11-18
> > > **Response to Reviewer VBft's Comments (Part 3)**
> > >
> > > >Q3: computational complexity
> > >
> > > **For MLP, CNN, LC models:** For MLP, CNN, and LC models, the updates of the error-hidden layer cross-correlation matrices and hidden layer activation correlation matrices have computational complexity proportional to the hidden layer size  ${N^{(k)}}^2$, which does not significantly impact overall computational complexity.  The primary source of computational cost lies in calculating the gradient for the layer entropy terms, $\log\det(\mathbf{R}_{\mathbf{h}}+\epsilon \mathbf{I})$.   The relative average runtimes from the simulations, normalized to BP, are presented below:
> > >
> > >
> > > Table: Average Runtimes in MNIST (relative to BP)
> > > | Model    | DFA      | DFA+E     | BP   | EBD |
> > > |:--------:|:--------:|:--------:|:------:|:---:|
> > > | MLP      | 3.40     | 7.68     | 1.0  | 8.06  |
> > > | CNN      | 1.68     | 2.95     | 1.0  | 3.85  |
> > > | LC       | 1.61     | 3.57     | 1.0  | 3.54  |
> > >
> > > Table: Average Runtimes in CIFAR-10 (relative to BP)
> > > | Model    | DFA      | DFA+E     | BP   | EBD |
> > > |:--------:|:--------:|:--------:|:------:|:---:|
> > > | MLP      | 2.85     | 6.94     | 1.0  | 7.61  |
> > > | CNN      | 2.10     | 3.24     | 1.0  | 4.11  |
> > > | LC       | 1.35     | 2.01     | 1.0  | 2.41  |
> > >
> > > Based on these results, entropy regularization in both EBD and DFA more than doubles the average runtime. However, these runtimes could be significantly reduced by more carefully implementing the entropy gradient terms by avoiding inverse calculations in the gradient terms. A practical approach would be to update the inverse of the correlation matrices directly, rather than recalculating both the matrices and their inverses at each step. This approach would mirror the structure of the CorInfoMax-(EP/EBD) networks. However, we did not pursue this path as CorInfoMax networks already effectively use this strategy. Our primary goal in presenting the results for DFA+E is to illustrate the runtime requirements of each method component more clearly. This allows us to distinguish the impact of the decorrelation and entropy maximization objectives. We also added these runtime comparison tables with a brief discussion to the Appendix Section-E.8
> > >
> > > Regarding the complexity comparison to NN-GEVB and MS-GEVB: We would also like to note, as mentioned in Appendix Section L of Clark et al., that NN-GEVB and MS-GEVB are vectorized models with a computational complexity for the forward pass that is higher by a factor of $K$ (typically, $K=10$) compared to conventional networks. In contrast, EBD preserves its inference-time efficiency without requiring architectural modifications or without increase in inference time.
> > >
> > > **For the CorInfoMax-EBD model**, which uses the same architecture as CorInfoMax-EP, entropy maximization is efficiently managed by the lateral weights. EBD reduces the two-phase update process of EP learning to a single-phase update, resulting in some computational savings. However, this saving is not substantial, as only one phase of EP involves fewer network dynamics iterations. It is important to note that the motivation for applying the EBD rule here is not computational efficiency but rather biological plausibility.

---

> ### Comment · Reviewer_VBft · 2024-11-25
>
> First, thank you for the detailed response - it is clear that you have put significant work into improving this work. Below I try to respond to all three sections of comments, first at an overview level, and then at a point-by-point level.
>
> **Overview:** The work has been improved significantly in terms of empirical results. As an example, the results in Table 1/2/3 are now far more appropriate for comparison. However, the written form of the work leaves much still to be desired, especially the complete absence of a serious discussion or explanation of the drawbacks of this work and considerations that a reader should make when drawing conclusions. I remain believing that this idea is an important contribution to the field, however there remain a number of issues with the presentation that I believe it should be improved further before acceptance (i.e. the form does not yet meet the quality level in my opinion). Based upon current changes, my score has improved by one point but not more.
>
> __Point by Point__
> - The results of Table 1 are much improved by a re-implementation and are now more comparable. They are however presented in an unorthodox way: bold values generally indicate the best performing model (with underscores used for second best). In this Table, BP should be bold and in many cases your methods as underscores. This should be corrected for reader clarity.
> - Scalability is ignored in favour of DFA's application in existing work to other domains (Launey et al. 2020). However, I do not believe that this applies to the current work. There has been a great proliferation of bio-plausible learning rules in recent years, and based upon the work by Bartunov et al. (2018) I believe that we should test all proposals against truly high-dimensional tasks such as ImageNet to verify that they are capable of credit assignment in such a regime. I believe that this would take some time to implement but is important.
> - The addition of an appendix to better illuminate how entropy updates can be computed without an inverse is helpful. However, given the complexity of this argument on biological plausibility, I believe that the main text could be much more clear in outlining how and why this implementation of CorrInfoMax is indeed more biologically plausible. At present this is rushed and unclear.
> - Modifications to the explanation of EBD are appreciated and it is now a clearer read.
> - The implications for an alternative loss function (CCE) are now present in an appendix, however this appendix is never referred to in the main text. This is a good demonstration of the way in which important considerations for application of this rule are not fully discussed in the main text. I am aware that the page limit is ... well ... limiting. But it is important to provide such context to make this paper of high quality.
> - Thank you for the addition of runtime comparisons, these are additionally useful.
>
> __Future Score Improvements__: To be clear about what I would need for a higher future score, I have included this section. Most importantly, a wider perspective on the explanation, implications, and drawbacks of this work is necessary. This would mean a re-writing of the narrative of the work, with more focus on bringing the reader along through both the positive contributions (more details on bio-plausibility and a clearer overall narrative), but also balanced with the potential remaining open-avenues (e.g. a real discussion of the implication of other loss functions) and remaining open questions. This would complete the written side of this work to a standard of having an acceptance score. To go further beyond this (in terms of score), I would request an application of the algorithm to a much more challenging task than those presented currently (specifically ImageNet classification would suffice) and for an excellent score an application to even other more general tasks, e.g. language models/transformers or such.

---

> > ### Author Response · Authors · 2024-11-25
> > **Response to Reviewer VBft's Comments**
> >
> > > thank you for the detailed response
> >
> > We thank the reviewer for the detailed constructive comments, which helped improving our article. Below you can find our descriptions about how your comments are handled in the final revision of our article.
> >
> > > Overview 1: ... absence of a serious discussion or explanation of the drawbacks
> >
> > In the revised article, we provided additional detailed discussions as described by the responses to point by point comments below. In addition, we modified and extended the limitations section. This section outlines the main limitations related to our approach and its coverage in the article. We added a discussion about the scalability issue:
> >
> > "**Limitations.** *The current implementation of EBD involves several hyperparameters, including multiple learning rates for decorrelation and regularization functions, as well as forgetting factors for correlation matrices. Although these parameters offer flexibility, they add complexity to the tuning process. Additionally, the use of dynamically updated error projection matrices and the potential integration of entropy regularization may increase memory and computational demands. Future work could explore more efficient methods for managing these components, potentially automating or simplifying the tuning process to enhance usability. Furthermore, while the scalability of EBD is left out of the focus of the article, we acknowledge its importance. Launay et al. (2020) demonstrated that DFA scales to high-dimensional tasks like transformer-based language modeling. Since DFA is equivalent to EBD with frozen projection weights and without entropy regularization, we anticipate that EBD could scale similarly."*
> >
> > > Point by Point
> >
> > >P.byP.1. ... Table, BP should be bold and in many cases your methods as underscores
> >
> > We thank the reviewer for this suggestion. In the initial version of the article, we used bold for the best broadcasting approach. In the revised article, we updated Tables 1/2/3 based on reviewer's suggestion, and marked the best-performing methods as bold, and the second-best as underlined. With this enhancement, the clarity of the tables is significantly improved.
> >
> > > P.byP. 2: Scalability is ignored in favour of DFA's application in existing work to other domains (Launay et al. 2020)....should test all proposals against truly high-dimensional tasks such as ImageNet...
> >
> > We believe the reference (Launay et al. 2020) on scaling the DFA algorithm is highly relevant, as DFA is equivalent to the EBD algorithm with frozen projection weights (and without added regularization functions.) For the future extensions of our method, we consider its application, especially to the sequence models (with recurrent structures), where the backpropagation encounters a serious drawback of vanishing or exploding gradients.
> >
> > Our article stands as a novel theoretical grounding for error broadcasting, and we believe that future algorithmic extensions would address more complex scenarios and architectures.

---

> > > ### Author Response · Authors · 2024-11-25
> > > **Response to Reviewer VBft's Comments (Part 2)**
> > >
> > > > P.byP. 3: ...appendix to better illuminate how entropy updates can be computed without an inverse is helpful...  main text could be much more clear in outlining how and why this implementation of CorrInfoMax is indeed more biologically plausible....
> > >
> > > We would like to thank reviewer for this improvement suggestion. We indeed restructured Section 3.2, where the biologically plausible CorInfoMax-EBD is introduced. At the end of Section 3.2, we provided more discussion to clarify the relative biological plausibility of the proposed CorInfoMax-EBD.
> > >
> > > First, we provided this discussion for comparing the CorInfoMaxEBD with the CorInfoMax-EP of (Bozkurt et al., 2024):
> > >
> > > *"By integrating EBD, we enable a single-phase update per input, eliminating the less biologically plausible two-phase learning mechanism required by CorInfoMax-EP. The two-phase approach of EP—comprising separate label-free and label-connected phases—is considered less plausible because biological neurons are considered unlikely to alternate between distinct global phases for learning. Our method not only simplifies the learning process but also aligns more closely with biological learning processes. Additionally, we achieve comparable or even superior performance compared to the CorInfoMax-EP (see Section 4.)"*
> > >
> > > We also added a detailed discussion about why the proposed CorInfoMax-EBD is more plausible than the MLP-based EBD:
> > >
> > > *"We also note that the CorInfoMax-EBD scheme proposed in this section is more biologically realistic than the MLP-based EBD approach in Section 2 due to several factors:*
> > >
> > > - *The MLP-based EBD approach employs an entropy regularizer in  (11), whose gradient involves the inverse of the layer-correlation matrix $\mathbf{B}^{(k)} = {\mathbf{R}^{(k)}_\mathbf{h}}^{-1}$, which in its direct  form appears non-biologically plausible. The same entropy term is an integral part of the CorInfoMax objective. As described in Appendix D, in the online optimization of the CorInfoMax objective, the entropy gradient can be implemented via lateral connections in the CorInfoMax network. Specifically, the learning gradients for this entropy function can be implemented as rank-1 (anti-Hebbian) updates on the $\mathbf{B}$ matrix when a batch size of $B=1$ is used. Note that the same lateral weights $\mathbf{B}^{(k)}$ are also updated by a three-factor rule due to the EBD update, as described in Algorithm 1.*
> > > - *Similarly, for CorInfoMax with $B=1$, the power normalization regularizer in  (10) reduces to the form $(h^{(k)}_l[n]^2 - P^{(k)})^2$ for each neuron. The gradient of this expression corresponds to local updates, enhancing biological plausibility. Even when a single sample is used for power calculation, the power regularizer remains effective due to the averaging effect across samples over time.*
> > > -  *In addition to the biologically plausible implementations of power and entropy regularizations in the online CorInfoMax setting, the neuron models used in CorInfoMax networks involve more realistic neuron models with apical and basal dendrite alongside the soma compartment.*
> > >  - *Another aspect contributing to the biological plausibility is the existence of feedback connections (corresponding to the backward predictors) in the CorInfoMax network structure."*
> > >
> > > We believe these changes contributed positively to the clarity of Section 3.2.
> > >
> > > > P.byP. 4: Modifications to the explanation of EBD are appreciated and it is now a clearer read
> > >
> > > Thank you, we appreciate your feedback.

---

> > > > ### Author Response · Authors · 2024-11-25
> > > > **Response to Reviewer VBft's Comments (Part 3)**
> > > >
> > > > > P.byP. 5: The implications for an alternative loss function (CCE) ... never referred to in the main text... in an appendix...not fully discussed in the main text....But it is important to provide such context to make this paper of high quality.
> > > >
> > > > Following this suggestion by the reviewer, in the new revision of the article, we modified and extended the conclusion part to include a discussion about extension to alternative loss functions with a link to Appendix F.2:
> > > >
> > > >
> > > > "*In this article, we introduced the Error Broadcast and Decorrelation (EBD) framework as a biologically plausible alternative to traditional backpropagation. EBD addresses the credit assignment problem by minimizing correlations between layer activations and output errors, offering fresh insights into biologically realistic learning. This approach provides a theoretical foundation for existing error broadcast mechanisms in biological neural networks and facilitates flexible implementations in neuromorphic and artificial neural systems. EBD's error-broadcasting mechanism aligns with biological processes where global error signals modulate local synaptic updates, potentially bridging the gap between artificial learning algorithms and natural neural computations. Moreover, EBD's simplicity and parallelism make it suitable for efficient hardware implementations, such as neuromorphic computing systems that emulate the brain's architecture.*
> > > >
> > > > *We believe that the MMSE orthogonality property underpinning the proposed EBD framework has great potential for developing new algorithms, deepening theoretical understanding, and analyzing neural networks in both artificial and biological contexts.* **We are currently unaware of similar theoretical properties for alternative loss functions. Notably, our numerical experiments in Appendix F.2 reveal that similar decorrelation behavior  occurs for networks trained with backpropagation and categorical cross entropy loss, suggesting that decorrelation may be a general feature of the learning process and an intriguing avenue for further investigation.**"
> > > >
> > > >
> > > > > P.byP. 6: Thank you for the addition of runtime comparisons, these are additionally useful.
> > > >
> > > > Thank you for this constructive comment which improved the quality of our article.

---

> > > > > ### Comment · Reviewer_VBft · 2024-11-26
> > > > >
> > > > > This current form of the work is better but still remains in having drawbacks. One is the issue of scale which I have previously mentioned. The other is unfortunately still in the written aspect.
> > > > >
> > > > > Your inclusion of further text has helped, but it does not solve the aspect of narrative completely for me. At present, the paper is presented as if this novel algorithm does or at least should be expected to scale. In this respect, your work is not convincing enough in empirical results (there are not even experiments demonstrating how networks of different scale might perform in these task - or even simulations with more than 10 output classes). I would recommend pitching this more as potential avenue to explore, with less strong claims in the abstract, and main text/conclusion. In this respect, my above comment also meant to encourage you to really re-consider the how the paper is pitched on the whole in terms of its strengths and drawbacks.

---

> ### Author Response · Authors · 2024-11-26
> **Response to Reviewer VBft's Comments**
>
> Thank you for your constructive feedback on our manuscript. We genuinely appreciate your continued engagement and the opportunity to clarify and improve our work. We understand your concerns regarding scalability and the clarity of our article’s narrative on this aspect.
>
> Our primary goal in this article is to present a novel theoretical framework for learning. This framework provides principled underpinnings for both error broadcast-based learning and three-factor learning—mechanisms that are believed to play crucial roles in biological networks. By rooting our approach in a major orthogonality property from estimation theory—the nonlinear orthogonality principle—we aim to lay down the foundational aspects of this new learning framework. This approach is in line with several recent works that focus mainly on deriving principled methods for biological and artificial learning mechanisms (e.g., Clark et al. (2021), Bozkurt et al. (2023), Dellaferrera, et al. (2022) and Kao et al. (2024)).
>
> To address your concern about the clarity of our claims regarding performance and scalability, we have carefully revised the manuscript to ensure it accurately reflects the intended scope. We emphasize that while scalability is an important future direction, our present work is centered on establishing a new learning mechanism based on the nonlinear orthogonality principle.
>
> In response to your feedback, we have revised key parts of the abstract, introduction, and conclusion to clarify that the current article does not focus on scalability. Instead, we highlight that scalability is a valuable extension of our proposed theoretical framework.
>
> Below, we describe the specific changes made in the latest revision to address your concerns:
>
> - **Abstract:** We have revised the abstract and added the bolded sentence and phrase:
>
>     *"We introduce the Error Broadcast and Decorrelation (EBD) algorithm, a novel learning framework that addresses the credit assignment problem in neural networks by directly broadcasting output error to individual layers. The EBD algorithm leverages the orthogonality property of the optimal minimum mean square error (MMSE) estimator, which states that estimation errors are orthogonal to any nonlinear function of the input, specifically the activations of each layer. By defining layerwise loss functions that penalize correlations between these activations and output errors, the EBD method offers a principled and efficient approach to error broadcasting. This direct error transmission eliminates the need for weight transport inherent in backpropagation. Additionally, the optimization framework of the EBD algorithm naturally leads to the emergence of the experimentally observed three-factor learning rule. We further demonstrate how EBD can be integrated with other biologically plausible learning frameworks, transforming time-contrastive approaches into single-phase, non-contrastive forms, thereby enhancing biological plausibility and performance. Numerical experiments demonstrate that EBD achieves performance comparable to or better than* **known error-broadcast methods** *on benchmark datasets.* **The scalability of algorithmic extensions of EBD to very large or complex datasets remains to be explored. However,** *our findings suggest that EBD offers a promising, principled direction for both artificial and natural learning paradigms, providing a biologically plausible and flexible alternative for neural network training  with inherent simplicity and adaptability that could benefit future developments in neural network technologies."*
> ---
> - **Introduction:** In the concluding paragraph of the introduction, we added a sentence to highlight scalability as an open question:
>
>     *"We demonstrate the utility of the EBD algorithm by applying it to both artificial and biologically realistic neural networks.* **While our experiments show that EBD performs comparably to state-of-the-art error-broadcast approaches on benchmark datasets, offering a promising direction for theoretical and practical advancements in neural network training, its scalability to more complex tasks and larger networks remains to be investigated.**"
> ---
> - **Related Work and Contribution:** In the final paragraph of Section 1.1, "Related Work and Contribution," we softened the language to temper claims about the advantages of our approach:
>
>     "*In summary, our approach provides a theoretical grounding for the error broadcasting mechanism and* **suggests ways to** *its effectiveness in training networks.*"

---

> ### Author Response · Authors · 2024-11-26
> **Response to Reviewer VBft's Comments (Part 2)**
>
> - **Limitations:** Lastly, we revised the Limitations section and added the final sentence, highlighted in bold, to more clearly articulate our paper's position on scalability.  This addition frames scalability as an important area for future exploration:
>
>     "**Limitations.** *The current implementation of EBD involves several hyperparameters, including multiple learning rates for decorrelation and regularization functions, as well as forgetting factors for correlation matrices. Although these parameters offer flexibility, they add complexity to the tuning process. Additionally, the use of dynamically updated error projection matrices and the potential integration of entropy regularization may increase memory and computational demands. Future work could explore more efficient methods for managing these components, potentially automating or simplifying the tuning process to enhance usability. Furthermore, while the scalability of EBD is left out of the focus of the article, we acknowledge its importance. Launay et. al. (2021) demonstrated that DFA scales to high-dimensional tasks like transformer-based language modeling. Since DFA is equivalent to EBD with frozen projection weights and without entropy regularization, we anticipate that EBD could scale similarly.* **However, this remains unvalidated empirically. Examining EBD’s scalability and streamlining its components to improve usability are important tasks for future work.**"
> ---
>
> In conclusion, we appreciate the reviewer’s feedback and have carefully addressed the concerns about narrative. The revised manuscript aims to present a more balanced perspective, clearly outlining the scope and limitations of this work. By framing scalability as a future direction rather than an immediate claim, we hope to make the paper’s contributions and positioning clearer while setting the stage for future research that builds on these findings.
>
>
>
> References:
> - Clark, et. al. "Credit assignment through broadcasting a global
> error vector.", Neurips, 2021.
> - Bozkurt, et. al. "Correlative information maximization: a biologically plausible approach to supervised deep neural networks without weight symmetry.", Neurips, 2023.
> - Kao et. al., "Counter-Current Learning: A Biologically Plausible Dual Network Approach for Deep Learning", Neurips, 2024.
> - Dellaferrera et al., "Error-driven Input Modulation: Solving the Credit Assignment Problem without a Backward Pass", ICML, 2022

---

> > ### Author Response · Authors · 2024-12-01
> > **Final Follow-Up Before Discussion Period Ends**
> >
> > We would like to thank you for your engagement and valuable feedback. As the discussion period concludes tomorrow, we wanted to ensure that our revisions and responses have fully addressed your comments and concerns. If there’s any additional information or clarification you need, please let us know—we would be happy to provide it. We appreciate that all reviewers recognized the novelty of our work, and we have thoughtfully integrated your suggestions to strengthen the manuscript. In its current form, we believe our article presents a novel learning paradigm grounded on an estimation-theoretical approach, with strong potential for impact on both neuroscience and machine learning, now with significantly improved clarity and depth.

---

### Official Review · Reviewer_JH3K · 2024-11-02

**Soundness:** 3
**Presentation:** 2
**Contribution:** 3
**Rating:** 8
**Confidence:** 3

**Summary:**

The authors propose a method for training neural networks based on decorrelating layer activities with the output error. This method avoids the need for backpropagation and is a potential solution to the weight transport problem.

**Strengths:**

- the text is generally well written
- the theoretical building block in which this is built - that optimal nonlinear MMSE estimators hvae error orthognoal to functions of input - is interesting and in my view certainly deserves the attention given by the authors. Its implementation - and therefore this paper - should be of value to both ML and neuroscience researchers.
- it is clear the authors have a good grasp of the theory and technical details of the models considered, and the approach in general seems well thought out
- relevant literature appears to be duly cited and compared, though I am a non-expert in this field
- the numerical results presented in section 5 appear impressive, though I would prefer they were elaborated upon a bit more.

**Weaknesses:**

- The paper is rather dense, and I worry that it harms its accessibility. It seems to me some of the technical details/results can be sacrificed to the appendix in place of more motivation/clarification. For example, section 3.2, and the relationship to corinfomax in general, is very difficult to grasp. The motivation seems to be that corinfomax is a biologically plausible model, but I don't understand why corinfomax-EBD is more biologically plausible than the implementation in 2.3. What was the original implementation lacking that corinfomax-EBD addresses?
- As per above, I would appreciate any more insight into the results and comparison vs other models. E.g. do you have any intuition as to why EBD outperforms NN-GEVB and MS-GEVB? For the corinfomax models it seems that the benefit of EBD is that it avoids the two-phases (?), but is there a reason it makes significantly improvements on the CIFAR-10 dataset?
- the notation can sometimes be sloppy (see below)

**Questions:**

- in line 156 is N_k the size of layer k? this should be specified
- that g can be any nonlinear function seems a powerful result. How much did you explore its possibilities? It seems a big hyperparameter to choose but I didn't get a feel for what it should be
- the error epsilon is a vector so why is it not in bold? (it currently appears as a scalar)
- for non-linear networks the error landscape is typically non-convex and has many local optima which are found during learning instead of one global optimum. How does the main theoretical results (lemmas A.1/A.2 tie in with this?
- for the equation in line 199 R is defined recursively, but what is R[0]?
- does the forgetting factor lambda lie in [0,1]?
- In section 2.3 what do W_1, W_2 mean? Do they directly related to W (e.g. the first/second column). I presume not given equations 6/7 but if they don't they should be called something else
- to what extent does EBD depend on batch size? It seems like it would require large batches to get a good correlation estimate, but this doesn't seem to fit in with the biological plausibility of the algorithm?
- Why is EBD a 3-factor learning rule but not backprop? is it not possible to consider the post-synaptic/modulatory signal as the error gradient with respect to the pre-synaptic neuron?
- in 3.2 why are the corinfomax equations which involve determinants etc biologically plausible? It's not clear to the non-expert reader. Given there are lateral connections, are we also dealing with RNNs instead of feedforward nets now?
- in algorithm 1 why are activations H and errors E and bias B now in caps? Also the link to the corinfomax equations above is not clear to me at all
- In section 4 line 393 it's written that these extensions are 'building on the CorInfoMax-EBD algorithm', but I don't understand why they can't also be applied to standard MLP?
- Could the power normalization equation in 4.1.1 not be written as a norm over the batch. I personally find the notation with [n] confusing
- out of interest is 4.1.2 itself bio-plausible?
- typos: line 398: period after stability; line 709: linear -> non-linear.

---

> ### Author Response · Authors · 2024-11-18
> **Response to Reviewer JH3K's Comments**
>
> > Strengths: well written, interesting theoretical block, certainly deserves attention, of value to both ML and neuroscience, approach well-thought out, numerical results apper impressive
>
> We really appreciate the positive assessment by the reviewer especially about the significance of our article's contributions. We would like to also thank you for the detailed review and the constructive comments.
>
> > Weakness 1: dense paper harming accessibility, section 3.2 difficult to grasp, why CorInfoMax-EBD more bio-plausible, and what is new in CorInfoMax-EBD
>
> We have revised article to address the concerns of the reveiewer to improve its accessibility. For this purpose,
>
> - We moved the EBD extensions section, which was plased after the section on the bioplausible CorInfoMax-EBD, to Section 2 where the EBD method is introduced.
> - We also moved some details about forward broadcasting to the appendix to improve readability.
> -  We added a new appendix (Appendix D) explaining bio-plausible CorInfoMax-EP model of Bozkurt et al., 2024. This new appendix section helps understanding the derivation and the operation of the CorInfoMax network dynamics as well as the equilibrium propagation based learning dynamics applied to the CorInfoMax network. This would clarify the Section 3.2 on the application of the EBD approach to reduce two phase learning to a single phase.
> -  In the main article, we have provided additional explanations and clarifications in each section, as much as the page limit allowed. (Please see the common response above for further details.)
> -  We have included cross-references to the new appendix sections to guide readers who wish to delve deeper into the technical details.
>
> Regarding CorInfoMax-EBD being more biologically plausible than the MLP based method in Section 2:
> -  *Online Operation vs. Batch Mode:* The MLP-based implementation in Section 2 requires batch mode and doesn't ensure biologically plausible entropy-gradient updates. In contrast, the CorInfoMax network operates in an online setting, can work with batch sizes as small as 1, and integrates entropy gradients into lateral weights, making the updates more biologically realistic.
> -  *Biologically Realistic Architecture:* CorInfoMax includes lateral (recurrent), feedforward, and feedback connections, mirroring biological neural networks. It employs a three-compartment neuron model—soma, basal dendrite, and apical dendrite—which better captures biological reality compared to standard artificial neurons. In this model, apical dendrites receive feedback, basal dendrites receive feedforward information, and prediction errors are represented by differences in membrane potentials.
> - To clarify these points, in Section 3.2, we added
>     *"The CorInfoMax-EBD scheme proposed in this section is more biologically realistic than the MLP based EBD approach in Section 2 due to multiple factors: Unlike the batch-mode operation required by the MLP-based EBD, CorInfoMax operates in an online optimization setting which naturally integrates entropy gradients into lateral weights, resulting in biologically plausible updates, whereas the MLP approach uses entropy regularization without ensuring biological plausibility. Besides, it employs a neuron model and network architecture that closely mirror biological neural networks."*
>
> We were not able to go into further details due to space limitations.
>
> We believe these revisions enhance the paper's clarity and better highlight our main concepts and contributions.

---

> > ### Author Response · Authors · 2024-11-18
> > **Response to Reviewer JH3K's Comments (Part 2)**
> >
> > > Weakness 2. why EBD outperforms NN-GEVB and MS-GEVB? the benefit of CorInfoMax-EBD over CorInfoMax-EP
> >
> > Regarding comparison with NN-GEVB and MS-GEVB, it is hard to pin point a special feature of our method that might cause improvements in performance. Our goal with these simulations is to illustrate that a theoretically backed bio-plausible error broadcast approach based on MMSE orthogonality principle can achieve comparable performance to the existing alternatives.
> >
> > We would also like to note, as mentioned in Appendix Section L of Clark et al., that NN-GEVB and MS-GEVB are vectorized models with a computational complexity for the forward pass that is higher by a factor of $K$ (typically, $K=10$) compared to conventional networks. In contrast, EBD preserves its inference-time efficiency without requiring architectural modifications or without increase in inference time.
> >
> > For the comparison of CorInfoMax-EBD with the CorInfoMax-EP: **Reducing two phase learning to single phase learning with three-factor-learning rule** is significant for more biological plausibility than the performance. For the improvement in the performance, we added the following statement in Section 4 of the revised article:
> >
> > *"These results confirm that the CorInfoMax network trained with the EBD method achieves equivalent performance on the MNIST dataset and significantly better performance on the CIFAR-10 dataset. One potential factor contributing to this improvement is that CorInfoMax-EBD incorporates error decorrelation in updating lateral weights, whereas CorInfoMax-EP relies only on anti-Hebbian updates."*
> >
> > > Weakness 3: the notation can sometimes be sloppy
> >
> > We have carefully applied the notation improvements you suggested and are grateful for your detailed feedback. In addition to your recommendations, we have also corrected typos and made further refinements to enhance the clarity and precision of the manuscript.
> >
> > >Questions 1. is $N_k$ layer size?
> >
> > Thank you. We added the description of $N^{(k)}$ in the revised article.
> >
> > >Q2. g= "any nonlinear function" is powerful result, how much did you explore?
> >
> > We are also curious about the wise choices for $g$. We tried  basic nonlinearities such as monomials, i.e., $g(x)=x^k$ and sinusoidal functions. Our experiments were not diverse and comprehensive enough, and we did not see significant improvement for these experiments.
> >
> > > Q3. why not epsilon bold?
> >
> > epsilon is indeed a vector, and we replaced it with bold-epsilon in the revised article. We appreciate the suggestion.
> >
> > > Q4. lemmas A.1/A.2 in relation to the loss landscape
> >
> > we have not yet developed a theoretical characterization of the loss landscape associated with our proposed method. The non-convexity in our loss function arises from the inherent non-convex relationship between the estimation error and the network parameters—a characteristic that is also present in networks trained using the MSE loss. Because both our method and the MSE-based training share this fundamental source of non-convexity, we expect that the optimization landscapes are similar in terms of their features and challenges.
> >
> > > Q5. what is R[0]?
> >
> > Thanks for making this point. We added "$\hat{\mathbf{R}}_{\mathbf{g}^{(k)}(\mathbf{h}^{(k)}){\epsilon}}[0]$ is the initial value for the correlation matrix, which is an algorithm hyperparameter" after the recursive update equation.
> >
> > >Q6. $\lambda\in [0,1]$?
> >
> > Yes. We modified $\lambda$ definition after the recursive equation as $\lambda\in [0,1]$ in the revised article.
> >
> > >Q7. what do W_1, W_2 mean?
> >
> > Subindex {1,2} are for $\Delta \mathbf{W}$ rather than $\mathbf{W}$, which corresponds to two different terms in the derivative. In the revised article, we added the following description after the equations where they first appeared: *"Here $\Delta\mathbf{W}^{(k)}_1,  \Delta{b}^{(k)}_1[m]$ ($\Delta\mathbf{W}^{(k)}\_2, \Delta b^{(k)}\_2[m]$)  represent the components of the gradients containing derivatives of activations (output errors) with respect to the layer parameters."*

---

> > > ### Author Response · Authors · 2024-11-18
> > > **Response to Reviewer JH3K's Comments (Part 3)**
> > >
> > > >Q8.  to what extent does EBD depend on batch size?
> > >
> > > We need to distinguish between the MLP, LC, and CNN implementations and the CorInfoMax-EBD, which is the biologically more plausible network with EBD learning rule. The MLP, LC, and CNN implementations are aimed at providing a proof of concept for implementing EBD in artificial neural networks. Consequently, they are primarily batch-algorithm implementations, using batch sizes of 20 for the MLP and 16 for the CNN and LC. These sizes are significantly smaller than the batch size of 128 used in Clark et al., 2021.
> > >
> > > CorInfoMax-EBD, on the other hand, is based on an online optimization setting, and it is not batch algorithm. However, it can be used to work in batch mode, therefore, we included batch size parameter $B$ in Algorithm 1 (CorInfoMax-EBD) description. Our new numerical experiments for CorInfoMax-EBD confirm that with a batch size of $1$, the network can still achieve satisfactory performance. For instance, on the CIFAR-10 dataset, CorInfoMax-EBD with a batch size of $1$ reaches an accuracy of over $53.4%$, compared to the CorInfoMax-EP of Bozkurt et al., 2020, which achieves $50.97%$ accuracy with a batch size of $20$. We are currently conducting further hyperparameter tuning for both the MNIST and CIFAR-10 datasets.
> > >
> > > Below is the updated version of Table 3 from our article, listing the current CorInfoMax-EBD accuracy values for a batch size of $1$. We will replace these with the most up-to-date values during the discussion period
> > >
> > >
> > > Table 3: Accuracy results (%) for EP and EBD CorInfoMax algorithms. Column marked with [x] is
> > > from reference Bozkurt et al. (2024).
> > > |Data Set |CorInfoMax-EP[x]  (batch size:20) | CorInfoMax-EBD (Ours) (batch size:20) | CorInfoMax-EBD (Ours)(batch size:1) |
> > > |:-------:|:--------:|:--------:|:--------:|
> > > | MNIST  | 97.58     | 97.53   |   94.7   |
> > > |CIFAR-10 |  50.97   | 55.79   |   53.4   |
> > >
> > > In summary, the CorInfoMax-EBD approach presented in (Section 3 of) our paper offers an online, biologically plausible framework that does not depend on batch learning. By utilizing three-factor lerarning as a consequence of the application of EBD method, together with Hebbian and anti-Hebbian learning rules and eliminating the need for batch-based regularizers, we address the feasibility of an online, biologically plausible approach
> > >
> > > >Q9. Is backprop 3-factor learning rule?
> > >
> > > This is a fair and insightful question.  It's true that backpropagation (BP) can be mathematically framed as a three-factor learning rule when dissected appropriately. Specifically, the gradient of the loss function with respect to the weights in BP can be expressed as the product of:
> > >
> > >  i.Pre-synaptic Activity
> > >  ii. The derivative of post-synaptic activation function
> > >  iii. backpropagated error term from the next layer, serving as the modulator.
> > >
> > >  From this perspective, considering the derivative of the activation function as a form of post-synaptic activity, BP does align with the three-factor rule framework. At the same time, we make the following observations about BP updates:
> > >
> > >  - BP requires the exact transposition of forward weights during the backward pass to compute the error signals.
> > >  - In BP, the learning rule's reliance on the derivative of the activation function.
> > >
> > > On the other hand, EBD employs a mechanism where error signals are broadcasted globally across the network, akin to neuromodulatory systems in the brain, which eliminates the need for precise symmetric feedback pathways. Furthermore, EBD allows for a broader range of post-synaptic activities beyond the strict derivative of an activation function, potentially accommodating the diverse behaviors of biological neurons.
> > >
> > > > Q11. ambiguity about capital letters in Algorithm 1?
> > >
> > > Thank you for highlighting the confusion regarding the notation in Algorithm 1. In our revision, we have added a new Appendix D that provides a summary of the CorInfoMax network derivation and its learning dynamics as presented in Bozkurt et al., 2024. This addition serves to:
> > >
> > > - Clarify the CorInfoMax-EP Approach and Network Architecture: We explain the existing CorInfoMax-EP method and detail the architecture of the CorInfoMax network.
> > > - Clarify Variable Notation: We define all variables used, particularly those in the Algorithm 1.
> > >
> > > Regarding the notation:
> > >
> > > - $\mathbf{H}$: Represents a matrix containing a batch of activations when using a batch size greater than 1. We use capital bold letters to denote matrices.
> > > - $\mathbf{E}$: Denotes the matrix of output errors for the batch.
> > > - $\mathbf{B}$: Stands for the lateral connection weights, not bias. This is clarified in new Appendix D.
> > >
> > > In the revised Algorithm 1, we have updated the "Input" section to clearly define all these variables and included references to the equations where they are introduced. This should make the notation and its connection to the CorInfoMax equations clear.

---

> > > > ### Author Response · Authors · 2024-11-18
> > > > **Response to Reviewer JH3K's Comments (Part 4)**
> > > >
> > > > >Q10. Why CorInfoMax objective with determinants bio plausible?
> > > >
> > > > At first look, the  online CorInfoMax objective in Section 3.2  involes determinants, and At first glance, the online CorInfoMax objective in Section 3.2 includes terms like
> > > > $\log\det(\mathbf{R}_h^{(k)}[m])$, which might seem to require batch processing and unsuitable for online, biologically plausible implementation. However, the CorInfoMax framework in Bozkurt et al., 2023  achieves online updates and maintains biological plausibility through the following key observations:
> > > >
> > > > - For the online implementation,  the correlation matrix for layer-$k$ is updated using a rank one update formula:
> > > >  $\mathbf{R}_h^{(k)}[m]=\lambda \mathbf{R}_h^{(k)}[m-1]+(1-\lambda)\mathbf{h}^{(k)}[m] \mathbf{h}^{(k)}[m]^T$.
> > > > - The gradient of the corresponding  correlative entropy $\log\det(\mathbf{R}_h^{(k)}[m])$, with respect to the activations $\mathbf{h}^{(k)}[m]$  given by  $(1-\lambda)\mathbf{B}_h^{(k)}[m]\mathbf{h}^{(k)}[m]$  where $\mathbf{B}^{(k)}_h={\mathbf{R}^{(k)}_h}^{-1}$ is the inverse correlation matrix. Therefore, updating layer activations with this gradient corresponds to recurrent (lateral) connections, where the synaptic strengths are given by $\mathbf{B}_h^{(k)}[m]$. Therefore, the corresponding network is indeed RNN. Biological neural networks are inherently recurrent, with neurons influencing each other through lateral connections. The recurrent structure arising from the gradient updates aligns the CorInfoMax network with this biological characteristic.
> > > > - Given the rank-1 update on the correlation matrix, through the use of matrix inversion lemma, the inverse correlation weights $\mathbf{B}_h^{(k)}$ are also updated by a rank-1 term.
> > > >
> > > > Therefore, the RNN structure of the CorInfoMax neural network is based on the gradient ascent update of the online CorInfoMax optimization. As the learning rule for synaptic weights, Bozkurt et al. 2023 employs two-phase contrastive updates of the equilibrium propagation scheme. We have replaced the two-phase contrastive learning rule with the EBD algorithm, eliminating the need for a two-phase learning rule.
> > > >
> > > > The new appendix section (Appendix D) offers details provided above to address the reviewer's concern.
> > > >
> > > > > Q12. on the phrase "building on the CorInfoMax-EBD algorithm"
> > > >
> > > > This statement was unclear. These extensions were actually intended for the standard MLP. In the revised version, we have reorganized these sections, placing the extensions immediately after the MLP section in Section 2.
> > > >
> > > > >Q13. on power normalization notation
> > > >
> > > > The proposed power normalization is calculated over the batch.  However, we could replace this expression with $\sum_{l=1}^{N^{(k)}}\left(\frac{1}{B}\|\mathbf{H}^{(k)}_{l,:}[m]\|_2^2-P^{(k)}\right)^2$, if the reviewer thinks it is more clear notation avoiding subindexing over the batch elements. Another alternative, which avoids batch calculation is to use an exponentially averaged power.
> > > >
> > > > >Q14. is $\log\det$ entropy in extensions bio-plausible?
> > > >
> > > > Yes, the layer entropy optimization in Section 4.1.2 can be implemented in a biologically plausible way, depending on the approach used. This is detailed in Appendix D, where we describe the CorInfoMax method with biologically plausible entropy maximization. In the CorInfoMax approach:
> > > >
> > > > - The layer correlation matrix is computed using an autoregressive process with a rank-one update.
> > > > - The derivative of the entropy function involves the inverse of this correlation matrix, denoted as $\mathbf{B}$, which is also updated using a rank-one update.
> > > > - This matrix  $\mathbf{B}$ corresponds to the lateral weights in a recurrent neural network (RNN).
> > > >
> > > > By structuring the computations in this manner, the layer entropy optimization aligns with biological plausibility. (See also related part of General Response to Reviewer's Comments-1 above)
> > > >
> > > > >Q15. period after stability, linear -> non-linear.
> > > >
> > > > Corrected these typos in the revision. Thank you.

---

> ### Comment · Reviewer_JH3K · 2024-11-26
>
> Thank you authors for your comprehensive reply.
>
> Performance wise, I'm impressed with the additional simulations that have been carried out and am convinced by the emperical results. I also think that the fact CorInfoMax-EBD can operate in a batch size = 1 does improve its bio plausibility. As I originally stated, I also repeat that I think the paper's central concepts are interesting and should be encouraged in the field of computational neuroscience.
>
> Based on this I will increase my score. That said, even after various notation fixes and the addition of individual sentences (which I do appreciate), I do slightly worry that the structure/delivery of the paper makes it difficult to grasp for a non-expert, and I do agree with VBft's comments that future efforts should prioritise the overall narrative/story of the work.

---

> > ### Author Response · Authors · 2024-12-01
> > **Final Follow-Up Before Discussion Period Ends**
> >
> > We would like to thank you for your engagement and valuable feedback. As the discussion period concludes tomorrow, we wanted to ensure that our revisions and responses have fully addressed your comments and concerns. If there’s any additional information or clarification you need, please let us know—we would be happy to provide it. We appreciate that all reviewers recognized the novelty of our work, and we have thoughtfully integrated your suggestions to strengthen the manuscript. In its current form, we believe our article presents a novel learning paradigm grounded on an estimation-theoretical approach, with strong potential for impact on both neuroscience and machine learning, now with significantly improved clarity and depth.

---

### Official Review · Reviewer_P7w2 · 2024-11-03

**Soundness:** 2
**Presentation:** 3
**Contribution:** 2
**Rating:** 5
**Confidence:** 3

**Summary:**

In this paper, the authors introduce the Error Broadcast and Decorrelation (EBD) algorithm as a novel method for implementing gradient descent in deep neural networks. This approach addresses the limitations of traditional backpropagation (BP), which requires biologically unrealistic components such as weight transport and symmetric feedback pathways. The EBD algorithm builds on a key theorem from minimum mean square error (MMSE) estimation, which states that the output error of an optimal estimator is orthogonal to any function of the input. Leveraging this property, the authors propose that the activations in each layer, as functions of the input, be orthogonal to the output error. This orthogonality condition forms the basis for their weight update rule, which aims to decorrelate activations and output errors at each layer. The proposed EBD framework demonstrates competitive performance with BP on benchmark datasets like MNIST and CIFAR-10, particularly in fully connected and simpler architectures. The authors also explore potential extensions to the algorithm, including regularization techniques and forward projection of activations, to enhance stability and performance.

**Strengths:**

1. **Compelling Biological Motivation**: Exploring error broadcasting and feedback alignment as methods for implementing gradient descent in neural networks holds significant promise for biological plausibility. These approaches circumvent the weight transport problem by directly transmitting error signals to deeper layers. This work introduces an innovative algorithm within this framework, which exhibits improved performance compared to previous error broadcasting methods.

2. **Normative Approach and Theoretical Foundation**: The algorithm’s development, rooted in the theoretical orthogonality property of an optimal MMSE estimator, is intriguing. Framing this method as a normative approach that leverages optimal predictor properties is commendable. However, as noted below, a potential misuse of this theorem raises concerns.

3. **Practical Demonstration with Numerical Results**: The empirical findings showcase the proposed algorithm's practicality, albeit with limitations. Its reported performance on benchmark datasets suggests that it is competitive with state-of-the-art alternatives under certain conditions.

**Weaknesses:**

1. **Theoretical Assumptions and Interpretation**: The reliance on the theorem regarding the orthogonality of an optimal estimator’s error to any function of its input, while foundational, is problematic when extended in reverse. The paper does not adequately explain the consequences of requiring orthogonality between output error and hidden layer activations. Furthermore, in most applications and architecture, the dimensionality of the hidden layers is very large. Constraining a solution to be orthogonal to a single direction is weak, and its benefits are poorly defined. This gap leaves uncertainty regarding how orthogonality aids learning or inference. Thus, the theoretical basis appears tenuous and potentially misapplied.

3. **Ambiguity in Performance Implications**: Although the algorithm performs well on real datasets, this success might stem from something other than the stated theoretical premise. The observed gains could be attributed to a different mechanism, such as orthogonal representations of different classes, rather than the error signal’s orthogonality. It would be valuable for the authors to test whether the performance improvement is due to orthogonalized class representations or if it is indeed a result of their premise. A comparative baseline using local class orthogonalization with an SGD-trained readout on the penultimate layer would provide insights into the true contribution of the proposed mechanism.

4. **Biological Plausibility of Batch Learning**: The paper’s claim of biological relevance is weakened by the batch learning requirement, which necessitates retaining and normalizing the entire batch at each layer. While replacing weight transport and feedback pathways with error broadcast is a step toward biological realism, the reliance on batch-based updates undercuts this claim. The authors should consider the feasibility of online, more biologically plausible approaches and address whether the proposed method truly enhances biological plausibility. Alternatively, the paper can focus on the mathematical foundation of error broadcasting and not on biological realistic implementations of gradient descent.

5. **Lack of Clarity in Possible Extensions**: In Section 4, the authors introduce several extensions to the EBD algorithm. The first involves regularization techniques aimed at preventing layer collapse. While preventing collapse is essential for maintaining active and diverse representations, the specific normalization methods proposed are neither novel nor particularly informative. Their inclusion does not substantially enhance the originality of the work.
   The second extension discussed is the forward projection of neural activations onto the output or penultimate layer, followed by an orthogonalization process at that stage. The rationale behind this step remains unclear. The manuscript provides no compelling biological basis for this projection mechanism, suggesting that its primary motivation is performance optimization rather than biological plausibility. Notably, the statement in line 448—“This projection facilitates the optimization of the decorrelation loss by adjusting the parameters of the final layer”—is ambiguous. It lacks a clear, rigorous mathematical explanation that would elucidate how this projection supports the training process. A more detailed formulation or analysis is necessary to justify the inclusion and clarify the impact of this component on the algorithm’s overall functionality.

6. **Performance Analysis on Complex Architectures**: The results presented in Section 5 show that the algorithm's performance is almost on par with backpropagation. However, demonstrating performance close to BP on simpler datasets, such as MNIST or fully connected networks trained on CIFAR-10, is not sufficiently informative. While useful as initial proof-of-concept validations, these comparisons do not substantiate the broader claims of the algorithm’s novelty or practical utility. Combined with the previously mentioned theoretical limitations, the results fall short of convincingly demonstrating the value and distinct advantages of the proposed EBD approach.

**Questions:**

1. The premise of this work hinges on the orthogonality of the error of the optimal estimator to the neural representations. However, the reverse is not addressed: a vector that is orthogonal to a set of functions of the input is not necessarily indicative of an optimal estimator, and functions orthogonal to the estimator may not be meaningful. Could you clarify the theoretical foundation of your algorithm and its precise connection to the theorem you reference?

2. To strengthen your claims, it would be helpful to demonstrate that the performance improvements are not solely due to orthogonalizing representations in each layer. Your experimental results primarily focus on training fully connected networks on MNIST and CIFAR-10, raising the possibility that similar gains could be achieved through layer-wise orthogonal class representations with SGD applied only at the output. Can you comment on this possibility or provide additional evidence?

3. Is there a potential for an online version of your algorithm that eliminates the need for batch learning? If so, how would this be implemented?

---

> ### Author Response · Authors · 2024-11-13
> **A kind request for clarification**
>
> Dear Reviewer,
> We would like to thank you for the comprehensive review. Before drafting our response, we want to ensure we fully understand the reviewer’s points. Therefore, we kindly request clarification on the following:
>
> In Weakness Point 2 and Question 2, the reviewer suggests that the performance gains reported in our article may result from 'orthogonalization of layer representations' and/or 'orthogonal representations of different classes.' We would be grateful if the reviewer could clarify precisely what is meant by these terms and indicate any mechanisms within our method that might be contributing to this effect. We note that in our framework, 'orthogonality' refers to being uncorrelated in a statistical sense. However, we believe the reviewer’s use of orthogonality pertains to vector orthogonality with respect to the Euclidean inner product.
>
> Additionally, could the reviewer provide more details about the exact implementation of the  “comparative baseline” that we are asked to compare against: for example, what is meant by and how is local class orthogonalization implemented in each layer? Is there an available article/codebase that the reviewer can point us to for this comparison baseline.

---

> > ### Comment · Reviewer_P7w2 · 2024-11-13
> >
> > Thank you for your questions. Let me clarify.
> >
> > By “orthogonal,” I mean uncorrelated—like vector orthogonality. My concern is that the performance gains you’re seeing might be due to decorrelating representations of different classes at each layer and not due to decorrelating them with the error signal.
> >
> > This came up because I had trouble following the logic of applying the reverse of the theorem about error signal orthogonality.
> >
> > To check this, you might compare your method to a baseline that decorrelates class representations at each layer using only the class labels without involving the error signal. I don’t have a specific algorithm in mind, but this could help show whether the key factor is the decorrelation with the error signal or just between classes.
> >
> > I hope this helps.

---

> > > ### Author Response · Authors · 2024-11-14
> > >
> > > Thanks a lot. We appreciate the clarification you provided.

---

> ### Author Response · Authors · 2024-11-18
> **Response to Reviewer P7w2's Comments**
>
> > Strengths: Compelling Biological Motivation, significant promise of error-broadcasting, improved performance, intriguing theoretical foundation, showcase practicality
>
> We would like to thank the reviewer for all of these positive comments about the novel contributions of our article.
>
> > Weakness 1 and Question 1: problematic reverse use of orthogonality principle.
>
> We appreciate the reviewer's comment, which give us an opportunity to clarify the main contribution point of our paper. In **linear** Minimum Mean Square Error (MMSE) estimation, the orthogonality principle states that the estimation error is orthogonal to the observations and their linear functions. Mathematically, this is expressed as $E(\epsilon_* \mathbf{x}^T)=\mathbf{0}$. Using this orthogonality condition in reverse to derive linear estimators is a standard practice in the field (see, for example, Kailath et.al. 2000) . Techniques like Kalman Filtering are based on this principle, which is firmly grounded in the Hilbert space projection theorem.
>
> For nonlinear MMSE estimation, the orthogonality condition is even stronger: the estimation error is orthogonal to any nonlinear function of the input. Exploiting this stronger condition to construct nonlinear MMSE estimators is an open problem, primarily because it raises questions about which nonlinear functions to choose and how many are needed.
>
> In our work, we model the neural network as a parameterized nonlinear MMSE estimator and seek as many equations from the orthogonality principle as possible to determine these parameters. This is exactly the same principle as how the orthogonality condition is used in reverse to find parameters for linear estimators. To address the challenge of selecting nonlinear functions that yield informative equations for determining network parameters, we choose the activations of the hidden layers in the neural network as these functions. This choice is natural because these activations are directly related to the network's parameters through differentiation. By enforcing orthogonality between the estimation error and the hidden layer activations, we create a framework that effectively guides the learning process. Our empirical results support the feasibility and advantages of this approach in training neural networks.
>
> Further refinement of this methodology—such as integrating additional nonlinear functions of the layer activations and inputs—not only holds significant potential for enhancing the model's capabilities but also represents an exciting avenue for future research.
>
> To enhance clarity on this matter, we have revised Section 2.2 to include the clarifications discussed above. We hope that, based on the reviewer's suggestion, the article now better positions the proposed method.
>
> Reference:
> Kailath, Thomas, Ali H. Sayed, and Babak Hassibi. Linear estimation. Prentice Hall, 2000.
>
> > Weakness 1 part 2: the hidden layers is very large, a solution to be orthogonal to a single direction is weak
>
> We believe there is a misunderstanding regarding our use of the term "orthogonality" and how it applies within the context of our methodology, especially in high-dimensional hidden layers.
>
> In our paper, "orthogonality" refers to the statistical condition where two scalar random variables are uncorrelated. When we state the orthogonality between a hidden layer vector and the output error vector, we are implying that each unit in the hidden layer is uncorrelated with each component of the error vector. This is not a constraint to a single direction in a high-dimensional space. Contrary to that, it is a set of multiple orthogonality conditions applied element-wise between hidden layer activations and error components. In fact, for a hidden layer with $𝑛$ units and an error vector of dimension $𝑚$, there are $n\times m$ orthogonality consraints. As the hidden layer size scales, so does the number of orthogonality constraints, ensuring that the learning process remains robust and well-defined in high-dimensional settings. By imposing these multiple orthogonality conditions, we are not facing dimensional degeneracy or a lack of directional information. Instead, we are enhancing the network's ability to learn meaningful representations by ensuring that each hidden unit contributes uniquely to reducing the output error.
>
> The success of our numerical experiments substantiates the effectiveness of our methodology. The performance improvements we observe are directly attributable to the enforcement of these orthogonality conditions across all hidden units and error components.

---

> > ### Author Response · Authors · 2024-11-18
> > **Response to Reviewer P7w2's Comments (Part 2)**
> >
> > >Weakness 2 and Question 2: whether the performance is due to layerwise orthogonal class representations (rather than decorrelation)
> >
> > In reference to our response to the previous point, (1), the learning capability of the proposed method is the direct application of the MMSE orthogonality condtion. It is a direct extension of the use of orthogonality condition in linear MMSE settings. In order to test the reviewer's hypothesis about layerwise clas orthogonal representations, we performed the following numerical experiment: We calculated the average cosine similarity between the representation vectors of different classes at each layer after training. This analysis was performed when network is trained with both the standard BP algorithm and our proposed EBD algorithm on the CIFAR-10 dataset with a CNN architecture.
> >
> > The results are summarized in Table A below:
> >
> > Table A: Average Inter-Class Cosine Similarities at Different Layers. Note that the larger the cosine similarity, less orthogonal the vectors are.
> > |Algorithm | Layer 1  | Layer 2  | Layer 3  | Layer 4  |
> > | :--------: | :--------: | :--------: | :--------: | :--------: |
> > | Random Initialization | 0.4005   | 0.5537    | 0.6566    | 0.5537   |
> > | After EBD Training      | 0.3918   | 0.5953   | 0.6734   | 0.8601   |
> >
> > Based on Table A above, we can conclude that the EBD training does not orthogonalize class representations (actually it may increase the alignment of class representations).  Thus, the performance gains cannot be attributed to this factor.
> >
> > Instead, the results support our theoretical premise that enforcing the MMSE orthogonality condition—specifically, the orthogonality between the estimation error and the hidden layer activations—effectively guides the learning process. By directly applying this principle, our method enhances the network's ability to minimize estimation errors, leading to better overall performance.
> >
> > Regarding the suggestion to include a comparative baseline using local class orthogonalization with an SGD-trained readout on the penultimate layer, we agree that this could provide additional insights. However, given the evidence from our experiment that orthogonal class representations are not the primary driver of performance improvements, we believe that our current results sufficiently support our proposed mechanism.
> >
> > > Weakness 4. lack of clarity: entropy and normalization originality, unclear presentation: forward projections
> >
> > - Regarding the comments of the reviewer on entropy and power normalization:  we do not consider them to be the key contributions of our work, although they nicely work out as structured and systematic solution to  potential collapse problem associated with the application the EBD rule.
> >
> > To clarify this issue in Section 2.4.1 of the revised article, we have added the following statement:
> >
> > *"We note that, while the use of entropy and power regularizers may not be entirely novel, they play a significant role in preventing the collapse problem."*
> >
> > - Regarding the comments about error broadcasting: we agree with the reviewer that the subsection on forward broadcasting was brief in our initial submission, which was mainly due to the length constraints of the article. In the revised article, we expanded Appendix B.3 to include more discussion on the motivation, workings and the its connection to biological networks. If we quote from this new Appendix section: *"The purpose of forward broadcasting is to enhance the network's ability to minimize the decorrelation loss by directly influencing the final layer's weights using the activations from the hidden layers... This mechanism allows the final layer to update its parameters in a way that reduces the correlation between the output errors and the hidden layer activations. Consequently, the errors at the output layer are steered toward being orthogonal to the hidden layer activations. While the proposed forward broadcasting mechanism is primarily motivated by performance optimization, it can conceptually be related to the  long-range (Leong et al., 2016) and bottom-up (Ibrahim et al., 2021) synaptic connections in the brain, which allow certain neurons to influence distant targets. These long-range bottom-up connections are actively being researched, and incorporating similar mechanisms into computational models could enhance their alignment with biological neural processes. By integrating mechanisms that mirror these neural pathways, forward broadcasting may be useful for modeling how information is transmitted across different neural circuits."*
> > We hope this newly expanded section in the revised article mostly addresses the concerns of the reviewer.
> >
> > References:
> >
> > Leong, A , et al. "Long-range projections coordinate distributed brain-wide neural activity with a specific spatiotemporal profile." PNAS (2016)
> >
> > Ibrahim, LA, et al. "Bottom-up inputs are required for establishment of top-down connectivity onto cortical layer 1 neurogliaform cells." Neuron (2021)

---

> > > ### Author Response · Authors · 2024-11-18
> > > **Response to Reviewer P7w2's Comments (Part 3)**
> > >
> > > > Weakness 3-Question 3: biological relevance is weakened by the batch learning requirement
> > >
> > > We would like to clarify and elaborate on how our work addresses these concerns by incorporating an online, more biologically plausible method:
> > >
> > >  - In our article, we employ batch optimization-based learning in numerical experimens for MLP, CNN and LC  implementations. The entropy and power normalization regularizers used in these numerical experiments are computed using batch calculations.
> > >  - However , In Section 3, titled "Error-Based Learning in a Biologically More Realistic Network", we introduce our **biologically more plausible network framework**:
> > >      -  We adopt the CorInfoMax network architecture from Bozkurt et al. (2023), but with a crucial modification: we replace the equilibrium propagation-based training with our proposed EBD training algorithm.
> > >      -   In Bozkurt et al. (2023), the CorInfoMax network is derived from the **online optimization** of the correlative information objective, which is naturally using online learning and aligns well with biological reality.
> > >      -   The **layer entropies** are part of this   optimization obtjective, and these entropies are optimized online through lateral (recurrent) connections among layer units, eliminating the need for batch-based entropy calculations.
> > >      -   The power normalization can be enforced with batch of size 1 by adjusting the learning rate.
> > >
> > >     -   Feedforward and feedback weights are updated using three factor learning rule, due to EBD updates and Hebbian learning rules, which are widely considered biologically plausible. Synaptic weights of recurrent (lateral) connections are updated using an anti-Hebbian learning rule, further enhancing the biological realism of the network.
> > >      -   All updates in Algorithm 1 (CorInfoMax-EBD) are designed to be executed online, using a single input per training update (i.e., a batch size of $1$). This means that the algorithm does not require retaining and normalizing entire batches at each layer, directly addressing the reviewer's concern about batch learning undermining biological plausibility. While the algorithm is extendable to batch data if needed, its fundamental design supports online learning, which is more consistent with biological learning mechanisms.
> > >      -   CorInfoMax-EP approach in Bozkurt et al. (2023) uses equilibrium propagation learning rule which relies on a two-phase update, time-contrastive process that raises questions about its biological realism. Our version, CorInfoMax-EBD, uses EBD learning rule, which allows for single-phase updates without the need for separate phases.
> > >      -   Our new numerical experiments for CorInfoMax-EBD confirm that with a batch size of $1$, the network can still achieve satisfactory performance. For instance, on the CIFAR-10 dataset, CorInfoMax-EBD with a batch size of $1$ reaches an accuracy of over $53.4%$, compared to the CorInfoMax-EP of Bozkurt et al., 2020, which achieves $50.97%$ accuracy with a batch size of $20$. We are currently conducting further hyperparameter tuning for both the MNIST and CIFAR-10 datasets.
> > >
> > > Below is the updated version of Table 3 from our article, listing the current CorInfoMax-EBD accuracy values for a batch size of $1$.
> > >
> > > Table 3: Accuracy results (%) for EP and EBD CorInfoMax algorithms. Column marked with [x] is
> > > from reference Bozkurt et al. (2024).
> > > |Data Set |CorInfoMax-EP[x]  (batch size:20) | CorInfoMax-EBD (Ours) (batch size:20) | CorInfoMax-EBD (Ours)(batch size:1) |
> > > |:-------: | :--------: | :--------: | :--------: |
> > > | MNIST  | 97.58     | 97.53   |   94.7   |
> > > |CIFAR-10 |  50.97   | 55.79   |   53.4   |
> > >
> > > In summary, the CorInfoMax-EBD approach presented in (Section 3 of) our paper offers an online, biologically plausible framework that does not depend on batch learning. By utilizing three-factor lerarning as a consequence of the application of EBD method, together with Hebbian and anti-Hebbian learning rules and eliminating the need for batch-based regularizers, we address the feasibility of an online, biologically plausible approach as suggested by the reviewer.
> > >
> > > To address the concerns of the reviewer, and to clarify this issue further in Section 3.2 of the revised article, we have added the following explanation:
> > > *"The CorInfoMax-EBD scheme proposed in this section is more biologically realistic than the MLP-based EBD approach in Section 2 due to multiple factors: Unlike the batch-mode operation required by the MLP-based EBD, CorInfoMax operates in an online optimization setting which naturally integrates entropy gradients into lateral weights, resulting inbiologically plausible updates, whereas the MLP approach uses entropy regularization without ensuring biological plausibility. Besides, it employs a neuron model and network architecture that closely mirror biological neural networks."*

---

> > > > ### Comment · Reviewer_P7w2 · 2024-11-22
> > > >
> > > > Thank you for your elaborate and careful response. I may be lacking some necessary background, but since I believe I am a typical representative of the ICLR audience, I feel I must press on this point since you did not directly answer my concern about applying the orthogonality principle in reverse. Perhaps my question was not clear enough.
> > > >
> > > > 1.
> > > > In your answer, you brought up examples of using this method in linear systems and Gaussian noise, such as the Kalman filters, and argued that the same method can be extended to any nonlinear function of the input. I have no problem with that, and I think it is an interesting and original observation.
> > > >
> > > > My problem is applying the reverse in a very high-dimensional space. In neural networks, the number of parameters is typically much larger than the number of samples. This difference is even more striking when you consider batch learning. **Finding the optimal estimator by orthogonality to the estimation error is a highly underconstrained problem in this regime.** While it probably converges at the limit of infinitely many samples or learning time, I could not follow the logic of the learning rule. Are you assuming you are in the statistical regime where the number of samples is as large as the number of parameters? **Are there additional constraints or implicit regularization at play here?**
> > > >
> > > > 2.
> > > > Thanking for taking the time to provide more numerical evidence. The classes' average representation does not decorrelate in the trained networks, as I suggested as an alternative mechanism. This analysis eases my concerns that the main change was due to simple decorrelations of the classes independently in each layer. However, the concerns that stem from my misunderstanding of applying the orthogonality principle are not lifted by this analysis alone.
> > > >
> > > > 3.
> > > > You write, “*We note that, while the use of entropy and power regularizers may not be entirely novel, they play a significant role in preventing the collapse problem.”* I argue that preventing collapse using entropy methods is straightforward and not insightful. However, I accept that this is not the main contribution of this work and is more of a side note.
> > > >
> > > > 4. Thank you for your explanation about the biological plausibility of batch learning. The term “biologically plausible” is a little vague, and most neuroscientists will probably strongly argue your learning rule is not biological. However, I believe it is a sufficient claim here, as your study does not claim to solve learning in the brain.
> > > >
> > > > I am increasing my overall rating to 5. While you addressed most of my questions, my main concern about the logic and correctness of applying the orthogonality principle in a large neural network still remains. However, I am also reducing my confidence in my response—there may be something that I am missing.

---

> ### Author Response · Authors · 2024-11-23
> **Response to Reviewer P7w2's New Comments**
>
> Thank you for your kind and detailed follow-up. We very much appreciate your thoughtful critique.
>
> > Point-1 and Point-2: Applying the orthogonality principle in reverse
>
> Thank you for restating your question; it has helped clarify your perspective on our side. We indeed agree that in neural networks, the number of parameters typically exceeds the number of samples, leading to a system that is underconstrained in a classical sense. To directly address your question, **our method does not assume that the number of samples matches the number of parameters**.
>
> In our paper, we define a decorrelation condition involving the number of neurons multiplied by the number of outputs. The orthogonality principle in our method is defined as the uncorrelatedness of a given hidden layer neuron’s activation with all components of the output error separately. Specifically, for  the $i^{th}$ neuron of layer-$k$  and  the $j^{\text{th}}$ component of the output error, $\epsilon_j$, the orthogonality condition is expressed as,
> $$ R_{h_i^{(k)}\epsilon_j}=E(h_i^{(k)}\epsilon_j)=0, \hspace{0.2in} j=1, \ldots m, \hspace{0.1in} i=1, \ldots n \hspace{0.2in} (\text{Eq.A})$$
> where $m$ is the number of output components, and $n$ is the size of activations for layer $k$.
>
> Based on the (Eq.A) above, for **each hidden layer, there are $m$ x $n$ orhogonality constraints**. Therefore, **even if the hidden layer dimensions and/or the network depth increase, the total number of constraints also increases**, making our system less underdetermined. We use constraints for different neurons separately to adjust their corresponding weight/bias parameters. In other words, the constraints in (Eq. A) are used to update the $i^{\text{th}}$ row of $\mathbf{W}^{(k)}$, i.e. $\mathbf{W}_{i,:}^{(k)}$, and the bias compoent $b_i$. To show this numerically, for a linear layer with 1024 units and error dimension of 10, there are 1024x10 = 10240 orthogonality conditions, and for a linear layer with 4096 units and error dimension of 10 there are 4096x10=40960 orthogonality conditions.
>
> Note that, more generality of the orthogonality condition for nonlinear estimators offers potential to increase the number of constraints per hidden layer neuron: We can increase the number of the orthogonality conditions per neuron even further by considering the fact that uncorrelatedness requirement is  for any function $g$ of hidden layer neuron activations, i.e.,
> $$ R_{g(h_i^{(k)})\epsilon_j}=E(g(h_i^{(k)})\epsilon_j)=0, \hspace{0.2in} j=1, \ldots m \hspace{0.2in} (\text{Eq.B})$$
> Therefore, the number of uncorrelatedness (orthgonality) constraints per hidden layer/output neuron can be increased by introducing multiple $g$ functions. (However, in our numerical experiments we haven't pursued this path.)
>
> Although the **orthogonality conditions scale with the increasing parameter size**, the total number of parameters in the network is in general larger than the number of decorrelation conditions. This results in fewer constraints than parameters, leading to an overparameterized system, where a unique optimal estimator cannot be determined solely based on these conditions. Your concern about infinite samples or learning time in overparameterized networks is valid, but **our results show that the learning rule converges effectively within practical timeframes**. Particularly in the case of using Locally Connected Networks (LC) (Refer to Table-1,2 in the paper), which are highly overparameterized, the improved performance and generalization observed strongly validate the practicality of our approach to successfully train in the overparametrized regime.
>
> Importantly, **this issue of overparameterization also exists in standard backpropagation**, where the number of parameters often exceeds the number of training samples, leading to an overparameterized and underdetermined system. In both cases, this overparameterization does not hinder learning; rather, it is a fundamental characteristic of deep learning. Research has demonstrated that the **implicit bias in gradient descent** introduces a regularization effect, steering the optimization process toward solutions that generalize well to unseen data (Soudry et. al, 2018).
>
> **Reference:** Soudry, D., Hoffer, E., Nacson, M. S., Gunasekar, S., & Srebro, N. (2018). The implicit bias of gradient descent on separable data. ICLR 2018

---

> ### Author Response · Authors · 2024-11-23
> **Response to Reviewer P7w2's New Comments (Part 2)**
>
> Similarly, in our method, while the number of orthogonality constraints is smaller than the total number of network parameters, the system is guided by the statistical properties of the error and activations. While we cannot claim to fully characterize the implicit regularization effect in our method, we suggest that these statistical constraints play a role similar to the implicit regularization observed in regular backpropagation. This helps ensure that the learned parameters are not arbitrary but are shaped by the decorrelation principles inherent to our framework, contributing to the model’s generalization capabilities. We believe that investigating the inherent implicit bias in Error Broadcast and Decorrelation (EBD) opens the door to further understanding how this framework naturally regularizes the learning process.
>
> We would also like to highlight that our method imposes even more constraints than Direct Feedback Alignment (DFA), where the feedback weights are fixed matrices. In contrast, EBD learns these matrices directly from the data as the cross-covariance matrices between the error and hidden layers, making it more scalable and data-dependent, even in the finite data regime. **Considerig our EBD mechanism as a generalization of the DFA framework, it becomes clearer that the orthogonality condition is neither simplistic nor underconstrained.** Because when we turn off the update on the cross-covariances, and leave it to initialization, it becomes exactly equal to DFA method (apart from entropy max. and power norm.).
>
> To further adress limited-data problems, our method incorporates several regularization techniques:
> - **Entropy regularization:** Encourages the network to utilize the full feature space by spreading activations.
> - **Sparsity regularization:** Enforces sparse activations to reduce redundancy.
> - **Weight decay:** Prevents overfitting by penalizing large weights.
> These regularizers supplement the orthogonality-based learning rule, particularly in the limited-data regime, improving generalization and stability.
>
>
> Based on your suggestions, to clarify this point, in the **new revision** of our article **we included a new appendix section** (Appendix E.9), with the title "*On the scaling of the orthogonality conditions*", discussing how the number of orthogonality conditions increase with the increasing model size, and the possible implicit and the existing explicit regularizations on our algorithm.
>
> > Point-3: The use of entropy and power regularizers
>
> Thank you again for raising this important point. We think clarifying this also in the main paper increased the clarity of our main contributions.

---

> ### Author Response · Authors · 2024-11-23
> **Response to Reviewer P7w2's New Comments (Part 3)**
>
> > Point-4: On the vagueness of biological plausibility
>
> Thanks for this comment. It is true that the term biologically plausible is somewhat vague; this partly stems from the fact that what constitutes a biologically plausible learning rule is not fully settled in neuroscience. Here, we adopted the following principles which are largely accepted among neuroscientists:
>
> - **Local Learning Rules:** Our method uses the orthogonality principle to enforce local relationships between the estimation error and hidden layer activations. This locality mimics the way biological systems may rely on local interactions for learning, as opposed to requiring global error signals.
>
> - **Three Factor Learning Rule:** There exists several experimental resutls supporting the existence of three factor learning mechanism in biological networks. Furthermore, our EBD framework provides a principled derivation of this rule.
>
> - **Training without weight symmetry:** Our method avoids the biologically implausible assumption of symmetric feedforward and feedback weights. Instead, it dynamically learns feedback matrices, mirroring the independence of connections seen in biological neural networks, unlike regular backpropagation.
>
> - **Lateral, feedforward  and feedback connections:** Experimental studies indicate that biological neural networks incorporate feedforward, feedback, and lateral connections among neurons. Our CorInfoMax-EBD framework provides a principled method for designing neural networks with these connectivity patterns, where output errors are propagated through adaptable feedback connections.
>
>
>
> - **More Realistic Neuron Model:** Our method parallels the three-compartment neuron model, where dendrites receive input signals, the soma processes these signals, and axons transmit output signals. In our method, the hidden layer activations represent the dendritic computations, while the feedback connections act as modulatory signals akin to axonal inputs in biological neurons, just like our CorInfoMax-EBD model.
>
> - **Online learning:** As you also pointed out, batch learning does not closely mirror learning in biological systems, which typically process data in an online and sequential manner.
>
> Our contribution, as the reviewer correctly points out, is not solving learning in the brain, but to provide a learning rule that is both mathematically principled and aligned with widely agreed-upon biological principles as much as possible.

---

> > ### Comment · Reviewer_P7w2 · 2024-11-27
> >
> > Thank you for explaining your use of the orthogonality principle. After careful consideration, I remain unconvinced by your arguments. Applying the orthogonality principle on a per-neuron basis appears to be incorrect. Therefore, the assertion that you have  $m \times n$  conditions is flawed, and the problem of finding the optimal estimator remains highly under-constrained.
> >
> > Your numerical analysis shows that your learning rule functions and presents an interesting idea. However, the theoretical foundation relies on problematic assumptions, making it unclear what your complex and elaborate learning dynamics actually achieve. Specifically, you rely on a theorem that provides **necessary** conditions and treat it as if it offers **sufficient** conditions for optimality. I do not see how this holds in the high-dimensional regimes characteristic of neural networks.

---

> ### Author Response · Authors · 2024-11-27
> **Response to Reviewer P7w2's New Comments (Part 1)**
>
> We would like to thank the reviewer for their active engagement during the discussion period. Below, we provide our responses to the reviewer's comments:
>
> > Applying the orthogonality principle on a per-neuron basis appears to be incorrect. Therefore, the assertion that you have  $m \times n$  conditions is flawed, and the problem of finding the optimal estimator remains highly under-constrained.
>
> We would like to clarify that applying orthogonality condition per-neuron basis is correct:
>
> (i). First, we need to clarify the terminology: in this context, "orthogonality" refers to statistical uncorrelatedness. The geometric term "orthogonality" is defined within a Hilbert space of random variables, where the inner product between two random variables  $a,b$ is defined as
>
> $\langle a , b \rangle \stackrel{\Delta}{=} E(ab)$.   (Eq. A)
>
> That is the inner product, which is defined as the expected value of their product, i.e., their correlation. Two random variables $a,b$ are said to be orthogonal if their inner product is zero:
>
> $\langle a , b \rangle {=} E(ab)=0$. (Eq. B)
>
> Therefore, two random variables are called orthogonal if their correlation is zero (see, for example, (Kailath et al., 2000), Chapter 3.)
>
> In our framework, each individual neuron in a given layer and each error component are modeled as random variables, with realizations varying across dataset samples. Defining the correlation on a per-neuron basis is thus appropriate and well-grounded in statistical theory.
>
> ---
> (ii). Second, we clarify the orthogonality theorem related to nonlinear minimum mean square error (MMSE) estimation. As stated in our article (and proved in Appendix A), the MMSE orthogonality condition can be expressed in more detail (Papoulis & Pillai, 2002):
>
> Let $\hat{\mathbf{y}}\in \mathbb{R}^p$ be the optimal nonlinear MMSE estimate of the desired vector $\mathbf{y}\in \mathbb{R}^p$  given the input $\mathbf{x} \in \mathbb{R}^m$. Let $\mathbf{e}_*=\mathbf{y}-\hat{\mathbf{y}}$ denote the corresponding estimation error. Then, for any poperly measurable function $\mathbf{g}(\mathbf{x})\in \mathbb{R}^q$ of the input, we have
>
> $E(\mathbf{g}(\mathbf{x}){\mathbf{e}_*}^T) = \mathbf{0}\_{q \times p}$.  (Eq. C)
>
> In other words, the cross correlation matrix of  the nonlinear function of the input, $\mathbf{g}(\mathbf{x})$ and the output error for the best MMSE estimator, $\mathbf{e}_*$ is equal to a $q \times p$ zero matrix. The matrix equation in (Eq. C), can be more explicitly written as $q\cdot p$ orthogonality conditions:
>
> $E(g_i(\mathbf{x}){e_*}_j)=0$ for $i=1, \ldots, q$ and $j=1, \ldots, p$. (Eq. D)
>
> In other words, the correlation between each component of $\mathbf{g}(\mathbf{x})$ and each component of $\mathbf{e}_*$ is equal to $0$. Using the Hilbert space terminology from item (i), the random variables $g_i(\mathbf{x})$ and $e\_{*j}$ are "orthogonal" to each other for any choice of $i\in \{1, \ldots, q\}$ and $j \in \{1, \ldots, p\}$.
>
> ---
> (iii). In our framework, we pose a neural network as a MMSE nonlinear estimator. Let $\mathbf{e}\in \mathbb{R}^p$ represent the output error (when there are $p$ outputs, e.g., $p=10$ for CIFAR 10) for the neural network that corresponds to the optimal MMSE estimator. Then we can pick the arbitrary nonlinear functions of the input as hidden layer activations $\mathbf{h^{(k)}}\in\mathbb{R}^{N^{(k)}}$, where $N^{(k)}$ is the number of neurons in hidden layer $k$. Then by (Eq. D) above
>
> $E(h^{(k)}_i(\mathbf{x})e_j)=0$  for $i=1,\ldots, N^{(k)}$ and $j=1, \ldots, p$. (Eq.E)
>
> In other words, the correlation between the random variables $i^{\text{th}}$ neuron of layer $k$, i.e., $h_i^{(k)}$ and the $j^{\text{th}}$ component of the output error $e_j$ is zero.  So, using the terminology defined in item (i) above, we call that the random variables $h_i^{(k)}$ and $e_j$ are "orthogonal" to each other (for any choice of $i \in \{1, \ldots, N^{(k)}\}$ and $j \in \{1, \ldots, p\})$.
>
> Consequently, for each hidden layer neuron activation $h_i^{(k)}$, there exists $p$ orthogonality conditions, which we can write
>
> $E(h^{(k)}_i(\mathbf{x})e_j)=0$  for  $j=1, \ldots, p$. (Eq.F)
>
> As a result, for the hidden layer $k$, where there are $N^{(k)}$ neurons, with $p$ orthogonality conditions per neuron, there are $N^{(k)} \cdot p$ orthogonality conditions in total. Therefore, there is no error or flaw in our arguments in our paper or responses.
>
> ---
> We believe the potential misunderstanding may stem from confusing the orthogonality of random variables (defined via the correlation inner product in item (i)) with the orthogonality of Euclidean vectors in $\mathbb{R}^{n}$, defined by the standard Euclidian inner product
>
> $\langle \mathbf{j},\mathbf{r} \rangle=\mathbf{r}^T\mathbf{j}$. (Eq. G)
>
> **If the reviewer still believes there is an issue, we kindly request a precise indication of which item, equation, or statement above is of concern so we can provide further clarification.**

---

> > ### Author Response · Authors · 2024-11-27
> > **Response to Reviewer P7w2's New Comments (Part 2)**
> >
> > > the theoretical foundation relies on problematic assumptions, making it unclear what your complex and elaborate learning dynamics actually achieve.
> >
> > We believe this point is clarified by the explanations above.
> >
> >  > you rely on a theorem that provides necessary conditions and treat it as if it offers sufficient conditions for optimality. I do not see how this holds in the high-dimensional regimes characteristic of neural networks.
> >
> > This comment, we believe, is mainly based on the previous misunderstanding. Regarding the use of orthogonality conditions to train network, we can provide the following clarification:
> >
> > Each neural network functions as a parametric estimator. Therefore, finding the optimal network  is equivalent to performing a finite-dimensional parameter search. To determine these parameters, we require multiple equations to constrain their values. The orthogonality conditions in (Eq. F) provide such equations, and this approach is indeed the one pursued in MMSE linear estimation and adaptive algorithms. However, as we stated in our previous response, the orthogonality conditions in (Eq. F) do not provide a sufficient number of equations, leading to an underdetermined system.
> >
> > We also mentioned that there are potential implicit regularization effects, such as stochastic gradient descent (SGD)-based norm regularization (Soudry et. al, 2018), in addition to  the explicit the entropy and activation sparsity regularizers.
> >
> > To address the insufficiency in the number of orthogonality conditions, we can leverage the generality of the nonlinear MMSE orthogonality condition in (Eq. C). Specifically, we can introduce several nonlinear functions of the hidden activations, such as $g^{(u)}(h_i^{(k)})$, for $u=1, \ldots, U$, where $U$ is the number of nonlinearities per hidden unit. By incorporating these nonlinear mappings, and using the general orthogonality condition in (Eq. C), we can extend (Eq. F) as follows:
> >
> > $E(g^{(u)}(h^{(k)}_i(\mathbf{x}))\mathbf{e}_j)=0$  for  $j=1, \ldots, p$, $u=1, \ldots, U$. (Eq.H)
> >
> > This would introduce additional $U \cdot p$ orthogonality conditions per neuron in the hidden layer, increasing the total number of orthogonality conditions for layer-$k$ to $U \cdot N^{(k)}\cdot p$. Of course, The choice of these nonlinear functions is a subject for further research and algorithmic development.
> >
> > However, we did not need to pursue this path in our numerical experiments. Our results suggest that the implicit and explicit regularizers were sufficient in guiding the gradient-based optimization toward desirable solutions in the optimization landscape. This indicates that even without increasing the number of orthogonality conditions through additional nonlinear functions, our approach remains effective. This is in the same vein as the behavior observed in neural networks trained with backpropagation in an overparameterized regime, where training leads to useful estimators or classifiers mainly due to implicit (Soudry et. al, 2018) and explicit regularization effects, as we stated in our previous response.
> >
> > References:
> > - Kailath, T. et al., Linear estimation. Prentice Hall, 2000.
> > - Papoulis A. et al., Probability, Random Variables, and Stochastic Processes. McGraw-Hill, 2002
> > - Soudry, D., Hoffer, E., Nacson, M. S., Gunasekar, S., & Srebro, N. (2018). The implicit bias of gradient descent on separable data. ICLR 2018.

---

> > > ### Author Response · Authors · 2024-12-01
> > > **Final Follow-Up Before Discussion Period Ends**
> > >
> > > We would like to thank you for your engagement and valuable feedback. As the discussion period concludes tomorrow, we wanted to ensure that our revisions and responses have fully addressed your comments and concerns. If there’s any additional information or clarification you need, please let us know—we would be happy to provide it. We appreciate that all reviewers recognized the novelty of our work, and we have thoughtfully integrated your suggestions to strengthen the manuscript. In its current form, we believe our article presents a novel learning paradigm grounded on an estimation-theoretical approach, with strong potential for impact on both neuroscience and machine learning, now with significantly improved clarity and depth.

---

### Official Review · Reviewer_qRs4 · 2024-11-04

**Soundness:** 3
**Presentation:** 2
**Contribution:** 2
**Rating:** 5
**Confidence:** 4

**Summary:**

This paper proposes a new learning framework for neural networks that directly broadcasts output error to individual layers. The main idea is to minimize the correlation between the layer activations and output errors, which is based on the orthogonality property of minimum mean square error estimators developed by Papoulis&Pillai 2002 [1]. The framework is implemented on MNIST and CIFAR10 benchmark tasks for MLP and CNN.

**Strengths:**

1.	The paper proposes a novel idea that uses the orthogonality property of the optimal MMSE, which avoids weight symmetry problem in conventional backpropagation.
2.	Compared with direct feedback alignment (DFA) method, the proposed EBD method provides better theoretical illustration and the results in MNIST and CIFAR-10 tasks are better.

**Weaknesses:**

1.	The method still faces the critical issue of scaling up, as most of non-BP learning frameworks exist.
2.	The experiments show EBD is only slightly better than DFA, while it is not comparable with other SOTA biologically plausible methods (e.g. Hebbian base method [2])

**Questions:**

See the above weaknesses. Moreover:

1.	Why does BP in Table 2 show such a low performance? Did the authors try to use a different CNN architecture to get a better performance?
2.	The MMSE estimator-based derivation looks great, but in terms of network training, is optimal MMSE estimator the best objective of an arbitrary defined network? (since different tasks might have different loss functions)

Ref:

[1] Athanasios Papoulis and S Unnikrishna Pillai. Probability, Random Variables, and Stochastic Processes. 2002

[2] Journe et.al., Hebbian deep learning without feedback, ICLR 2023

---

> ### Author Response · Authors · 2024-11-18
> **Response to Reviewer qRs4's Comments**
>
> > Strengths: novel idea, the orthogonality property, avoids weight symmetry, better theoretical ill. & performance than DFA.
>
> We would like to thank the reviewer for the positive assessment about the main contributions of our article.
>
>
>
> > Weakness 1: critical issue of scaling up, as most of non-BP learning frameworks exist
>
> Our main contribution lies in offering a fresh theoretical backing for the error broadcast mechanism, which is believed to be a key process in biological learning, especially through the involvement of neuromodulators. While it is true that most biologically plausible frameworks face scaling issues, recent work by Launay et al. (2020) demonstrates that algorithmic enhancements can enable error broadcast approaches like Direct Feedback Alignment (DFA) to scale to large-scale, complex problems.
>
> Our framework is essentially similar to DFA, with the crucial distinction that our broadcasting weights are adaptive rather than fixed. Given these similarities with DFA and the demonstrated scalability in Launay et al.'s work, we believe that our proposed method can likewise be extended to train more complex network structures effectively.
>
> Ref:
> Launay, Julien, et al. "Direct feedback alignment scales to modern deep learning tasks and architectures." Advances in Neural Information Processing Systems 33 (2020): 9346-9360.
>
> > Weakness 2.  experiments: slightly better than DFA, not comparable with SOTA Hebbian
>
> While the performance on a machine learning tasks is an important indicator, it may not be the only or  most significant metric in evaluation of learning frameworks. In biologically realistic models, capturing more realism and providing mathematical explanations for the existing phenomena are also important criteria. The proposed error-broadcast decorrelation may not achieve state of the art performance, with the existing choice of architecture and parameter settings, however, numerical experiments demonstrate that it performs on par or better than the existing biologically plausible error-broadcast approaches (Note that Journe et al,, 2023 is not a error broadcasting approach.). Due to its groundedness on the estimation theoretic orthogonality principle, its potential expressive power for the mathematical modelling of error-broadcast based learning through neuromodulators holds a signifcant value. Furthermore, the proposed bio-plausible CorInfoMax-EBD approach captures several features of the biological neural networks such as lateral connections, feedforward and feedback connections, multi-compartment neuron models and three factor update rule as a result of the application of the EBD method.
>
> > Question 1. why low BP performance in Table 2
>
> For Table 2, we used results (including BP) from Clark et al. (2021) for all algorithms except our EBD algorithm. To clarify this point and ensure consistency, we have now performed our own BP and DFA training implementations. For the revised experiments, we ensured that all methods (EBD, BP, and DFA) were trained for the same number of epochs specific to each architecture and dataset (for MLPs 120 epochs and for CNN/LC 100 epochs for MNIST and 200 epochs for CIFAR-10). We performed extensive hyperparameter tuning for BP and DFA, similar to what was done for EBD.
>
> The following is the updated version of Table 2 containing new simulation results for DFA, DFA with entropy regularization (DFA+E) and BP.
>
> Table 2: CIFAR-10 Dataset:
> [x]: values from Clark et al., 2021
> [ours]: our numerical experiments
> DFA+E: DFA with correlative entropy regularization
> |     | DFA [x] | DFA [ours] | DFA+E [ours] | NN-GEVB [x] | MS-GEVB [x]| BP[x] | BP [ours] | EBD [ours] |
> | :--------: | :--------: | :--------: |:-----:|:-----:|:-----:|:------:|:-----:|:---:|
> | MLP    |   50.46  | 52.09  | 52.22 | 52.38 | 51.14 | 55.31  | 56.37 | 55.17  |
> | CNN    |   55.93  | 58.39  | 58.56 | 66.26 | 61.57 | 71.2   | 75.24 | 66.42  |
> | LC     |   60.59  | 62.19  | 62.12 | 58.92 | 59.89 | 67.68  | 67.81 | 64.29  |
>
> According to this table, the BP values from our new experiments with optimized hyperparameters are higher than those reported in Clark et al. (2021). In this article, we used these new values in Table 2.

---

> > ### Author Response · Authors · 2024-11-18
> > **Response to Reviewer qRs4's Comments (Part 2)**
> >
> > > Question 2. MMSE  derivation looks great, but is MMSE best objective for arbitrary network?
> >
> > Our proposed framework exploits the nonlinear orthogonality property specific to MMSE (Minimum Mean Square Error) estimators. While the MMSE objective is a sensible choice for especially for regression problems, we acknowledge that different tasks potentially require different loss functions—such as cross-entropy.
> >
> > We are not currently aware of similarly powerful theoretical properties for loss functions like cross-entropy. This presents an intriguing opportunity for future research to explore whether analogous properties exist for other loss functions used in network training.
> >
> > However, our numerical experiments detailed in Appendix F.2 ("Correlation in Cross-Entropy Criterion-Based Training") show that even when training with the cross-entropy loss, we observe a similar decrease in correlation between errors and layer activations. This observation suggests that the decorrelation phenomenon might be a more general and fundamental aspect of the learning process, extending beyond the MMSE objective.
> >
> > Therefore, while the MMSE estimator may not always be the best objective for every network and task, the underlying decorrelation feature might still play a crucial role across different loss functions. We believe that further investigation into this area could yield valuable insights into the fundamental mechanisms of learning in neural networks.

---

> > > ### Comment · Reviewer_qRs4 · 2024-11-26
> > >
> > > I thank the authors for their detailed response. While I do think the paper proposed a very interesting idea, my original concerns still remain. Purely providing the paper on DFA (Launay et al NeurIPS 2020) does not automatically prove the current method can scale up to more complex tasks, I would suggest at least a task with larger scale should be tested (either some tasks as in Launay et.al. or more complex vision task like ImageNet).
> > >
> > > Regarding the accuracy on these machine learning tasks, I understand this is not the only indicator for a learning framework, but generally other indicators are only under discussion when the accuracy is above a threshold. I raise the example of Journe et.al on Hebbian learning is because my intuition is that an error broadcast method should perform better than such a pure local learning like Hebbian as the former has a (implicit) global error to guide the learning, maybe the authors could have different explanations?
> > >
> > > Overall, as other reviewers mentioned, either the authors should provide more evidences on the scale up capability of this method, or the authors should revise the current manuscript in a better narrative on the limitations and open-questions of this method.

---

> ### Author Response · Authors · 2024-11-26
> **Response to Reviewer qRs4's Comments**
>
> We appreciate the reviewer's constructive feedback and for acknowledging that our work presents an interesting idea. We understand the concerns regarding the narrative of our paper. In response to the reviewer's suggestions, we have revised the manuscript to better address the limitations and open questions of our method.
>
> >I raise the example of Journe et.al on Hebbian learning is because my intuition is that an error broadcast method should perform better than such a pure local learning like Hebbian as the former has a (implicit) global error to guide the learning, maybe the authors could have different explanations?
>
> Currently, unfortunately, we lack a rigorous explanation of how a feedback-free Hebbian approach—where hidden layer weights are trained using an unsupervised Hebbian mechanism and then frozen, and only the final layer is trained with a supervised mechanism—can achieve better performance than error feedback methods.
>
> >Overall, as other reviewers mentioned, either the authors should provide more evidences on the scale up capability of this method, or the authors should revise the current manuscript in a better narrative on the limitations and open-questions of this method.
>
> Our primary goal in this article is to present a novel theoretical framework for learning. This framework provides principled underpinnings for both error broadcast-based learning and three-factor learning—mechanisms that are believed to play crucial roles in biological networks. By rooting our approach in a major orthogonality property from estimation theory—the nonlinear orthogonality principle—we aim to lay down the foundational aspects of this new learning framework. This approach is in line with several recent works that focus mainly on deriving principled methods for biological and artificial learning mechanisms (e.g., Clark et al. (2021), Bozkurt et al. (2023), Dellaferrera, et al. (2022) and Kao et al. (2024)).
>
> To address your concern about the clarity of our claims regarding performance and scalability, we have carefully revised the manuscript to ensure it accurately reflects the intended scope. We emphasize that while scalability is an important future direction, our present work is centered on establishing a new learning mechanism based on the nonlinear orthogonality principle.
>
> In response to your feedback, we have revised key parts of the abstract, introduction, and conclusion to clarify that the current article does not focus on scalability. Instead, we highlight that scalability is a valuable extension of our proposed theoretical framework.
>
> Below, we describe the specific changes made in the latest revision to address your concerns:
>
> - **Abstract:** We have revised the abstract and added the bolded sentence and phrase:
>
>     *"We introduce the Error Broadcast and Decorrelation (EBD) algorithm, a novel learning framework that addresses the credit assignment problem in neural networks by directly broadcasting output error to individual layers. The EBD algorithm leverages the orthogonality property of the optimal minimum mean square error (MMSE) estimator, which states that estimation errors are orthogonal to any nonlinear function of the input, specifically the activations of each layer. By defining layerwise loss functions that penalize correlations between these activations and output errors, the EBD method offers a principled and efficient approach to error broadcasting. This direct error transmission eliminates the need for weight transport inherent in backpropagation. Additionally, the optimization framework of the EBD algorithm naturally leads to the emergence of the experimentally observed three-factor learning rule. We further demonstrate how EBD can be integrated with other biologically plausible learning frameworks, transforming time-contrastive approaches into single-phase, non-contrastive forms, thereby enhancing biological plausibility and performance. Numerical experiments demonstrate that EBD achieves performance comparable to or better than* **known error-broadcast methods** *on benchmark datasets.* **The scalability of algorithmic extensions of EBD to very large or complex datasets remains to be explored. However,** *our findings suggest that EBD offers a promising, principled direction for both artificial and natural learning paradigms, providing a biologically plausible and flexible alternative for neural network training  with inherent simplicity and adaptability that could benefit future developments in neural network technologies."*

---

> ### Author Response · Authors · 2024-11-26
> **Response to Reviewer qRs4's Comments (Part 2)**
>
> ---
> - **Introduction:** In the concluding paragraph of the introduction, we added a sentence to highlight scalability as an open question:
>
>     *"We demonstrate the utility of the EBD algorithm by applying it to both artificial and biologically realistic neural networks.* **While our experiments show that EBD performs comparably to state-of-the-art error-broadcast approaches on benchmark datasets, offering a promising direction for theoretical and practical advancements in neural network training, its scalability to more complex tasks and larger networks remains to be investigated.**"
>
> ---
> - **Related Work and Contribution:** In the final paragraph of Section 1.1, "Related Work and Contribution," we softened the language to temper claims about the advantages of our approach:
>
>     "*In summary, our approach provides a theoretical grounding for the error broadcasting mechanism and* **suggests ways to** *its effectiveness in training networks.*"
>
> ---
> - **Limitations:** Lastly, we revised the Limitations section and added the final sentence, highlighted in bold, to more clearly articulate our paper's position on scalability. This addition frames scalability as an important area for future exploration:
>
>     "**Limitations.** *The current implementation of EBD involves several hyperparameters, including multiple learning rates for decorrelation and regularization functions, as well as forgetting factors for correlation matrices. Although these parameters offer flexibility, they add complexity to the tuning process. Additionally, the use of dynamically updated error projection matrices and the potential integration of entropy regularization may increase memory and computational demands. Future work could explore more efficient methods for managing these components, potentially automating or simplifying the tuning process to enhance usability. Furthermore, while the scalability of EBD is left out of the focus of the article, we acknowledge its importance. Launay et. al. (2021) demonstrated that DFA scales to high-dimensional tasks like transformer-based language modeling. Since DFA is equivalent to EBD with frozen projection weights and without entropy regularization, we anticipate that EBD could scale similarly.* **However, this remains unvalidated empirically. Examining EBD’s scalability and streamlining its components to improve usability are important tasks for future work.**"
>
> ---
> In conclusion, we appreciate the reviewer’s feedback and have carefully addressed the concerns about narrative. The revised manuscript aims to present a more balanced perspective, clearly outlining the scope and limitations of this work. By framing scalability as a future direction rather than an immediate claim, we hope to make the paper’s contributions and positioning clearer while setting the stage for future research that builds on these findings.
>
> References:
> - Clark, et. al. "Credit assignment through broadcasting a global
> error vector.", Neurips, 2021.
> - Bozkurt, et. al. "Correlative information maximization: a biologically plausible approach to supervised deep neural networks without weight symmetry.", Neurips, 2023.
> - Kao et. al., "Counter-Current Learning: A Biologically Plausible Dual Network Approach for Deep Learning", Neurips, 2024.
> - Dellaferrera et al., "Error-driven Input Modulation: Solving the Credit Assignment Problem without a Backward Pass", ICML, 2022

---

> > ### Author Response · Authors · 2024-12-01
> > **Final Follow-Up Before Discussion Period Ends**
> >
> > We would like to thank you for your engagement and valuable feedback. As the discussion period concludes tomorrow, we wanted to ensure that our revisions and responses have fully addressed your comments and concerns. If there’s any additional information or clarification you need, please let us know—we would be happy to provide it. We appreciate that all reviewers recognized the novelty of our work, and we have thoughtfully integrated your suggestions to strengthen the manuscript. In its current form, we believe our article presents a novel learning paradigm grounded on an estimation-theoretical approach, with strong potential for impact on both neuroscience and machine learning, now with significantly improved clarity and depth.

---

### Author Response · Authors · 2024-11-18
**General Response to Reviewers' Comments**

We would like to thank all reviewers for their efforts, comprehensive reviews, and constructive comments. **All reviewers have acknowledged the novelty and utility of the new learning framework proposed in our article, describing it as a "novel idea" (Reviewer 1), an "intriguing theoretical foundation" (Reviewer 2), "should be of value to both ML and neuroscience" (Reviewer 3), and "contributes a new perspective" (Reviewer 4).**  Indeed, our work introduces a new theoretical foundation for error broadcast-based learning, grounded in the orthogonality principle of the nonlinear minimum mean square error (MMSE) estimation framework.

This fresh perspective has the potential to enhance our understanding of error broadcast mechanisms and the three-factor learning rule in biological neural networks, offering a plausible explanation for how such processes might occur naturally. Additionally, it provides a pathway for developing flexible and efficient algorithms and architectures for artificial neural networks. Our numerical experiments confirm the functionality and performance of the proposed approach.

We believe that this new framework, based on the orthogonality principle, will benefit both neuroscience and machine learning communities, leading to the development of further computational models, algorithms, and analytical tools. By bridging the gap between biological plausibility and computational effectiveness, our Error Broadcast and Decorrelation (EBD) algorithm opens new avenues for research in both fields.

---

**Summary of Changes in the Revised Article**
We have carefully considered all comments from the reviewers and have made several significant revisions to our manuscript. The revised version has been uploaded for potential review. Below, we outline the main changes:

1.*Reorganization of the EBD Extensions Section:* In response to the reviewers' feedback, we have moved the section on extensions of EBD—which was previously positioned after Section 3—to become a subsection within Section 2. This restructuring enhances the flow of the presentation by grouping related content together, making it easier to follow.

2.*Addition of Appendix on CorInfoMax-EP Approach:* We have included a new appendix section detailing the CorInfoMax-EP approach by Bozkurt et al., 2024. This appendix offers a summarized derivation and description of CorInfoMax networks. It also describes how equilibrium propagation is employed to train CorInfoMax networks in two phases. The inclusion of this appendix aims to facilitate a better understanding of Section 3, where we introduce the CorInfoMax-EBD algorithm for training CorInfoMax networks via error broadcast in a single phase using a three-factor learning rule. This section also provides precise definitions of network and algorithm parameters.

3.*Expanded Discussion in Appendix B.3:* Technical details concerning forward projections have been relocated to Appendix B.3. This new appendix section provides a more extensive discussion on the motivation and algorithmic specifics of this approach, offering readers better insights into the methodology.

4.*Enhanced Explanations Across Sections*:
- Section 2: We have added explanations regarding the use of the orthogonality principle.
- Section 3: Additional discussion on the biological plausibility of CorInfoMax networks compared to MLPs trained with EBD, as presented in Section 2.
- Section 4: Numerical results have been updated, along with a more interpretation of these results.
- Section 5: Conclusions have been extended to reflect the new insights.
- Appendix: Supplementary material on new experiments involving DFA, BP, and CorInfoMax-EBD has been included with their hyperparameters and learning curves, together with runtime comparisons.

5.*Inclusion of Cross-References:*  We have included cross-references to the new appendix sections to guide readers who wish to delve deeper into the technical details.

---

> ### Author Response · Authors · 2024-11-18
> **General Response to Main Comments of Reviewers**
>
> 1. **Bio-plausibility of Entropy and Power Normalizations** :
> One common concern was whether the regularizations introduced in extensions of EBD are bio-plausible.
>
> - **Power Regularization**
> Even when a single sample is used for power calculation, the power regularizer remains effective due to the averaging effect across samples over time. Since all hidden layer activations have decoupled power terms, the gradient-based implementation is biologically plausible in the single-sample mode. The implementations for MLP, CNN, LC, and the CorInfoMax-EBD are biologically plausible with batch sizes of $1$.
>
> - **Entropy Regularization**
> As introduced in Section 2.4.1,  entropy regularization corresponds to the term $\frac{1}{2}$ $\log\det$ $(\mathbf{R}\_{\mathbf{h}}+\varepsilon \mathbf{I} )$.  The derivative of this expression with respect to activations involve the inverse of the correlation matrix $\mathbf{R}_\mathbf{h}$.
>
>     - *MLP,CNN,LC Models:* The current batch based implementations can be converted into more bio-plausible form.
>
>     - *The CorInfomax-EBD implementation of Section 3:*  As described in Appendix D (new appendix), the entropy regularizer is an integral part of the correlative information maximization objective in the CorInfoMax framework, and its online maximization is implemented in a biologically plausible manner. In this implementation, the inverse of the layer-correlation-matrix $\mathbf{B}=\mathbf{R}_\mathbf{h}^{-1}$ manifests as lateral (recurrent) connections of the CorInfoMax network. The CorInfoMax objective gradients for this regularizer function can be implemented based on rank-1 updates on the $\mathbf{B}$ matrix (in addition to the three-factor learning updates from EBD). Therefore, The CorInfoMax-EBD algorithm with a batch size of $1$ updates is bio-plausible. To address concerns about batch (with sizes greater than 1) based operation of CorInfoMax-EBD, we have repeated numerical experiments with a batch size of $1$. The following is the updated form of Table 3, including the CorInfoMax-EBD results for a batch size of $1$:
>
> Table 3: Accuracy results (%) for EP and EBD CorInfoMax algorithms. Column marked with [x] is
> from reference Bozkurt et al. (2024).
> |Data Set|CorInfoMax-EP[x]  (batch size:20)|CorInfoMax-EBD (Ours) (batch size:20)|CorInfoMax-EBD (Ours)(batch size: 1)|
> |:-------:|:--------:|:--------:|:--------:|
> |MNIST|97.58|97.53|94.7|
> |CIFAR-10|50.97|55.79|53.4|
>
> 2. **The use of nonlinear MMSE orthogonality condition:**
> In **linear** MMSE estimation, it is the most fundamental approach to utilize the orthogonality condition, i.e., the uncorrelatedness of errors with input, to obtain the parameters of the estimator. Well known estimators such as Kalman Filters as well as adaptive algorithms are derived based on the use of this orthogonality principle (Kailath et al., 2000).
> Building upon this approach for linear MMSE estimator, we pose a neural network with MSE loss as a parameterized nonlinear MMSE estimator and use the nonlinear orthogonality condition—that is, the uncorrelatedness of the output error components with any nonlinear function of the input—to obtain its parameters. While choosing which nonlinear functions of the input to use is an open problem, for deriving biologically plausible update mechanisms for layer weights and biases, the layer activations are natural choices. The proposed layer-dependent decorrelation objective functions are formed based on this choice. The experiments performed with these objectives confirm the functionality of the corresponding learning method.
>
> Reference:
> Kailath, T. et al., . Linear estimation. Prentice Hall, 2000.
>
> 3. **The EBD framework for different loss functions**
> Our framework utilizes the nonlinear orthogonality property unique to MMSE (Minimum Mean Square Error) estimators. While MMSE is especially suitable for regression, we acknowledge that other tasks may benefit from different loss functions, such as cross-entropy.
> Although we are currently unaware of similar theoretical properties for cross-entropy, this presents an intriguing area for future research. Notably, our numerical experiments (Appendix F.2, "Correlation in Cross-Entropy Criterion-Based Training") show that even with cross-entropy, a similar decorrelation between errors and layer activations occurs, suggesting that decorrelation may be a general feature of the learning process.
> Thus, while MMSE may not be ideal for all tasks, decorrelation could still play a critical role across various loss functions. Further research could deepen our understanding of this learning mechanism.

---

> > ### Author Response · Authors · 2024-11-18
> > **General Response to Main Comments of Reviewers (Part 2)**
> >
> > 4. **Performance and the scalability of the EBD approach**
> > Our numerical experiments for the proposed EBD approach confirm that it achieves similar or better performance compared to other state-of-the-art error-broadcasting approaches. Scalability is a common concern for biologically plausible methods. The recent work by Launay et al., 2020 demonstrated the scalability of the Direct Feedback Alignment (DFA) approach—which is closely related to our proposed EBD method—to complex deep learning tasks. Consequently, the positive scalability results of the closely related DFA method suggest that EBD could similarly extend to complex deep learning applications.
> >
> > Ref:
> > Launay, Julien, et al. "Direct feedback alignment scales to modern deep learning tasks and architectures." Advances in Neural Information Processing Systems 33 (2020): 9346-9360.

---

### Author Response · Authors · 2024-12-03
**A Concise Summary of Our Article’s Contributions and Key Highlights of the Review and Discussion Period**

We thank the reviewers for their valuable efforts and engagement during the discussion period. To provide clarity, we decided to include a concise summary of our contributions along with the reviews and our responses. We have aimed to make this summary as clear and accurate as possible.


# A. Summary of our article's contributions:

Summary of basic contributions/novelty of the proposed "Error Broadcast and Decorrelation" method:
- A biologically plausible alternative to traditional backpropagation, that adresses the credit assignment problem by minimizing correlations between (nolinear functions of) layer activations and output errors. This approach replaces the rigid propagation paths and weight symmetry constraints of backpropagation with a more flexible error propagation mechanism.

-  A **principled method**  for **error broadcast** and **three-factor learning rule** based on the  orthogonality property of nonlinear MMSE estimators. This approach holds immediate potential implications for neuroscience, where neuromodulator-based error broadcasting and three-factor learning rules are experimentally observed phenomena. Additionally, the proposed "decorrelation" paradigm offers promising avenues for the development and analysis of machine learning algorithms and architectures.
-  Error projection weights determined by the **cross-correlation** between the output errors and the layer activations, which are **dynamically updated** by Hebbian rule, as opposed to random fixed weights of DFA,

-   Learning updates involving **arbitrary nonlinear functions** of layer activities, encompassing a family of three-factor learning rules,

- The use of the proposed EBD mechanism successfully transforms the learning paradigm for biologically plausible networks from a two-phase, time-contrastive approach (CorInfoMax-EP) to a single-phase, three-factor learning method (CorInfoMax-EBD), which is demonstrated to achieve comparable and better performance even for a batch size of $1$,

- The proposed EBD algorithm performs similar or better performance than the existing error broadcasting methods,


-   The option to project layer activities forward to the output layer.

In summary, our approach provides a **theoretical grounding** for the **error broadcasting** mechanism and suggests ways to its effectiveness in training networks. This work opens avenues for several extensions in both neuroscience and machine learning, including scalability to larger networks and tasks, as well as adaptation to diverse loss functions, as emphasized in our revised article.

# B. Strengths reflected by reviewers:

All reviewers have acknowledged the novelty and utility of the new learning framework proposed in our article:

- Reviewer 1-qRs4: Describes it as a novel and very interesting idea and stating "better theoretical illustration and performance than DFA".

- Reviewer 2- P7w2: Positions it as an approach with compelling biological motivation ... circumventing the weight transport problem...introducing an innovative algorithm.

- Reviewer 3- JH3K: "The theoretical building block in which this is built  is interesting and in my view certainly deserves the attention given by the authors. Its implementation - and therefore this paper - should be of value to both ML and neuroscience researchers.", " paper's central concepts are interesting and should be encouraged in the field of computational neuroscience."

- Reviewer 4- VBft: This work contributes a new perspective for measurement of a final optimum of network training based upon the MMSE estimator in a principled manner based upon the orthogonality of error and layer-activations.

---

> ### Author Response · Authors · 2024-12-03
> **A Concise Summary of Our Article’s Contributions and Key Highlights of the Review and Discussion Period (Part 2)**
>
> # C. Concerns of reviewers, in reviews/discussions, and how they are addressed in the revised article and responses.
>
> - **biological plausibility of entropy and power regularization:**
>
> **Reviewer P7w2:** "*The paper’s claim of biological relevance is weakened by the batch learning requirement., ..., Is there a potential for an online version of your algorithm that eliminates the need for batch learning?*". **Reviewer JH3K:** "*To what extent does EBD depend on batch size? It seems like it would require large batches to get a good correlation estimate.*"  **Reviewer VBft:** "*a power normalizing and entropy encouraging mechanism is added ... not discussed whether these are reasonable mechanisms within a biologically plausible context.*"
>
>   In our response and the revised article, we clarified that CorInfoMax architecture’s lateral connections provide entropy regularization and even operate in batch of size 1, increasing its biologically plausibility. In the revised article we included CorInfoMax-EBD numerical examples with batch-1.
>
> The reviewers found the explanations in the revision/response and the new example as the satisfactory answer:
>         **Reviewer JH3K:** "*I also think that the fact CorInfoMax-EBD can operate in a batch size = 1 does improve its bio plausibility.*"". **Reviewer P7w2:** "*Thank you for your explanation about the biological plausibility of batch learning.*".
>
> ---
> - **result comparability with other error-broadcasting mechanisms:**
>
> **Reviewer P7w2:** "*Why does BP in Table 2 show such a low performance?*". **Reviewer VBft:** "*the original work by Clark et al. has no learning rate scheduler and far fewer training hyperparameters in general. This suggests that the comparison is entirely inappropriate.*".
>
> To address these concerns, we re-implemented the compared methods—specifically Backpropagation (BP) and Direct Feedback Alignment (DFA)—ensuring they were trained under the same conditions as our proposed EBD algorithm. This adjustment standardizes the optimization settings across all methods, enabling a fairer and more accurate comparison. We also added runtime comparisons in appendix, for better comparison.
>
> By aligning the training conditions, we enhance the validity and soundness of the results presented in the paper, addressing the reviewers' critiques and improving the reliability of our findings: **Reviewer JH3K:** "*Performance wise, I'm impressed with the additional simulations that have been carried out and am convinced by the emperical results.*". **Reviewer VBft:** "*The results of Table 1 are much improved by a re-implementation and are now more comparable... Thank you for the addition of runtime comparisons, these are additionally useful.*"
>
> ---
> - **presentation flow of the article:**
>
> **Reviewer JH3K:** "*The paper is rather dense, and I worry that it harms its accessibility ... structure/delivery of the paper makes it difficult to grasp for a non-expert*". **Reviewer VBft:** "*The description of this work’s method is comprehensive but requires a reader to go back and forth to understand it well ... the paper in general is too heavy on the methods aspects*".
>
> To address these concerns, we have reorganized the article's sections to improve its readability and ensure a smoother flow. Specifically, we relocated the variations on MLP to Section 2 and moved detailed explanations about forward broadcast to the appendix, making the main text less dense. Additionally, we expanded discussions and added clarifying comments, particularly in Section 3 (CorInfoMax-EBD), the conclusions and limitations to enhance the accessibility of the content. We included a more thorough discussion of the biological plausibility of various components of our algorithms, such as the regularizations and the structure of the CorInfoMax-EBD network. These changes aim to make the paper more comprehensible, even for readers who are not experts in the field.
>
> As appreciated by the reviewers:    **Reviewer VBft:** "*Modifications to the explanation of EBD are appreciated and it is now a clearer read.*"
>
> ---
> - **discussion of alternative loss functions:**
>
> **Reviewer qRs4:** "*Is optimal MMSE estimator the best objective of an arbitrary defined network? (since different tasks might have different loss functions)*". **Reviewer VBft:** "*The implications for an alternative loss function (CCE) are now present in an appendix, however this appendix is never referred to in the main text.*"
>
> For this, we mentioned that our proposed framework exploits the nonlinear orthogonality property specific to MMSE (Minimum Mean Square Error) estimators. However, our numerical experiments in Appendix F.2 reveal that even with cross-entropy loss, errors and layer activations decorrelate, suggesting this phenomenon is a fundamental aspect of learning, not limited to the MMSE objective. We also included an additional paragraph regarding this in the conclusion, discussing possible extensions to different losses.

---

> > ### Author Response · Authors · 2024-12-03
> > **A Concise Summary of Our Article’s Contributions and Key Highlights of the Review and Discussion Period (Part 3)**
> >
> > - **the clarification of the narrative about performance and scalability:**
> >
> > **Reviewer VBft:** "*At present, the paper is presented as if this novel algorithm does or at least should be expected to scale ... I would recommend pitching this more as potential avenue to explore, with less strong claims in the abstract, and main text/conclusion ... This would complete the written side of this work to a standard of having an acceptance score.*" **Reviewer qRs4:** "*either the authors should provide more evidences on the scale up capability of this method, or the authors should revise the current manuscript in a better narrative on the limitations and open-questions.*"
> >
> >  In the revision we addressed this concern by clearly stating that scalability is a future direction rather than an immediate claim of this paper, and the performance comparison is against the other error broadcast approaches and CorInfoMax-EP approach. We believe our method gives both a theoretical ground and imposing numerical results. In line with these, we modified key sections of the paper, including the Abstract, Introduction, and Limitations; by clearly outlining the contributions and novelties, together with the drawbacks of our method and positioning clearer while setting the stage for future research that builds on these findings.
> >
> > We believe these revisions make the narrative more precise and aligned with the expectations expressed by the reviewers for a higher evaluation, better reflecting the strengths of the method within its scope.
> >
> > ---
> >
> > - **critiques about our theoretical framework:**
> >
> >
> > **Reviewer P7w2:** "*Applying the orthogonality principle on a per-neuron basis appears to be incorrect. Therefore, the assertion that you have $m$ x $n$ conditions is flawed, and the problem of finding the optimal estimator remains highly under-constrained.*"
> >
> > We responded to this by stating that applying the orthogonality principle on a per-neuron basis is correct and well-supported by statistical theory. Specifically, each neuron and error component are treated as random variables, and their orthogonality is defined through the correlation-based inner product in the Hilbert space of random variables (with realizations varying across dataset samples). This aligns with the nonlinear MMSE orthogonality condition, which provides multiple equations to constrain the parameter space of the network, resulting in multiple orthogonality conditions. While these conditions are necessary, we acknowledged they are not sufficient for fully determining the parameter set. However, implicit and explicit regularization effects, such as entropy and sparsity constraints, guide optimization effectively. Furthermore, we proposed extensions using nonlinear mappings of activations to increase the number of orthogonality conditions, though our numerical experiments showed this was not necessary for achieving desirable solutions. We believe this detailed clarification addresses the reviewer’s comments.
> >
> > ---
> >
> > - **comparison with distinct methods**
> >
> > **Reviewer qRs4:** "*The experiments show EBD is only slightly better than DFA, while it is not comparable with other SOTA biologically plausible methods. (e.g. Hebbian base method)*"
> >
> > Our experiments demonstrate that our framework achieves performance comparable to or surpassing other biologically plausible error-broadcasting methods, such as those proposed by Clark et al. (2021), on datasets of similar scale, using the same architectures and training objectives. As detailed in the Appendix and Introduction, our study specifically compares the EBD method with other error-broadcasting approaches, including Direct Feedback Alignment (DFA) and the three-compartment neuron model of CorInfoMax-EP (Bozkurt et al., 2023). This comparison is grounded in a rigorous framework of biological plausibility, highlighting features such as the incorporation of a three-factor learning rule, the absence of weight symmetry requirements, and the ability to perform online learning with dynamic updates to all layers for each new sample in the CorInfoMax-EBD framework.
> >
> > We believe our method represents a significant theoretical advancement in error-broadcasting frameworks, offering meaningful implications for the development of biologically plausible neural networks. The CorInfoMax-EBD approach introduced in our study is a principled method where many biologically observed phenomena naturally emerge, including lateral and feedback connections, three-factor learning rules, and more realistic multi-compartment neuron models. While the proposed framework may not achieve the performance of models lacking these biologically grounded components, its ability to capture key aspects of biological reality is an important and favorable feature that underscores its relevance and potential impact.

---

### Meta-Review · Area_Chair_bcgu · 2024-12-26

**Metareview:**

This paper introduces the Error Broadcast and Decorrelation (EBD) algorithm, a novel learning framework for neural networks that addresses the credit assignment problem by directly broadcasting output errors to individual layers. The key theory is based on the orthogonality property of minimum mean square error (mmse) estimators, which states that estimation errors are orthogonal to any nonlinear function of the input. The authors demonstrate EBD's performance on MNIST and CIFAR-10 benchmarks.

Positive:
- new theoretical foundation for error broadcasting based on MMSE orthogonality principles
- achieves competitive performance compared to existing error-broadcast methods on very tiny problems.
- alternative to backpropagation without weight transport
- integration with CorInfoMax shows potential for single-phase learning

Weaknesses:

- limited empirical validation on larger-scale problems and architectures
- insufficient discussion of computational complexity trade-offs
- biological plausibility of power normalization and entropy components needs stronger justification
- paper structure and presentation could be clearer, especially regarding extensions

The authors have provided comprehensive feedback on the points raised by the reviewers. The skeptical reviewers remained unconvinced of the soundness of the assumptions made on the theoretical claims of the paper. I recommend rejection and encourage the authors to integrate the points made by the reviewers into a revised version for the next venue.

**Additional Comments On Reviewer Discussion:**

The main points raised during the discussion were:

- Fairness of comparisons with previous methods (raised by Reviewer VBft) - Authors addressed this by re-implementing baseline methods under identical conditions with proper hyperparameter tuning.
- Theoretical validity of per-neuron orthogonality (raised by Reviewer P7w2) - Authors tried to address this in detailed mathematical form but some concerns remained.
- Biological plausibility of batch operations (raised by multiple reviewers) - Authors demonstrated CorInfoMax-EBD can operate with batch size 1.
- Paper organization and clarity (raised by multiple reviewers).

The authors were highly responsive and made substantial revisions to address most concerns. The main theoretical concerns remained unresolved, but the authors convinced the reviewers about the practicality of their algorithm on small benchmarks.

---

### Decision · Program_Chairs · 2025-01-22

Reject